# Self-regulating arousal via pupil-based biofeedback

Sarah Nadine Meissner [1] ✉, Marc Bächinger[1], Sanne Kikkert [1,2,3], Jenny Imhof [1,2], Silvia Missura [1], Manuel Carro Dominguez [1,2] & Nicole Wenderoth [1,2,4] ✉

The brain's arousal state is controlled by several neuromodulatory nuclei known to substantially influence cognition and mental well-being. Here we investigate whether human participants can gain volitional control of their arousal state using a pupil-based biofeedback approach. Our approach inverts a mechanism suggested by previous literature that links activity of the locus coeruleus, one of the key regulators of central arousal and pupil dynamics. We show that pupil-based biofeedback enables participants to acquire volitional control of pupil size. Applying pupil self-regulation systematically modulates activity of the locus coeruleus and other brainstem structures involved in arousal control. Furthermore, it modulates cardiovascular measures such as heart rate, and behavioural and psychophysiological responses during an oddball task. We provide evidence that pupil-based biofeedback makes the brain's arousal system accessible to volitional control, a finding that has tremendous potential for translation to behavioural and clinical applications across various domains, including stress-related and anxiety disorders.

The brain's arousal state is controlled by key neuromodulatory nuclei including the noradrenergic (NA) locus coeruleus (LC), dopaminergic substantia nigra/ventral tegmental area (SN/VTA), serotonergic dorsal raphe nucleus (DRN) and the cholinergic nucleus basalis of Meynert (NBM)[1–4]. Previous research indicates that under constant lighting conditions, pupil size is an indirect 'indicator' of the brain's arousal state. Arousal-related neuromodulatory systems have been directly or indirectly linked to non-luminance related changes in pupil size[5–7] with the strongest evidence for the LC-NA system[7–11]: selective chemogenetic or optogenetic activation of the LC causes substantial pupil dilation in mice[11–13]. Further, another animal study suggests the involvement of both cholinergic and noradrenergic systems in pupil dynamics[7]. However, only prolonged pupil dilations during movement were accompanied by sustained cholinergic activity, while moment-to-moment pupil fluctuations during rest closely tracked noradrenergic activity[7]. Furthermore, noradrenergic activity preceded cholinergic activity

relative to peak pupil size[7] and can depolarize cholinergic neurons[1], suggesting that noradrenergic activity drives cholinergic activation. In humans, functional magnetic resonance imaging (fMRI) demonstrated that LC activity correlates with pupil size, both at rest and during various tasks including the oddball paradigm[6,8,9,14].

Theories based on intracranial recordings in animals suggest that the LC-NA system modulates functional circuits related to wakefulness, sleep[15–19] and cognitive processes relevant for task engagement and performance[4,18,20–22]. These theories postulate that LC neurons exhibit tonic and phasic discharge patterns, where tonic activity is thought to closely correlate with the brain's arousal state (that is, high tonic LC activity is associated with high arousal) and phasic discharge facilitating behavioural responses to task-relevant events[4]. Probing these task-relevant processes with a two-stimulus oddball paradigm revealed that phasic LC responses and task performance depend on the level of tonic activity. For instance, recordings in monkeys showed that

[1]Neural Control of Movement Laboratory, Department of Health Sciences and Technology, ETH Zurich, Zurich, Switzerland. [2]Neuroscience Center Zurich, University and ETH Zurich, Zurich, Switzerland. [3]Spinal Cord Injury Center, Balgrist University Hospital, University of Zurich, Zurich, Switzerland. [4]Future Health Technologies, Singapore-ETH Centre, Campus for Research Excellence and Technological Enterprise (CREATE), Singapore, Singapore. ✉e-mail: sarah.meissner@hest.ethz.ch; nicole.wenderoth@hest.ethz.ch

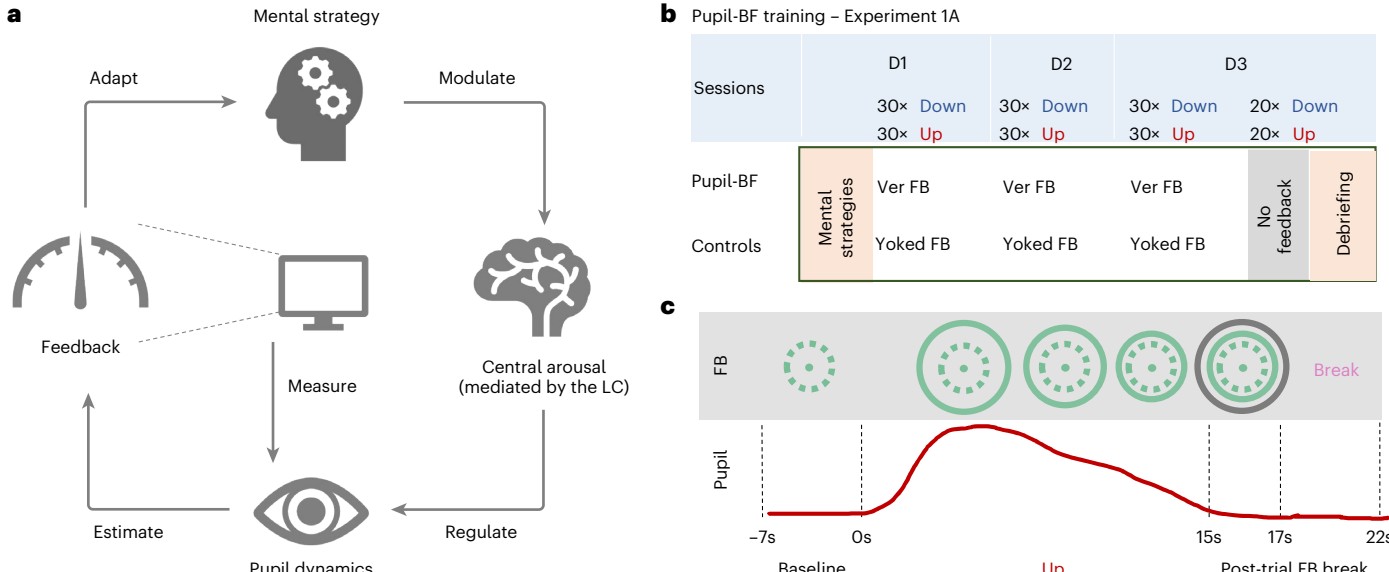

**Fig. 1 | Pupil-BF training. a**, Participants apply mental strategies that are believed to modulate the brain's arousal levels mediated by nuclei such as the LC. Pupil size was measured by an eye tracker and fed back to the participant via an isoluminant visual display. **b**, In experiment 1A, healthy volunteers were informed about potential mental strategies of arousal regulation and then participated in 3 days (D1, D2, D3) of upregulation and downregulation trainings (30 trials each) while receiving either veridical pupil feedback (Ver FB) (pupil-BF group) or visually matched input/yoked feedback (Yoked FB) (control groups I and II). At the end of day 3, all participants performed 20 Up and 20 Down trials without receiving any feedback and were debriefed on which strategies

they have used. **c**, Example trial of the experiment. Each trial consisted of (1) 7 s baseline measurements, (2) 15 s modulation phase where the pupil-BF group sees a circle that dynamically changes its diameter as a function of pupil size (veridical feedback), (3) 2 s of colour-coded post-trial performance feedback (green, average circle size during modulation; black, maximum (Up) or minimum (Down) circle size during modulation) and (4) 5 s break. The upper panel shows an example of what participants would see on their screen, while the red line in the lower panel indicates measured pupil size. Note that the control groups I and II see a circle that changes independently of pupil size but resembles that for a participant in the pupil-BF group.

elevated tonic LC activity is associated with reduced phasic responses and detection performance of salient oddball stimuli[23–26].

Regulating arousal is challenging and existing approaches in humans rely mainly on pharmacological agents with side effects. Here we investigated an approach utilizing the mechanistic link between the brain's arousal state and pupil dynamics via an innovative pupil-based biofeedback (pupil-BF) approach. Only a few previous studies have trained volunteers to self-regulate pupil size with varying degrees of success[27–30]. Volitional pupil size downregulation is especially difficult to acquire[30]. Our main idea is that participants apply different arousing or relaxing mental strategies while receiving online pupil-BF (Fig. 1a). Considering the strong link between pupil diameter and LC-NA activity, we derived several hypotheses from the rodent literature and tested whether pupil self-regulation will affect specific aspects of neural processing[20,31] and cardiovascular function[32], which is influenced by the LC through its projections to autonomic control structures in the brainstem and spinal cord[18]. Specifically, we hypothesized that (1) pupil-BF allows participants to discover suitable mental strategies for volitional up- vs downregulation of pupil size, such self-regulation (2) is associated with up- vs downregulating activity in brain regions involved in arousal control including the LC, and (3) causes systematic changes in cardiovascular parameters. To further probe the link to the LC-NA system shown to be associated with behavioural and psychophysiological measures of the oddball task, we combined our pupil-BF approach with an auditory version of this paradigm. We hypothesized that (4) self-regulating pupil size modulates stimulus detection behaviour and pupil dilation responses to oddball stimuli.

Numerous neuro- or biofeedback studies have shown that providing feedback enables humans to gain remarkable control over specific body functions[33–36]. Acquiring a suitable mental strategy, explicitly or implicitly, is crucial for effective self-regulation and feedback can

indeed facilitate this process[33,36]. Using our pupil-BF training, healthy volunteers learned to volitionally up- and downregulate their pupil size (experiments 1A and B). Crucially, combining pupil self-regulation with fMRI and cardiovascular measurements, we observed systematic modulation of (1) activity in arousal-regulating brainstem centres including the LC and SN/VTA and (interconnected) cortical and subcortical brain regions (experiment 2) and (2) heart rate (experiments 1B and 2). Finally, self-regulating pupil size modulated behavioural performance and psychophysiological markers of LC-NA activity during an auditory oddball task (experiment 3).

## Results

### Experiment 1. Pupil-BF training enables pupil self-regulation

Participants were randomly assigned to either a pupil-BF ($n = 28$) or control group ($n = 28$) and underwent 3 days of pupil-BF training (experiment 1A). Before training, all participants received instructions on mental strategies derived from previous research[27,30,37–39], such as imagining emotional (that is, fearful or joyful) situations to upregulate pupil size (Up); or relaxing, safe situations together with focusing on their body and breathing to downregulate pupil size (Down; see Supplementary Table 1 for strategies). Each of the three sessions consisted of 30 Up and 30 Down trials, with participants self-regulating their pupil for 15 s (Fig. 1c). The pupil-BF group received isoluminant visual feedback on pupil size in quasi real-time during regulation and average performance feedback after regulation (post-trial feedback; Fig. 1c). The control group received feedback from a randomly selected participant of the pupil-BF group (that is, receiving the same visual input as the pupil-BF group) and participants were instructed to focus on their mental strategies, ensuring to control learning effects due to mental rehearsal[40]. Importantly, the control group was not aware of the existence of a pupil-BF group. At the end of day 3, all participants underwent an additional 'no-feedback' session (20 Up and 20 Down

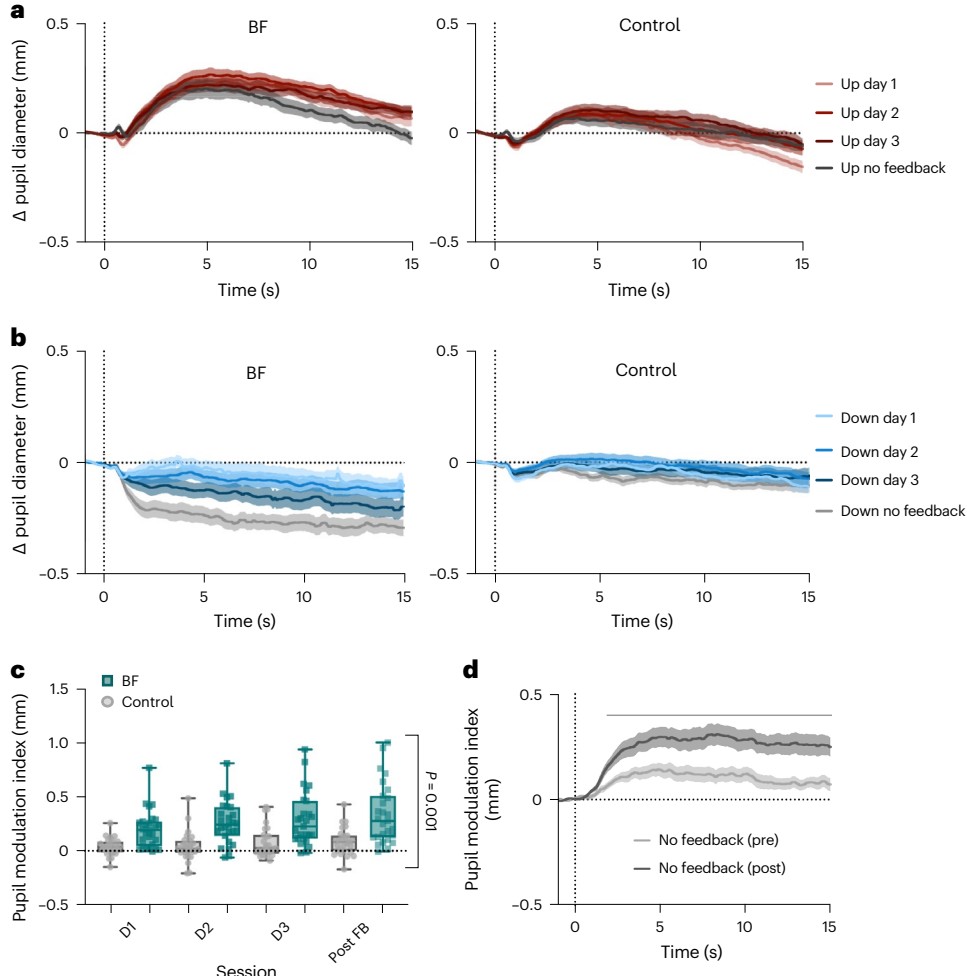

**Fig. 2 | Pupil-BF training results. a,b,** Average changes in pupil size during 15 s upregulation (**a**) and downregulation (**b**) are shown for the pupil-BF ($n = 27$) and initial control group ($n = 27$) for training sessions on days 1, 2 and 3, and for the no-feedback post-training session of experiment 1A. **c,** The pupil modulation index reflects the difference between the average pupil size during the two conditions (Up–Down) and is shown for each session (days 1, 2 and 3, and for the no-feedback post-training session) and group (initial control group vs pupil-BF group of experiment 1A; dots and squares represent individual participants). Pupil modulation indices were generally higher in the pupil-BF group ($n = 27$) compared with the initial control group ($n = 27$; robust ANOVA, main effect of group: $F_{(1,21.58)} = 21.49$; $P = 0.001$, $\eta_p^2 = 0.50$; other main effects/interaction

$P \geq 0.07$) **d,** Time series of pupil modulation index measured during the no-feedback session before (pre, light grey) and after pupil-BF training (post, dark grey) in experiment 1B (independent cohort, $n = 25$). Solid black line at the top indicates clusters of significantly higher modulation indices after training compared with before (SPM1D repeated-measures ANOVA; main effect session; $z^* = 11.84$; largest cluster $P = 0.037$; smallest cluster $P = 0$). Shaded areas indicate s.e.m. Boxplots indicate median (centre line), 25th and 75th percentiles (box), and maximum and minimum values (whiskers). For a replication of results in control group II, see Supplementary Fig. 3. All post-hoc tests were two-tailed and corrected for multiple comparisons. For more detailed information on statistical parameters, see Supplementary Table 5.

trials) using the same self-regulation strategies as during training, but without receiving feedback.

Descriptively, participants of both groups ($n = 27$ in each group for final analyses) showed some ability to upregulate pupil size (Fig. 2a), with greater upregulation observed in the pupil-BF group. The ability to downregulate pupil size became gradually more successful during training in the pupil-BF group, while this ability was generally reduced in the control group (Fig. 2b; Supplementary Fig. 1 displays statistical comparisons of up- and downregulation compared to baseline). Importantly, the pupil-BF participants maintained their self-regulation abilities during 'no-feedback trials', indicating that they acquired a transferable skill that can be applied without constant feedback (Fig. 2a,b; grey lines).

Pupil self-regulation was quantified calculating a pupil modulation index (Up–Down; that is, the difference between pupil diameter changes in the two conditions), which was significantly larger in the pupil-BF than in the control group (Fig. 2c; $F_{(1,21.58)} = 21.49$;

$P = 0.001$, $\eta_p^2 = 0.50$; 95% confidence interval (CI)$\eta_p^2$ (0.17, 0.67)). A control analysis confirmed that these group differences were not driven by differences in absolute pupil size at baseline (all $P \geq 0.48$; Supplementary Fig. 2a). These findings indicate that training with veridical BF was superior to mental rehearsal and that the effects were not driven by visual input alone. We tested an additional control group (control II) that received yoked feedback but believed that it was veridical BF (see methods) and thereby controlled for additional motivational and perceived success factors. We obtained similar results as with control group I (Supplementary Fig. 3a–c).

Next, we conceptually replicated these results in an independent pupil-BF cohort (experiment 1B). Twenty-six participants ($n = 25$ for final analyses) followed a similar 3-day training protocol with a no-feedback session before and after training (Supplementary Fig. 4a,b). Pupil modulation index time series (Up–Down) were significantly higher after training compared with before training during most of the modulation phase (main effect session: significant differences,

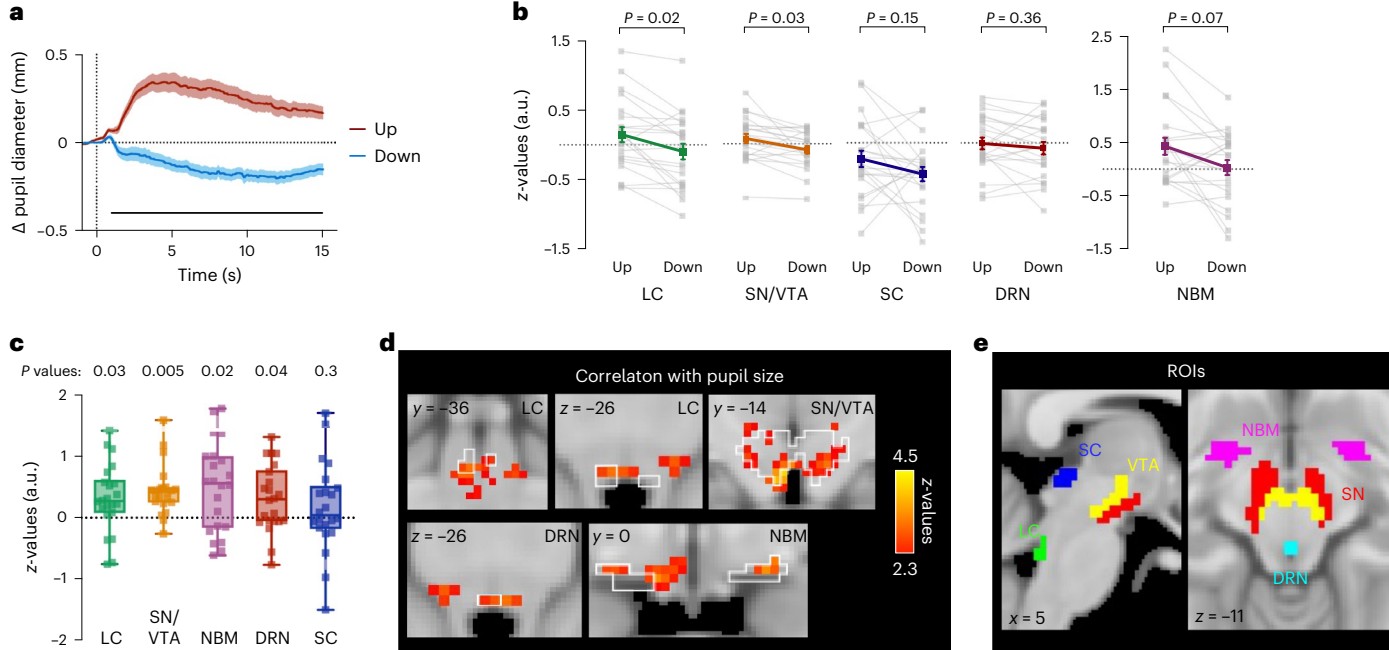

**Fig. 3 | Brainstem fMRI results. a**, Pupil size changes averaged across participants for Up and Down trials showing successful pupil size self-regulation during brainstem fMRI recording. The solid black line at the bottom indicates a cluster of significantly higher baseline-corrected pupil sizes during Up than during Down trials (two-tailed SPM1D paired-samples $t$-test; $P = 0$; $z^* = 3.45$). **b**, Activity during Up versus Down phases of pupil self-regulation in the different ROIs. Statistical comparisons ($n = 22$) revealed significant effects (Up > Down) in the LC and the SN/VTA but not in the SC and DRN (two-tailed paired-samples $t$-test). Results for the NBM (two-tailed Wilcoxon signed-rank test) did not survive multiple comparison correction. **c**, Correlation between continuous pupil size changes and BOLD response changes shown as $z$-values for the different ROIs. Statistical comparison (against 0; two-tailed; $n = 22$) revealed significant effects for the LC, SN/VTA, NBM and DRN (one-sample $t$-test) but not for the SC (Wilcoxon signed-rank test). All ROI analyses in **b** and **c** were sequential

Bonferroni-corrected for multiple comparison. Squares represent individual data **d**, Top: correlation analysis revealed that LC BOLD activity covaries significantly with continuous changes in pupil diameter (GLM; cluster-corrected for multiple comparisons at $z = 2.3$; $P < 0.05$). Bottom: brainstem areas other than the LC exhibited a significant correlation between changes in pupil diameter and BOLD activity (GLM; cluster-corrected for multiple comparisons at $z = 2.3$; $P < 0.05$). For a complete overview of regions, see Supplementary Table 2b. White outlines in **d** indicate different brainstem[43–49] and basal forebrain regions[98]. **e**, A-priori-defined ROIs in the brainstem[43–49] and basal forebrain[98] in MNI space. Boxplots indicate median (centre), 25th and 75th percentiles (box), maximum and minimum values (whiskers). Shaded areas and error bars indicate s.e.m. Post-hoc comparisons were corrected for multiple comparisons. For detailed information on statistical parameters, see Supplementary Table 5.

1.9 s–15 s after modulation onset; all $P < 0.05$; Fig. 2d; Supplementary Fig. 4e displays condition effects). Interestingly, before training, the modulation index was already significantly higher than 0 (mainly between 1.7 s and 10.4 s after modulation onset; all $P < 0.05$; Supplementary Fig. 4d), indicating the participants' ability to voluntarily modulate pupil size only with instructions on mental strategies, an ability that was further improved by pupil-BF training.

In experiments 1A and B, we found that healthy volunteers can learn to self-regulate pupil size. Significant differences in self-regulation capabilities between (1) pupil-BF and control groups, and (2) before versus after training performance in a separate pupil-BF cohort highlight the benefits of biofeedback in improving volitional pupil size control.

## Experiment 2. Pupil self-regulation combined with fMRI

Previous studies have repeatedly linked non-luminance-related pupil size changes to activity changes in arousal-regulating centres, including the LC[6,8–11]. Here we tested the hypothesis that self-regulating pupil size is associated with activity changes in these regions. Twenty-five trained pupil-BF participants from experiments 1A and B performed pupil up- vs downregulation during two fMRI sessions, one measuring activity across the whole brain and one specifically measuring brainstem activity (counterbalanced session order). Participants performed 15 s of pupil size modulation and received post-trial performance feedback. Pupillometry data confirmed that participants were able to self-regulate pupil size (significant differences between Up and

Down from 965.72 ms (whole-brain, $P < 0.001$) and 983.72 ms (brainstem, $P < 0.001$) to the end of modulation, respectively; Figs. 3a and 4a).

## Self-regulation is linked to brainstem activity changes

On the basis of previous findings, we predefined the LC, DRN, NBM and SN/VTA as regions of interest (ROI) that substantially contribute to arousal control[1–4]. In addition, the LC, NBM and DRN have been linked to pupil size changes[5,7,41] even though it is still debated whether the NBM and DRN modulate pupil size directly or via the LC-NA system. Anatomically, pupil size is controlled by the tone of (1) the dilator pupillae muscle receiving innervation via noradrenergic sympathetic neurons, and (2) the constrictor pupillae muscle receiving innervation via cholinergic parasympathetic neurons[18]. Thus, pupil diameter reflects the relative activity of these opposing outputs of the autonomic nervous system[18,31]. The LC is assumed to modulate the pupil by (1) facilitating sympathetic activity via projections to the intermediolateral (IML) cell column of the spinal cord, and (2) inhibiting parasympathetic activity via its projections to the Edinger–Westphal nucleus (EWN[18,31] reviewed in ref. 41). Alternatively, a parallel activation of the LC and the sympathetic nervous system through a third player, the rostral ventrolateral medulla, has been discussed in the literature[42]. The superior colliculus (SC) is another candidate implicated in non-luminance-related pupil control during multisensory integration and orienting responses[10,41], assumed to influence parasympathetic activity through direct and indirect projections to the EWN via the mesencephalic cuneiform nucleus (MCN), and sympathetic

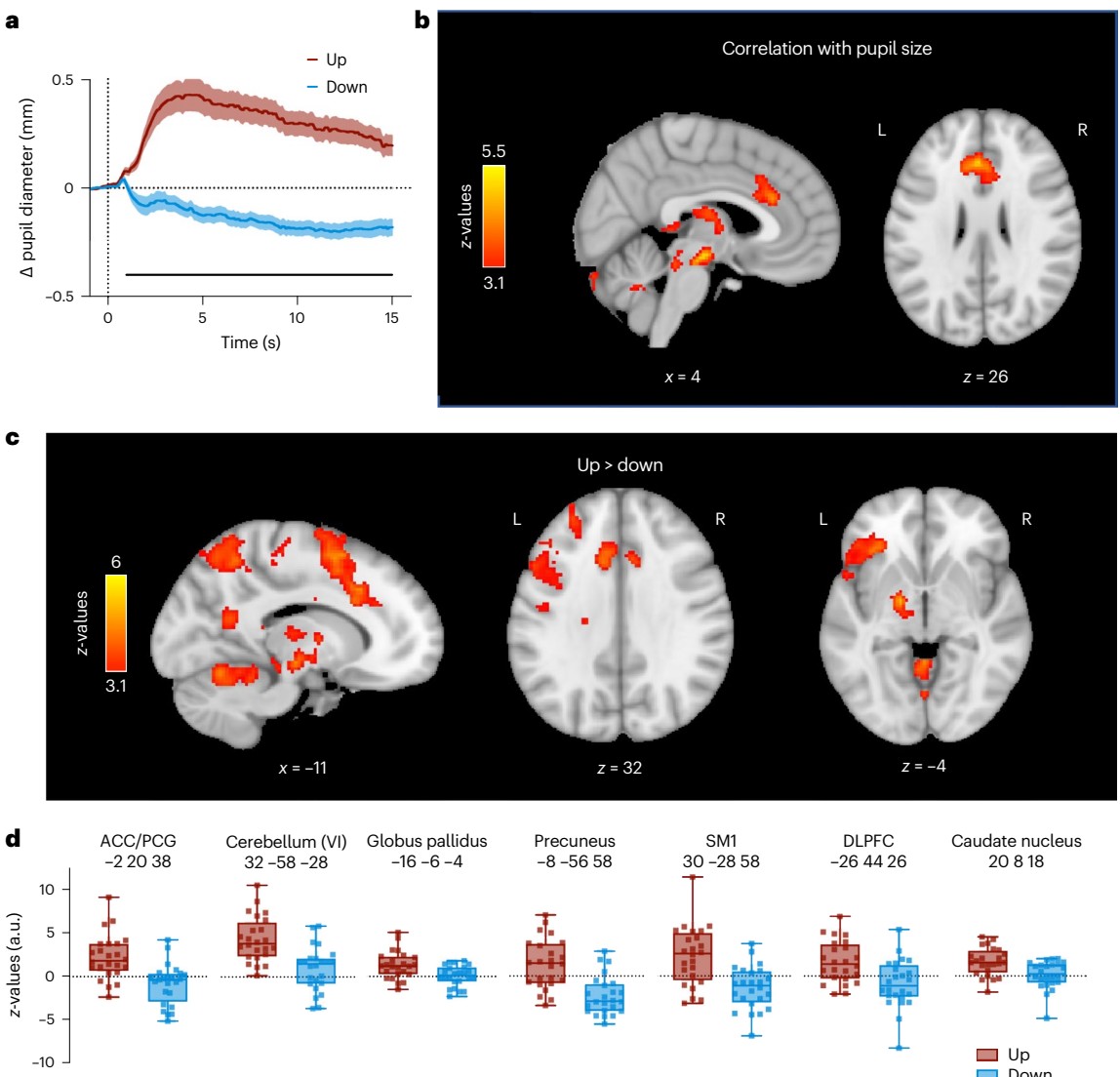

**Fig. 4 | Whole-brain fMRI results. a**, Changes in pupil size averaged across all participants for Up (red) and Down (blue) trials showing successful self-regulation of pupil size during whole-brain fMRI recordings. The solid black line at the bottom indicates a cluster of significantly higher baseline-corrected pupil sizes during Up than during Down trials (two-tailed SPM1D paired-samples *t*-test; $P = 0$; $z^* = 3.38$). **b**, Whole-brain maps showing brain regions where BOLD activity correlates with pupil size changes throughout the fMRI runs (GLM). **c**, Whole-brain maps depicting brain regions that showed significant activation during Up (as compared to Down) trials (GLM). All activation maps in **b** and **c** are thresholded at $z > 3.1$ and FWE-corrected for multiple comparisons using a cluster significance level of $P < 0.05$. **d**, Estimated BOLD response represented by *z*-values for Up vs rest and Down vs rest extracted from the peak voxel of each significant cluster shown in **c** ($n = 24$). Boxplots indicate median (centre line), 25th and 75th percentiles (box), and maximum and minimum values (whiskers). Squares indicate individual participants ($n = 24$). Shaded areas indicate s.e.m.

activity through projections to the IML via the MCN[41]. Therefore, we also included the SC as a control ROI in our analysis (Fig. 3e).

After preprocessing brainstem data (that is, removing physiological noise), we extracted mean signal intensities during Up > rest and Down > rest from the LC, SN/VTA, DRN, NBM and SC ($n = 22$). We observed significantly higher LC activation during pupil upregulation compared with downregulation ($t_{(21)} = 3.40$; $P = 0.015$, Cohen's $d = 0.73$, 95% CI $d$ (0.25, 1.19)). Similar results were found for the SN/VTA ($t_{(21)} = 2.96$, $P = 0.03$, $d = 0.63$, 95% CI $d$ (0.17, 1.08)). Activation differences in all other ROIs did not survive correction for multiple comparisons (NBM: $z = 2.26$, $P = 0.072$, $r = 0.48$) or did not reach significance (SC and DRN, all $P \geq 0.15$; Fig. 3b). For additional control analyses extracting mean signal intensities from masks covering (1) the complete brainstem including the midbrain, pons and medulla oblongata, and (2) the 4th ventricle as well as for all ROI analyses without spatial smoothing of the data, see Supplementary Figs. 5a–c and 6.

Further correlating pupil size modulations throughout fMRI runs (pupil shifted by 1 s) with continuous blood oxygenation level-dependent (BOLD) time series extracted from the predefined ROIs revealed a significant association in the LC ($t_{(21)} = 2.64$; $P = 0.03$, $d = 0.56$, 95% CI $d$ (0.11, 1.01)), SN/VTA ($t_{(21)} = 5.13$; $P = 0.005$, $d = 1.09$, 95% CI $d$ (0.56, 1.62)), NBM ($t_{(21)} = 3.21$; $P = 0.02$, $d = 0.68$, 95% CI $d$ (0.21, 1.14)) and DRN ($t_{(21)} = 2.69$; $P = 0.04$, $d = 0.57$, 95% CI $d$ (0.12, 1.02)). SC activity was not significantly related to pupil size ($P = 0.32$). For additional analyses on unsmoothed data, see Supplementary Fig. 6.

Next, complementing our a-priori-defined ROI analysis, we conducted exploratory analyses for Up > Down contrasts across the brainstem. However, no region in the brainstem survived cluster corrections for multiple comparisons (for uncorrected results $P < 0.05$, see Supplementary Fig. 5d and Supplementary Table 2a; Supplementary Fig. 5e displays Down > Up results). Conducting correlations of continuous pupil diameter with continuous BOLD activity changes

for every voxel in the brainstem, we found significant correlations in regions covering the SN, VTA, LC, NBM and the DRN (cluster-corrected, Fig. 3d). In addition, pupil size correlated with activity in other brainstem regions involved in arousal and autonomic regulation[43–49] (Supplementary Table 2b).

In summary, our brainstem fMRI analyses revealed that pupil self-regulation is linked to activity changes in brainstem regions involved in arousal and autonomic regulation, including the LC-NA system, with activation during upregulation and deactivation during downregulation. Although the LC is not the only area modulated by pupil self-regulation, our results demonstrate that the reported mechanism of LC activity driving pupil size can be inverted in the context of pupil-BF, making the brain's arousal system accessible to voluntary control.

### Self-regulation is linked to (sub-)cortical activity changes

Investigating the effects of pupil self-regulation on cortical and subcortical structures, we contrasted Up > Down phases in our whole-brain fMRI data ($n = 24$). Up- vs downregulation was associated with significantly higher activation in various brain regions closely connected to the LC, including the dorsal anterior cingulate cortex (dACC)/paracingulate gyrus (PCG), dorsolateral prefrontal cortex (DLPFC), orbitofrontal cortex (OFC), precuneus and thalamus. In addition, we observed significant activation in primary somatosensory regions (SM1), basal ganglia (globus pallidus, caudate nucleus) and cerebellum (Fig. 4c,d and Supplementary Table 3a).

Examining in which brain areas BOLD activity covaried with pupil size throughout the task, we observed significant effects in the dACC, precuneus, thalamus, globus pallidus and cerebellum (Fig. 4b and Supplementary Table 4), partially overlapping with regions activated during pupil size upregulation (Up > Down; Supplementary Fig. 5f,g and Supplementary Table 3b display Down > Up results).

Taken together, pupil self-regulation is linked to an interplay of activation and deactivation in circumscribed brain regions interconnected with the LC, including prefrontal, parietal, thalamic and cerebellar areas.

### Pupil self-regulation modulates cardiovascular parameters

We tested the influence of self-regulating pupil size on cardiovascular parameters using electrocardiogram (ECG) signals during pupil-BF training (recorded in 15 participants, experiment 1B) and peripheral pulse data during subsequent fMRI sessions (experiment 2). We observed higher heart rates during Up than Down trials (Fig. 5a,c). This difference became more pronounced across pupil-BF training ('condition × session' interaction; $F_{(2.34, 30.37)} = 3.37$, $P = 0.04$, $\eta_p^2 = 0.21$, 95% CI$\eta_p^2$ (0.00, 0.39)) and remained stable during fMRI (main effect 'condition'; $F_{(1,22)} = 72.25$, $P < 0.001$, $\eta_p^2 = 0.77$, 95% CI$\eta_p^2$ (0.54, 0.85); Fig. 5a,c). Before training, pupil diameter and heart rate changes were largely unrelated ($P = 0.86$, Fig. 5e left). However, during fMRI, a higher pupil modulation index (Up−Down), thus better pupil self-regulation, was associated with larger differences in heart rate (Up−Down; rho = 0.63; $P = 0.002$, 95% CIrho (0.30, 0.82); sequential Bonferroni-corrected; Fig. 5e right).

To assess heart rate variability (HRV) measures suggested to represent a marker of parasympathetic activity[50,51], we calculated the root mean square of successive differences (RMSSD) and the percentage of successive cardiac interbeat intervals exceeding 35 ms (pNN35). There was no clear pupil self-regulation training effect (all $P ≥ 0.33$) on RMSSD (Fig. 5b,d left) and pNN35 (Supplementary Fig. 7a) and pupil diameter changes were not significantly related to RMSSD (Fig. 5f) and pNN35 (Supplementary Fig. 7b) changes before and following training during fMRI (all $P ≥ 0.19$). However, pNN35 values were generally higher during down- than during upregulation, both during training ($z = 2.84$; $P = 0.005$; $r = 0.76$) and subsequent fMRI sessions (whole-brain: $z = 3.06$; $P = 0.002$; $r = 0.62$; brainstem: $z = 3.371$; $P = 0.002$; $r = 0.69$; sequential

Bonferroni-corrected, Supplementary Fig. 7a). Similarly, during fMRI, RMSSD was descriptively higher during pupil down- than during upregulation, but effects did not reach significance (brainstem: $z = 1.71$, $P = 0.086$, $r = 0.35$; whole-brain: $z = 1.91$; $P = 0.056$, $r = 0.39$; Fig. 5b,d right). We further showed that (learning) effects on heart rate or HRV were weaker or absent when participants trained without BF (control group II; Supplementary Fig. 3d,e). However, correlations between heart rate and pupil modulation indices revealed similar links as in the pupil-BF group (Supplementary Fig. 3f).

Taken together, pupil self-regulation modulated cardiovascular parameters, particularly heart rate, consistent with the model of the role of LC in autonomic function.

### Experiment 3. Pupil self-regulation and the oddball task

To determine whether pupil self-regulation modulates behavioural and psychophysiological measures previously linked to LC-NA activity, we combined our pupil-BF approach with an auditory oddball task (Fig. 6a). Twenty-two participants who underwent pupil-BF training (experiments 1A and B) performed the task while (1) upregulating pupil size (1) downregulating pupil size or (3) executing a cognitive control task of silently counting backwards in steps of seven. The control task was included to control for cognitive effort effects that may arise from simultaneously executing pupil self-regulation and the oddball task.

Participants ($n = 20$ for final analyses) successfully self-regulated pupil size even when performing the oddball task simultaneously (robust repeated-measures analysis of variance (ANOVA), main effect 'condition': $F_{(1.52,16.67)} = 9.33$, $P = 0.003$, $\eta_p^2 = 0.46$, 95% CI$\eta_p^2$ (0.08, 0.65); Fig. 6b,c) as indicated by smaller (baseline-corrected) pupil sizes during Down than during Up and control trials. Also, absolute pupil size was increased for Up as compared with Down trials only during self-regulation but not during the baseline phase (Supplementary Figs. 2d and 8).

Unexpectedly, absolute pupil size was increased for control trials (Supplementary Figs. 2d and 8). Since this increase was observed during the baseline 'and' self-regulation phase, baseline-corrected pupil sizes were at intermediate levels compared with Up and Down conditions (Fig. 6b). This suggests a potentially higher cognitive load during counting than during pupil self-regulation throughout all phases of this control condition.

### Self-regulation modulates physiological LC activity markers

Previous research in monkeys suggests that elevated tonic LC activity constrains the intensity of phasic LC responses to salient stimuli[24–26]. Here we tested the prediction that phasic activity varies depending on tonic activity levels. We particularly investigated whether sustained pupil self-regulation is associated with changes in pupil dilation responses to target sounds during the oddball task, which has been linked to phasic LC activity. Consistent with previous research, we observed that pupil self-regulation led to differences in pupil dilation responses to target sounds ($z* = 7.29$; 714 to 3,000 ms, $P < 0.001$, Fig. 6d left), with significantly larger initial and prolonged elevated pupil dilation during Down than during Up trials ($z* = 4.07$; 718 to 2,669 ms, $P < 0.001$; 2,851 to 3,000 ms, $P = 0.005$, Bonferroni-corrected).

The cognitive control and Up conditions caused a similar initial pupil dilation but their time courses after peak dilation differed significantly ($z* = 4.12$; between 1,381 and 3,000 ms, all $P ≤ 0.02$, Bonferroni-corrected, Fig. 6d left). Differences between control and Down trials were not significant ($z* = 4.17$; no significant clusters, all $z ≤ 2.96$; all $P > 0.05$). In an exploratory analysis, we examined the single-trial relationship between baseline pupil size and pupil dilation. We found that smaller baseline pupil sizes before target onset were significantly related to larger pupil dilation responses towards target sounds ($\beta_{Baseline} - 1 = -0.20$, $t_{(19)} = -10.42$, $P < 0.001$; Cohen's $d = -2.33$, 95% CI$d$ (−3.18, −1.47); Fig. 7a; Supplementary Fig. 9a displays results within conditions).

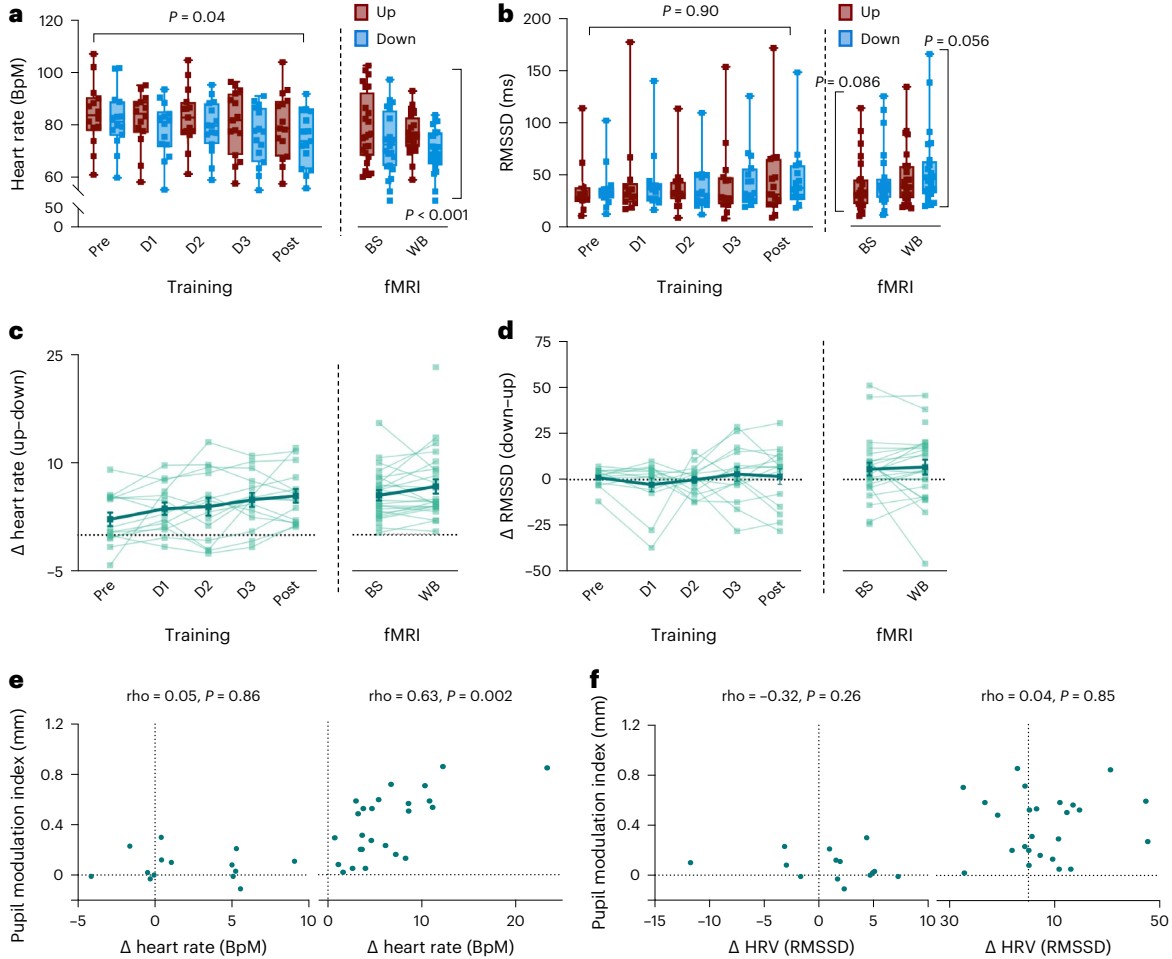

**Fig. 5 | Effects of pupil self-regulation on cardiovascular parameters. a,b,** Heart rate (**a**) and heart rate variability (HRV) (**b**) averaged for Up and Down trials across all participants for pupil-BF training (left; $n = 14$) and fMRI sessions (right; $n = 24$). HRV was estimated as the root mean square of successive differences (RMSSD). Self-regulation of pupil size systematically modulated heart rate with an increasingly larger difference between Up and Down trials over training sessions (repeated-measures ANOVA: 'condition × session' interaction: $F_{(2.34, 30.37)} = 3.37, P = 0.04, \eta_p^2 = 0.21$; Greenhouse–Geisser-corrected), which remained stable after training during fMRI (repeated-measures ANOVA; main effect 'condition'; $F_{(1,22)} = 72.25, P < 0.001, \eta_p^2 = 0.77$). Self-regulation did not significantly modulate HRV during training (robust ANOVA: $P = 0.90$). After training during fMRI, HRV was descriptively higher during Down than during Up, but statistical comparisons did not reach significance (two-tailed Wilcoxon signed-rank test; brainstem session: $P = 0.086$; whole-brain session: $P = 0.056$; not corrected for multiple comparisons). **c,d,** Individual differences in heart rate (Up–Down differences) and RMSSD (Down–Up differences) for training (left; $n = 14$) and fMRI sessions (right; $n = 24$). The thick solid line represents the group average, thin lines represent individual data. **e,** Spearman rho correlation coefficients (two-tailed, sequential Bonferroni-corrected) between pupil modulation indices (that is, the difference between pupil diameter changes in the two conditions, Up–Down) and differences in heart rate (Up–Down) revealing a significant link following (right; during fMRI) but not before pupil-BF training (left). **f,** Non-significant Spearman rho correlation coefficients (two-tailed) between pupil modulation indices (Up–Down) and RMSSD differences (Down–Up) before (left) and after pupil-BF training during fMRI (right). Boxplots indicate median (centre line), 25th and 75th percentiles (box), and maximum and minimum values (whiskers). Error bars in **c** and **d** indicate s.e.m. BS, brainstem fMRI; WB, whole-brain fMRI. For more detailed information on statistical parameters, see Supplementary Table 5.

Standard tones evoked only a minor pupil response (Fig. 6d right). Significant differences in responses between conditions were mainly driven by sustained elevation of pupil size in Up compared with Down trials ($z^* = 3.93, 1,081$ to $2,251$ ms, $P < 0.001$, Bonferroni-corrected) and a faster decrease in pupil size in Down compared with control trials ($z^* = 4.23$, between 916 and 1,233 ms, all $P \leq 0.02$, Bonferroni-corrected). However, these differences were observed 'after' peak dilation and probably reflect the effects of pupil self-regulation.

**Self-regulation modulates oddball task performance**

LC activity has been linked to the detection of task-relevant target stimuli[8,23,25,52]. Here we examine whether pupil self-regulation, which modulates LC activity, influences oddball task performance. Overall, accuracy was high (control: 94.3%, Up: 95.8%, Down: 97.9%). Analysing

reaction times to target sounds, we found that participants responded slowest during cognitive control and fastest during Down trials (Fig. 6e left; repeated-measures ANOVA main effect 'condition': $F_{(2,40)} = 35.97, P < 0.001, \eta_p^2 = 0.64, 95\%$ CI$\eta_p^2$ (0.43, 0.74)). Evaluating the relationship between single-trial pre-target baseline pupil size and reaction times to targets revealed a significant but only weak positive correlation, indicating that smaller pupil size at target onset may be associated with faster responses at single-trial level ($r_{rm} = 0.14, P < 0.001, 95\%$ CI$r_{rm}$ (0.11, 0.17); Fig. 7b left). Considering that both baseline pupil size and reaction times were influenced by a time-on-task effect (linear mixed-effects models on reaction times, estimate: 0.00039, $t = 7.97, P < 0.001$; on pupil size, estimate: $-0.00075, t = -6.95, P < 0.001$) with 'decreases' in pupil size and 'increases' in reaction time with time spent on task, we detrended both variables and repeated the analysis, revealing a slightly stronger

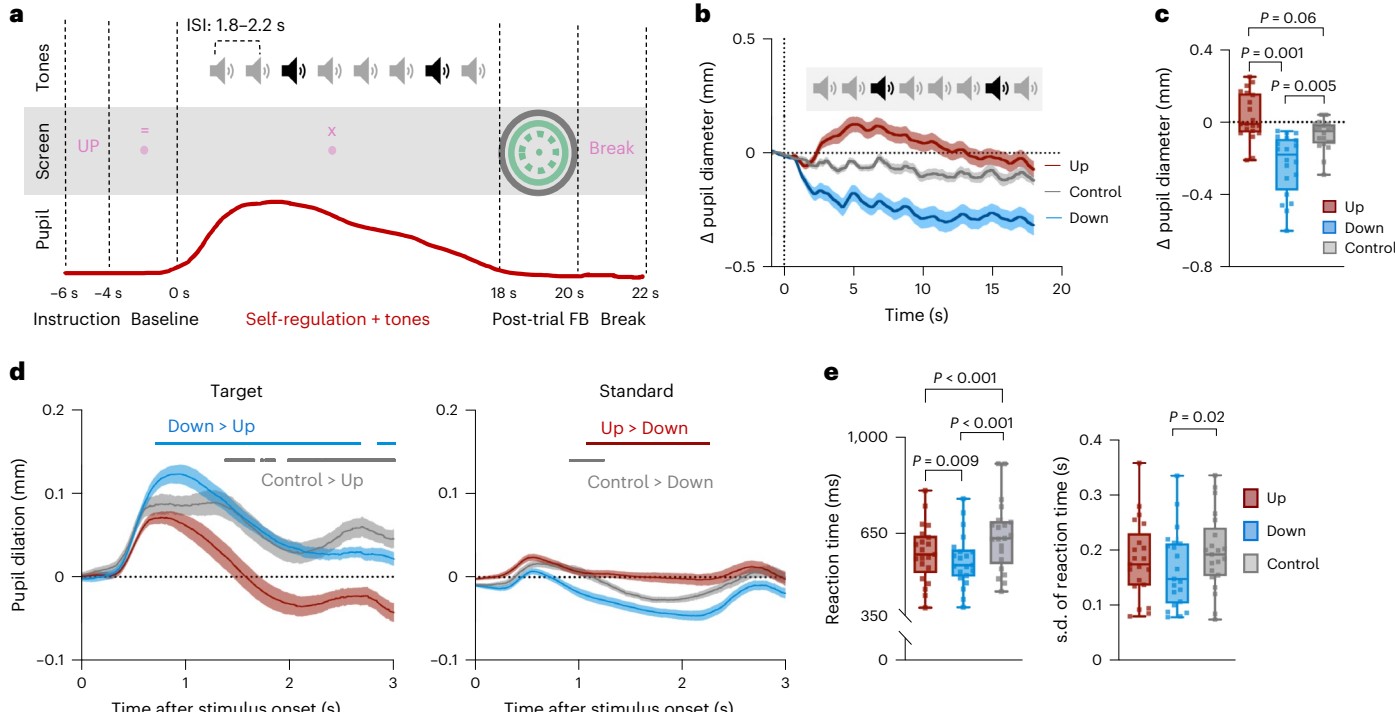

**Fig. 6 | Pupil self-regulation combined with an oddball task. a**, Schematic depiction of an example trial (Up) of experiment 3. Participants reacted to targets (black sound-icon) by button press and ignored standards (grey sound-icon) while simultaneously upregulating, downregulating pupil size or counting backwards in steps of seven (control). **b**, Pupil size changes averaged across participants for Up, Down and control trials showing 1 s of the baseline and the 18 s modulation phase. **c**, Pupil size changes from baseline during modulation averaged across the respective condition showing significantly lower values in Down than control and Up trials (robust repeated-measures ANOVA; $n = 20$; $F_{(1.52,16.67)} = 9.33$, $P = 0.003$, $\eta_p^2 = 0.46$; Down vs Up: $\hat{\psi} = -0.23$; $P = 0.001$; Down vs control: $\hat{\psi} = -0.14$; $P = 0.005$; for Up vs control: $\hat{\psi} = 0.10$; $P = 0.06$; two-tailed post-hoc tests; corrected for multiple comparisons using Hochberg's method). **d**, Baseline-corrected pupil dilation evoked by targets (left) and standards (right) for Up, Down and control trials. Solid lines indicate time windows of significantly

smaller responses to targets in Up than in Down and control trials (left) and significantly larger responses to standards in Up and control than in Down trials (right; two-tailed post-hoc tests of SPM1D repeated-measures ANOVA; largest $P = 0.017$; smallest $P = 0$; Bonferroni-corrected). **e**, Left: behavioural performance of 21 participants depicting faster responses to targets during Down than during Up trials (repeated-measures ANOVA: $F_{(2,40)} = 35.97$, $P < 0.001$, $\eta_p^2 = 0.64$, Down vs Up: $t_{(20)} = -2.87$, $P = 0.009$, $d = 0.63$) and control trials (Down vs control: $t_{(20)} = -7.19$, $P < 0.001$, $d = 1.57$; Up vs control: $t_{(20)} = -6.04$, $P < 0.001$, $d = 1.32$). Right: responses were also less variable in Down than in control trials ($t_{(20)} = -3.01$, $P = 0.02$, $d = 0.66$; post-hoc tests of repeated-measures ANOVA on reaction time and s.d. of reaction times were two-tailed and sequential Bonferroni-corrected). Squares in **c** and **e** represent individual data. Boxplots indicate median (centre), 25th and 75th percentiles (box), maximum and minimum values (whiskers). Shaded areas indicate s.e.m.

positive correlation ($r_{rm} = 0.16$, $P < 0.001$, 95% CI $r_{rm}$ (0.13, 0.20); Fig. 7b right; Supplementary Fig. 9b,c display correlations within conditions).

Finally, task performance variability measured as the s.d. of reaction times differed significantly between conditions (repeated-measures ANOVA main effect 'condition': $F_{(2,40)} = 4.84$, $P = 0.01$, $\eta_p^2 = 0.19$, 95% CI $\eta_p^2$ (0.01, 0.37); Fig. 6e right), mainly driven by less variable responses in Down than in control trials.

In summary, self-regulating pupil diameter as a proxy of (LC-mediated) arousal influences behavioural responses as predicted by current theories of noradrenergic function.

## Discussion

In a series of experiments, we showed that participants gain volitional control over their brain's arousal state via a pupil-BF approach based on the previously suggested mechanistic link between LC-NA activity, arousal and pupil size. We found that healthy adults can learn to self-regulate pupil size during a 3-day training. This ability was significantly reduced when receiving no veridical feedback (Fig. 2, and Supplementary Fig. 3). Investigating the neural, physiological and behavioural consequences of up- vs downregulated pupil size revealed three main findings: First, pupil self-regulation significantly modulates activity in arousal-regulating centres in the brainstem, including the LC and the SN/VTA (Fig. 3). Second, consistent with findings showing that

the LC exerts a strong influence on the cardiovascular system[18,32,53], we observed systematic changes in cardiovascular parameters, particularly in heart rate (Fig. 5). Third, pupil self-regulation significantly influenced task performance and a psychophysiological readout of phasic LC activity during an oddball task (Figs. 6 and 7).

Previous research demonstrated the feasibility of achieving volitional control over body and brain functions through bio- or neurofeedback combined with suitable mental strategies in an appropriate task setting[33,35,36,54]. We showed that feedback on pupil diameter significantly improved the ability to volitionally up- vs downregulate pupil size when compared (1) within participants from before to after training and (2) between veridical pupil-BF and control groups. Upregulation was already strong during the first biofeedback session, while downregulation gradually improved over training. This is in line with previous reports[28,30], where upregulating but not downregulating pupil diameter was successful when participants received one biofeedback-training session[30]. Further, once acquired and optimized, pupil self-regulation became a feedforward-controlled skill independent of constant feedback. We tested this ability in a no-feedback phase immediately after training; however, previous studies showed that self-regulation of central nervous system activity can last beyond the training period, indicating that this skill can be retained over time[33]. One concern when considering pupil measurements during real-time feedback relates

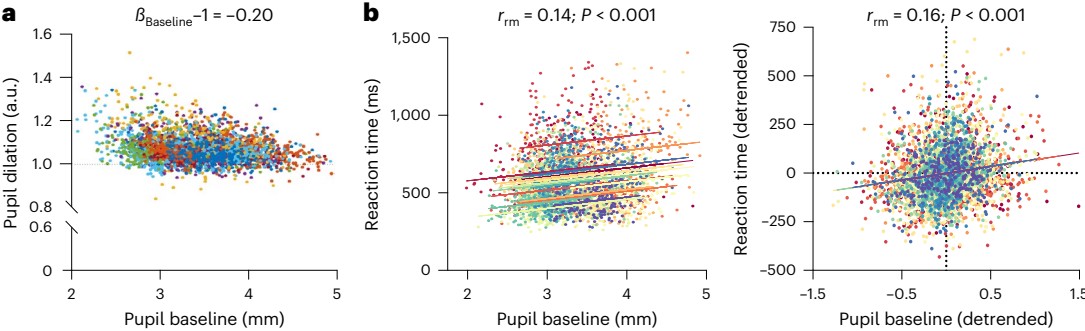

**Fig. 7 | Link between pupil data and behavioural performance during the auditory oddball task. a,b,** Single-trial analyses linking baseline pupil size (500 ms before target onset) with (**a**) relative pupil dilation responses ($\frac{\text{pupil dilation response peak}}{\text{baseline pupil size}}$; two-tailed one-sample $t$-test: $t_{(19)} = -10.42, P < .001$) and (**b**) reaction times towards targets. Left: two-tailed repeated-measures correlations for raw values: $r_{rm} = 0.14; P = 1.33 \times 10^{-15}$. Right: for detrended values: $r_{rm} = 0.16; P = 1.16 \times 10^{-20}$.

to screen and perceived colour luminance. Although we matched perceived colour luminance to the grey background of the screen, we cannot rule out inter-individual differences in perceptual luminance. Therefore, the number of coloured pixels on the screen was kept constant throughout the feedback phase. Furthermore, our replication experiment without visual interference and only post-trial feedback on the last training day confirmed stable pupil self-regulation even in the absence of online feedback. Another potential concern is the lack of double-blinding in experiment 1A, since the experimenter knew whether participants belonged to control or pupil-BF groups. However, throughout training, participants were physically isolated from the experimenter in a shielded room. Also, data preprocessing was conducted in an automated manner without knowledge of condition (and group) assignments.

Our hypothesis that volitional pupil size modulation is linked to activity changes in brain nuclei regulating the brain's arousal level including the LC was confirmed by our ROI analysis on brainstem fMRI data, showing that pupil size up- vs downregulation was indeed associated with systematic LC BOLD activity changes. Since neuromodulatory systems do not act in isolation, self-regulating pupil diameter did not exclusively modulate LC activity but led to analogous activity changes in the dopaminergic SN/VTA and less consistently in the cholinergic NBM. Interestingly, only activation differences in the LC but not in the SN/VTA or NBM significantly exceeded general activation changes across the whole brainstem (Supplementary Fig. 5c). Our results, especially regarding LC, were confirmed in additional ROI control analyses without the application of spatial smoothing which led to comparable result patterns. These findings are generally in line with a recent study in humans that linked noradrenergic, cholinergic and dopaminergic activity to pupil responses during a cognitive task[6]. The co-activation of the noradrenergic and dopaminergic system is not surprising since previous studies in non-human primates and rodents have identified noradrenergic LC projections to the VTA[55,56] and SN[55]. On the basis of our methodology, however, we cannot differentiate whether SN/VTA modulation during pupil self-regulation occurs directly through top–down control or indirectly via the LC.

In our study, links between up- vs downregulating pupil size and the NBM did not survive multiple comparison correction and were less consistent than for dopaminergic and noradrenergic regions. Previous comparisons of cholinergic and noradrenergic systems and their role in pupil dynamics in mice provided correlative evidence for the involvement of both systems. However, sustained cholinergic activity was observed mainly during longer-lasting pupil dilations, such as during locomotion, while moment-to-moment pupil fluctuations during rest closely tracked noradrenergic activity[7]. Furthermore, noradrenergic activity preceded cholinergic activity relative to peak pupil dilation[7],

suggesting together with the finding that noradrenergic neurons can depolarize cholinergic neurons[1], that noradrenergic activity may drive cholinergic activation. However, the temporal resolution of fMRI measures is insufficient to reliably test this proposal in our human dataset.

ROI analyses revealed that up- vs downregulating pupil diameter did not systematically modulate BOLD responses in the DRN and SC, brainstem regions implicated in non-luminance-dependent pupil size changes[5,41]. The absence of an effect in the SC aligns with the theory that it modulates the pupil mainly in the context of specific orienting responses towards salient events, while the LC modulates the pupil in the context of arousal[57,58].

Explorative general linear model (GLM) analyses on brainstem fMRI and its covariation with pupil size throughout the experiment revealed activation in the LC, SN/VTA, NBM and DRN but also in other critical nodes for producing a waking state and regulating autonomic activity, including the pedunculopontine nucleus and periaqueductal grey (Supplementary Table 2b). Even though GLM analyses contrasting up- vs downregulation trials did not survive cluster corrections, they revealed qualitatively similar activation patterns (including the predefined ROIs; Supplementary Fig. 5d and Supplementary Table 2a). This emphasizes that pupil self-regulation may modulate a distributed brainstem network associated with arousal regulation. However, it remains unclear whether these effects are directly driven by cortical and/or subcortical top–down control mechanisms or mediated through the LC as a major relay station. Future research could combine pupil-BF with pharmacological agents targeting different systems to unravel whether the LC is orchestrating the seemingly synchronized activity changes of different neuromodulatory systems and brainstem nuclei during pupil self-regulation.

Importantly, imaging the human brainstem, particularly the LC, is challenging due to the small-sized nuclei, high susceptibility to physiological noise and lower signal-to-noise ratio than cortical signals. We obtained consistent results across different smoothing levels and additionally applied stringent noise control through independent component analysis (ICA) and physiological noise modelling (PNM) which considers heart rate as a nuisance regressor. Given that up- vs downregulation was associated with significant differences in heart rate, the fMRI analyses revealed BOLD changes over and above this heart rate effect, indicating that our analysis revealed a conservative estimate of how pupil self-regulation modulates the activity of arousal-related brainstem nuclei.

Our study identified the ACC, OFC, DLPFC, precuneus, thalamus, globus pallidus and cerebellum as candidate areas that might exert top–down control of the arousal system in the brainstem. All these brain regions are heavily interconnected with the brainstem and particularly the LC. It is tempting to speculate that frontal areas such as the ACC and

OFC that have dense projections to the LC in non-human primates[4,59] form the brain's intrinsic control system of arousal and exert top–down control of LC activity. However, the nature of this activity, whether it is causal or consequential to arousal modulation, cannot be determined from our data.

Together, our fMRI results demonstrate that pupil size may provide an active information channel for self-regulating activity in areas involved in arousal regulation. Our results implicate the LC as one of the brainstem areas that are significantly modulated by pupil self-regulation, potentially influencing downstream areas involved in arousal control.

The oddball task has been closely linked to LC-NA activity in animal models and human research, and pupil dilation responses evoked by oddball stimuli have been considered a psychophysiological marker of phasic LC activity[4,60,61]. In addition, well-known theories derived from work in animal models postulate that phasic LC activity in response to task-relevant events depends on tonic LC activity: when tonic activity is upregulated, phasic responses are weak, whereas when tonic activity is relatively lower at an intermediate level, phasic responses are strong[4,26]. A similar relationship has been observed in human pupil measurements, reporting an inverse relationship between naturally fluctuating baseline pupil size and pupil dilation responses[60–62]. Consistently, we found that downregulating pupil diameter in a sustained way led to larger pupil dilation responses to target sounds. By contrast, upregulating pupil diameter led to smaller pupil dilation responses. These results are consistent with previous work, suggesting that self-regulating pupil diameter modulates tonic LC activity. One concern is that mechanisms specific to the structure of the eye or the pupil's musculature might have limited pupil responses if baseline pupil diameter is already high. However, this is unlikely as previous research has shown that varying pupil diameter through different luminance conditions did not affect task-evoked pupil dilation responses[60,63]. Accordingly, the more likely conclusion is that pupil dilation responses depend on the brain's arousal state as reflected in the baseline pupil size.

Our behavioural findings that task performance was better when baseline pupil diameter was low during downregulation than when it was high during upregulation or counting require careful interpretation. First, our design does not allow determination of whether pupil downregulation enhances behavioural performance compared with no dual task. This should be addressed in future studies implementing a resting control condition. Second, although our findings align with some results of previous studies analysing spontaneous[60] or experimentally induced pupil size fluctuations[64], there are studies reporting opposite effects in sustained attention tasks[65,66]. This inconsistency may be attributed to a suggested inverted-U relationship between arousal levels and task performance[4]: tasks that are naturally 'non-arousing' (for example, due to few external stimuli or low cognitive demands) might benefit from upregulation, while tasks with more arousing properties (for example, processing frequent external stimuli or dual-task conditions) might benefit from downregulation[67]. Overall, our data support the idea that attentional performance is influenced by baseline pupil size before target onset, reflecting arousal levels and potentially tonic LC activity. This influence may be achieved through modulation of cortical processes through arousal-regulating brainstem nuclei including the LC that facilitate adequate behavioural responses.

In our oddball task, we included a control condition where participants counted backwards in steps of seven while responding to target tones. Surprisingly, absolute pupil size was substantially higher in this condition already at baseline, suggesting increased cognitive effort throughout this task. Despite the differences in absolute pupil size, there was no significant difference in pupil dilation responses to target sounds between the control and downregulation conditions. Furthermore, participants exhibited the slowest and most variable responses to task-relevant sounds in control trials, possibly due to dual-task costs of counting and responding to target sounds. These costs may have

been reduced in self-regulation trials as participants had practiced this skill over several days, resulting in more automated processes. This interpretation is consistent with previous electroencephalography (EEG)-neurofeedback findings comparing task performance during veridical neurofeedback vs a control condition[67].

Since the LC plays a role in controlling autonomic activity through projections to cardiovascular regulatory structures[18,32,53], we investigated whether pupil self-regulation affects cardiovascular parameters. Consistent with our hypothesis, heart rate was generally higher during pupil up- than downregulation, an effect that increased across training sessions and correlated with the ability to self-regulate pupil size after pupil-BF training. However, in our second control group in which we also recorded ECG data throughout training, we saw a similarly strong link between pupil self-regulation and differences in heart rate at the end of training. Thus, whether this established link is due to feedback training or rather linked to the repeated exposure to explicit mental strategies needs to be clarified in future studies. Further, the effects on HRV were less clear. Whereas RMSSD did not significantly differ between pupil up- and downregulation, we observed significant effects for the pNN35 (Supplementary Fig. 7a). However, it is worth noting that our pupil self-regulation period was only 15 s, which is rather short for determining HRV changes. Extending pupil self-regulation duration will enable us to investigate whether this intervention can modulate HRV over longer time scales.

In summary, our study demonstrates that our pupil-BF approach enables healthy volunteers to volitionally control their pupil size. Self-regulation of pupil size is associated with systematic activity changes in brainstem nuclei that control the brain's arousal state, including the LC and the SN/VTA. Moreover, we observed that self-regulation of pupil size modulates (1) cardiovascular parameters and (2) psychophysiological and behavioural outcomes of an oddball task previously linked to LC activity. Our pupil-BF approach may constitute an innovative tool to experimentally modulate arousal-regulating centres in the brainstem including the LC. Considering the strong modulatory effects of such centres on cognitive function and various behaviours including stress-related responses, pupil-BF has enormous potential to be translated to behavioural and clinical applications across various domains.

## Methods

### General information

All experimental protocols were approved by the research ethics committee of the canton of Zurich (KEK-ZH 2018-01078) and were conducted in accordance with the declaration of Helsinki. All participants included in the study were healthy adults, free of medication acting on the central nervous system, with no neurological and psychiatric disorders and with normal or corrected-to-normal vision by contact lenses. All participants were asked to abstain from caffeine intake on the day of testing. Except for fMRI measurements, all studies were conducted in a noise-shielded room (Faraday cage) to allow the participants to focus on their task and to keep lighting settings constant at dim light. All participants provided written informed consent before study participation and received monetary compensation (that is, CHF20 per hour of participation). None of the experiments were preregistered.

### Experiment 1A. Pupil-BF to learn to self-regulate pupil size

**Participants.** A-priori power analyses based on our own pilot data of a single training session with 30 trials (7 participants; Supplementary Fig. 10) aiming for a power level of 80% resulted in a necessary sample size of 31 participants. As we expected training effects of our multisession approaches, we recruited 28 participants for the BF (24 ± 5 years old, 16 female) and 28 participants for our initial control group (24 ± 5 years old; 13 female) for experiment 1A, exceeding the sample size of other neurofeedback studies[33,68,69]. Participants were randomly assigned to the pupil-BF or initial control group. The experimenter was

aware of an individual's group assignment, but all participants received identical standardized instructions and performed the measurements while sitting alone in a shielded room without any interactions with the experimenter. The processing of the data was done in an automated and blinded way for all participants together, that is, without knowing to which group or condition the data belonged. For control group II (see below), we recruited additional 16 participants (25 ± 7 years old, 13 female). The data from control group II (which was collected at a later stage) was analysed separately using the same automated algorithm, which was again blind to the experimental condition. One participant of the pupil-BF group needed to be excluded from final data analyses due to the development of an eye blink-related strategy instead of a mental strategy. One participant of the initial control group dropped out after the first session due to personal reasons. This led to a final sample size of $n = 27$ for the pupil-BF group and $n = 27$ participants for control group I.

**Pupil-based biofeedback.** Participants sat alone in a shielded room in a comfortable chair with their chin placed in a chin rest to ensure a stable head position without putting too much strain on the neck of the participants. We kept the height of the chin rest constant across participants and adjusted the height of the chair to accommodate participants[70]. Their eyes were ~65 cm away from the eye tracker (Tobii TX300, Tobii Technology) that was positioned below the screen (240B7QPJ, resolution: 1,680 × 1,050 pixels; Philips) to allow for optimal eye tracking and measurement of pupil size. They were instructed to look at the fixation dot displayed in the centre of the screen. Pupil diameter and eye gaze data of both eyes were sampled at 60 Hz using the Tobii TX300 SDK for MATLAB v.3 and MATLAB 2013a. To ensure that participants did not use eye movement-related strategies (for example, vergence movements, squinting), we additionally videorecorded the right eye of the participants and visually inspected these videos. At the start of each session, the eye tracker was calibrated using a 5-point calibration.

In all three training sessions, participants received online (referred to as 'quasi real-time' due to slight delays in feedback, related to processing and averaging costs) and post-trial feedback on their pupil modulation performance which was based on estimating pupil size of the dominant eye. We accounted for artefacts caused by eye blinks, physiological and measurement-based noise with the following preprocessing steps: (1) rejection of data samples containing physiologically implausible pupil diameter values ranging outside of a pupil size of 1.5 and 9 mm; this step also ensured that blinks, recorded with a value of −1, would not be included in feedback shown to participants; (2) rejection of physiologically implausible pupil size changes larger than $0.0027$ mm s$^{-1}$. This previously implemented approach[30,71] is based on specifications of a study reporting peak velocity of the pupillary light reflex[72]. Finally, to ensure a smooth feedback display, the last two collected and processed pupil size samples were averaged and displayed[30]. During the pupil self-regulation phase, pupil size was displayed on the screen by means of a moving circle (Fig. 1c) centred around the fixation dot. This moving circle showed pupil size relative to a dashed circle representing the mean pupil size of a 5 s baseline phase (that is, when participants did not self-regulate pupil size). Importantly, the moving circle only indicated pupil size changes in the required direction, for example, getting larger for upregulation and smaller for downregulation trials. If pupil size did not change in the required direction, the circle stayed at the size of the baseline circle. To ensure constant screen luminance levels, the thickness of the circle was adjusted relative to its size so that the numbers of pixels shown on the screen were kept constant throughout the modulation phase. After completion of the modulation phase, post-trial performance feedback was displayed. Here, valid pupil diameter samples were averaged across the modulation phase, the maximum change was extracted and displayed on the screen, with average feedback being colour-coded: If participants successfully modulated pupil size into the required direction (that is, pupil size during modulation was larger than baseline pupil size in

upregulation trials or smaller than baseline pupil size in downregulation trials), the circle indicating the average change was shown in green (Fig. 1c). If pupil size modulation was not successful (that is, pupil size during modulation was similar to or smaller than the baseline pupil size in the upregulation trials or bigger than baseline pupil size in downregulation trials), the circle was depicted in red. The maximum change was always indicated in black. This post-trial performance feedback was displayed for 2 s.

Throughout the experiment, we ensured that all used colours were isoluminant to the grey background (RGB (150 150 150)) by calculating relative luminance as a linear combination of red, green and blue components on the basis of the formula: $Y = 0.2126 R + 0.7152 G + 0.0722 B$. It follows the idea that blue light contributes the least to perceived luminance, while green light contributes the most (https://www.w3.org/Graphics/Color/sRGB). Stimulus presentation throughout the experiment was controlled using the MATLAB-based presentation software Psychtoolbox 3.0.17.

All participants underwent three sessions of pupil-BF on separate days within a period of 7 d. The pupil-BF training session took place roughly at the same time of the day to keep circadian influences constant. Before pupil-BF training on day 1, participants read an instruction sheet explaining the procedures and providing recommended mental strategies derived from previous publications of dilated or constricted pupil size during different mental states and cognitive or emotional tasks[27,30,37–39]. Participants were instructed to rely on these (or similar) mental strategies. Furthermore, we determined the dominant eye of each participant (right eye dominance: $n = 22$ pupil-BF group; $n = 20$ control group I; $n = 13$ control group II) using the Miles test[73] since the displayed feedback during training was determined by the data recorded for the dominant eye.

In each of the training sessions, three up- and three downregulation blocks were performed, each consisting of 10 trials (30 Up/30 Down trials per day; Fig. 1b). Each trial (Fig. 1c) started with the display of the direction of modulation (either up- or downregulation) in green for 2 s on a grey background, followed by a baseline phase of 7 s. Participants saw a green fixation dot in the centre of a screen surrounded by a dashed green circle during baseline. During this baseline phase, participants were instructed to silently count backwards in their heads in steps of four to bring them into a controlled mental state. Then, a 15 s modulation phase, indicated by the display of an additional solid green circle, started in which participants were asked to use mental strategies to up- or downregulate their own pupil size while receiving online pupil-BF. This modulation phase was followed by post-trial performance feedback for 2 s. After a break of 5 s, indicated by the words 'short break' in green, a new trial started. After each block, participants could take a short self-determined break before they continued with the next block.

On training day 3, participants performed additional no-feedback trials following the same overall structure. However, the start of the modulation phase was indicated by a change from an '=' to an 'x' sign in the same colour (green) and same position above the fixation dot. No baseline circle was shown and no feedback, whether online or post-trial performance, was provided.

After the pupil-BF sessions, we conducted a debriefing in which participants reported in their own words which mental strategies they used for up- and downregulation.

**Training sessions of control groups.** Participants of the initial control group received the same instructions on mental strategies for up- and downregulations and the same amount of training as participants of the pupil-BF group. However, veridical feedback was not provided. To control for visual input on the screen, each participant of the control group was randomly matched to one participant of the pupil-BF group and saw the exact same feedback and received the same visual input as the veridical BF participant by providing a 'replay' of the feedback

screen of the pupil-BF group. This approach ensured the same visual input during the feedback phase and the same proportion of positive/negative feedback in all groups. Importantly, participants knew that the visual input was unrelated to their own performance (without the knowledge of a 'true feedback group') to prevent the development of illusory correlations and to exclude learning effects related to mental rehearsal[40]. As in the pupil-BF group, control participants were instructed to look at the fixation dot in the centre of the screen and to apply the mental strategies introduced to them (or similar).

This control group accounts for learning effects related to repeatedly using mental strategies as well as for visual input; however, it is possible that control participants may lack motivation and perception of success[40]. Therefore, we recruited additional 16 participants for a second control group (control group II), which received the same amount of training and instructions on mental strategies as the two other groups. Again, participants received 'yoked feedback', that is, they were randomly matched to one participant of the pupil-BF group and saw the exact same feedback and received the same visual input as the veridical BF participant by providing a 'replay' of the feedback screen. However, they were told to apply their mental strategies 'and' use the biofeedback for optimizing performance.

**Cardiovascular measurements.** For control group II, we additionally recorded cardiac data with a Biopac MP160 system and the accompanying AcqKnowledge software (Biopac Systems). ECG was recorded continuously and sampled at 1,000 Hz from two electrodes, one attached to the left lower rib and the other under the right collarbone. An electrode attached to the left collarbone was used as a reference. All physiological data were recorded and stored on a PC for offline analysis. We also recorded continuous respiration data by means of a breathing belt (Biopac Systems) that was affixed to the participant's chest; however, respiratory data are not reported here.

**Offline processing and analysis of pupil data.** (Pre-)processing of the pupil data was conducted using MATLAB R2018a (MathWorks). Recorded pupil size (in mm), gaze data and video recordings were visually inspected to ensure that participants followed the instructions to look at the fixation dot in the centre of the screen during the baseline and modulation phase of the experiment, and did not use eye movement-related strategies (for example, vergence movements or squinting). In case of violations during these phases (that is, squinting/opening eyes systematically, large eye movements/saccades, that is, deviations >-16° and -10° of visual angle on the x and y axes, respectively) that can potentially bias the validity of measured pupil size[70], trials were excluded from further analysis (pupil-BF group, 8.50%; control group I, 8.13%; Mann–Whitney $U = 341.50$, $z = -0.40$, $P = 0.69$; control group II, 9.52%; Mann–Whitney $U$ to compare with pupil-BF group: $U = 234.00$, $z = 0.45$, $P = 0.65$). Then, pupil data of both eyes were systematically preprocessed using the guidelines and standardized open-source pipeline[74]. Invalid pupil diameter samples representing dilation speed outliers and large deviation from trend line pupil size (repeated four times in a multipass approach) were removed using a median absolute deviation (MAD; multiplier in preprocessing set to 12 (ref. [74])), an outlier resilient data dispersion metric. Further, temporally isolated samples with a maximum width of 50 ms that border a gap larger than 40 ms were removed. Then, we generated mean pupil size time series from the two eyes, which were used for all further analyses reported (see ref. [74] on how mean times series can be calculated even with sparsely missing samples of one pupil). The data were resampled with interpolation to 1,000 Hz and smoothed using a zero-phase low-pass filter with a recommended cut-off frequency of 4 (ref. [74]). In a next step, the resulting data were inspected and trials with more than 30% of missing data points across baseline and modulation phases were excluded. Sessions with more than 50% of missing trials of a participant were excluded from further analysis ($n = 1$ in the pupil-BF group: participant who developed the

eye blink-related strategy). Finally, preprocessed pupil diameter was corrected relative to baseline by computing mean pupil size of the last 1,000 ms before the start of the modulation phase of each trial and subtracting this value from each sample of the respective modulation phase, as previously recommended[75].

In a next step, we averaged the baseline-corrected pupil diameter time series during baseline and modulation phases across the Up and Down condition of each session (days 1, 2, 3, no-feedback post) and each participant. Furthermore, for each participant and each session, absolute baseline diameter used for trial-based baseline correction (that is, averaged across 1,000 ms before modulation onset) was calculated for each condition (that is, Up and Down). To determine whether it is possible to learn to volitionally modulate pupil size and to determine whether veridical pupil-BF is essential for successful self-regulation, we calculated a pupil modulation index for all participants of both groups for each session (days 1, 2, 3, no-feedback post) as the average difference between baseline-corrected pupil values during up- vs downregulation across all $n = 15,000$ data points in the 15 s interval (that is, upsampled to 1,000 Hz)

$$\text{Pupil modulation index} = \frac{\sum_{t=1}^{15,000} \left(\text{pupilsize}_{\text{Up}}(t) - \text{pupilsize}_{\text{Down}}(t)\right)}{n} \quad (1)$$

Since successful upregulation is reflected in positive baseline-corrected pupil size values and successful downregulation in negative baseline-corrected pupil size values, larger pupil modulation indices indicate better condition-specific modulation.

Statistical analyses were performed using IBM SPSS 28, R v.4.1.2/4.2.2 (R Core Team) and JASP (v.0.16.2; https://jasp-stats.org/). Since Shapiro–Wilk tests revealed significant deviations from Gaussian distribution of some of the residuals of the pupil modulation index (Up–Down) for both groups, we computed robust mixed-design ANOVAs on the basis of 20% trimmed means using the R package WRS2 (v.1.1-3; https://cran.r-project.org/web/packages/WRS2/index.html) to compare the pupil modulation indices with the within-subjects factor 'session' (day 1 vs day 2 vs day 3 vs no-feedback post) and the between-subjects factor 'group' (pupil-BF vs control I). In case of significant effects, we derived post-hoc P values using bootstrap-based functions implemented in the WRS2 package. To test whether motivational factors or perceived success played an additional role, we repeated the same analysis with control group II (that is, factor session: day 1 vs day 2 vs day 3 vs no-feedback post; factor group: pupil-BF vs control II). Finally, we performed statistical analyses to ensure stable pupil size during baseline phases. To this end, we subjected absolute baseline pupil size averaged for each session and condition of participants of each group to a mixed-design ANOVA with the within-subjects factors 'session' (day 1 vs day 2 vs day 3 vs no-feedback post) and 'condition' (Up vs Down) and the between-subjects factor 'group' (pupil-BF vs control I). Sphericity was assessed using Mauchly's sphericity test and violations were accounted for with the Greenhouse–Geisser correction. In case of significant effects, we derived post-hoc P values corrected for multiple comparisons using sequential Bonferroni correction[76]. Due to violations from normal distribution of baseline pupil size in control group II, we conducted Wilcoxon signed-rank tests to compare baseline pupil size of up- and downregulation trials of each day. In addition, we ran a Bayesian ANOVA and Bayesian paired-samples t-test using the same factors to be able to evaluate whether there is evidence for the null hypothesis (that is, no difference in baseline pupil size between conditions in either session or group) and default priors.

**Cardiac data (pre-)processing and analyses.** ECG R peaks in the second control group were detected automatically and, if necessary, manually corrected using the MATLAB-based toolbox PhysioZoo[77]. Data segments consisting of non-detectable peaks or poor quality were excluded from further analyses. Resulting R-R intervals for which both

R peaks were occurring in the modulation phase of the pupil-BF training were extracted and further processed in MATLAB (R2021a). Unfortunately, some of the data were corrupted after data saving ($n = 1$ for all days, $n = 1$ for training day 2) or had incomplete trigger information due to technical issues with the trigger box ($n = 3$ for no-feedback trials at day 3) leading to $n = 15$ datasets on day 1 and day 3, $n = 14$ datasets on day 2 and $n = 12$ datasets for no-feedback trials on day 2. Heart rate, reflecting cardiovascular dynamics controlled by an interplay between the sympathetic and parasympathetic nervous system, was calculated by dividing 60 through the respective R-R intervals of the modulation phase. Furthermore, we computed the RMSSD on the basis of R-R intervals. We chose RMSSD because it is relatively free from breathing influences[78] and can be computed for intervals as short as 10 s[79,80]. It represents a beat-to-beat measure of HRV in the time domain that is assumed to give meaningful insights about parasympathetic activity of the autonomic nervous system[50,51]. Heart rate and HRV (RMSSD) calculated for each modulation phase were averaged across the respective up- and downregulation condition of each training session.

To investigate whether pupil self-regulation without veridical biofeedback systematically modulates cardiovascular parameters, we subjected heart rate averages for each condition (Up vs Down) and training session (days 1, 2, 3, no-feedback post) to a two-way repeated-measures ANOVA. Sphericity was assessed using Mauchly's sphericity test and violations were accounted for with the Greenhouse–Geisser correction. Since our HRV measure, RMSSD, significantly deviated from normal distribution (Shapiro–Wilk tests, all $P < 0.01$), we calculated difference scores in RMSSD between up- and downregulation trials (Down–Up) for each session (days 1, 2, 3, no-feedback post). Positive values indicated larger HRV in Down than Up trials and negative values indicated larger HRV in Up than Down trials. These difference scores were subjected to a non-parametric Friedman ANOVA. In cases where our analyses yielded significant effects, post-hoc tests were conducted. To determine whether potential differences in cardiovascular dynamics between Up and Down trials were associated with volitional pupil modulation performance at the beginning (that is, day 1) and at the end of training (that is, day 3), we calculated Spearman correlation coefficients between the pupil modulation index (that is, Up–Down) and heart rate (Up–Down). Reported statistical analyses were two-tailed tests and corrected for multiple comparisons using sequential Bonferroni correction[76].

**Experiment 1B. Replication of experiment 1A**
**Participants.** We recruited an independent cohort of 26 participants (18 female, 26 ± 7 years old). Technical issues with the eye tracker led to an interruption of training and exclusion of one participant ($n = 25$ for final analyses).

**Pupil-based biofeedback.** The paradigm used was identical to experiment 1A with two implemented changes: (1) we added a no-feedback session 'before' training in which participants only used instructed mental strategies without receiving any feedback; (2) online feedback on training day 3 was removed and participants only received post-trial performance feedback (Supplementary Fig. 4a). Similar to no-feedback trials, the baseline phase was indicated by an '=' sign above the fixation dot on the screen, changing to an 'x' as soon as the modulation phase started. Trial timing and instructions remained the same as in experiment 1A.

**Cardiovascular measurements.** For 15 participants (11 female, 25 ± 6 years old), we additionally recorded cardiac and respiratory data with the same system and settings as described in experiment 1A. Respiratory data are not reported here.

**Pupil data (pre-)processing and analysis.** The no-feedback session (before and after pupil-BF training) allowed us to directly compare self-modulation performance without feedback effects in the same group of participants. We excluded 5.4% of all trials due to violations during baseline and modulation phases (that is, squinting/opening of the eye in a systematic way, large eye movements/saccades). Baseline-corrected pupil size time series during self-regulation were calculated as described in experiment 1A. We statistically compared these time series before and after pupil-BF training by subjecting the data to a two-way repeated-measures ANOVA with the within-subjects factors 'condition' (Up vs Down) and 'session' (before and after pupil-BF training) using the MATLAB-based SPM1D toolbox for one-dimensional data (SPM1D M.0.4.8; https://spm1d.org/). SPM1D uses random field theory to make statistical inferences at the continuum level regarding sets of 1D measurements. It is based on the idea to quantify the probability that smooth random 1D continua would produce a test statistic continuum whose maximum exceeds a particular test statistic value and has been previously used to analyse 1D kinematic, biomechanical or force trajectories[81,82]. In cases of significant interaction effects, post-hoc tests implemented in the SPM1D software were used and results were corrected for multiple comparisons using Bonferroni correction. In addition, we tested whether before pupil-BF training, pupil modulation index time series were already significantly different from 0 via a one-sample $t$-test against 0 implemented in the SPM1D toolbox. A significant result would indicate that participants were able to self-regulate pupil size to a certain extent already before training.

**Cardiac data (pre-)processing and analyses.** ECG R peaks were detected, extracted and further processed as described in experiment 1A. Data of one participant were disrupted and excluded since it was not possible to detect R peaks for no-feedback trials before training as well as for the first training session. Heart rate and RMSSD based on R-R intervals were calculated. As an additional HRV measure, we further calculated the percentage of successive normal cardiac interbeat intervals greater than 35 ms (pNN35) on the basis of R-R intervals. The pNN35 is another common HRV measure and was used in other neurofeedback studies aimed at arousal regulation[67]. Heart rate and HRV (RMSSD and pNN35) calculated for each modulation phase were averaged across the respective up- and downregulation condition of each training session.

To investigate whether pupil self-regulation systematically modulates cardiovascular parameters, we subjected heart rate averages for each condition (Up vs Down) and training session (no-feedback pre, days 1, 2, 3, no-feedback post) to a two-way repeated-measures ANOVA. Sphericity was assessed using Mauchly's sphericity test and violations were accounted for with the Greenhouse–Geisser correction. Since our HRV measure, RMSSD, significantly deviated from normal distribution (Shapiro–Wilk tests, all $P < 0.01$), we calculated difference scores in RMSSD between up- and downregulation trials (Down–Up) for each session (no-feedback pre, days 1, 2, 3, no-feedback post). Positive values indicated larger HRV in Down than in Up trials and negative values indicated larger HRV in Up than in Down trials. These difference scores were subjected to a robust ANOVA using the WRS2 package in R. In cases where our analyses yielded significant effects, post-hoc tests were conducted. To determine whether potential differences in cardiac dynamics between Up and Down trials were associated with volitional pupil modulation performance before training, we calculated Spearman correlation coefficients between the pupil modulation index (that is, Up–Down) and heart rate (Up–Down) or RMSSD (Down–Up). Reported statistical analyses were two-tailed tests and were corrected for multiple comparisons using sequential Bonferroni correction[76] or Hochberg's approach implemented in the WRS2 package in R.

**Experiment 2. Pupil-BF combined with simultaneous fMRI**
**Participants.** For experiment 2, we re-recruited 25 participants (17 female; age 25 ± 5 years) from the pupil-BF groups of experiments 1A and B. No statistical methods were used to pre-determine sample sizes but our sample size is similar to or larger than those in previous fMRI

studies showing significant correlations between pupil measures and brain activity[6,8,9]. Additional exclusion criteria were: (1) metal implants and/or fragments in the body, (1) suffering from claustrophobia and (3) pregnancy at the time of the experiment. In case the last session of pupil-BF took place more than 10 days before the scheduled fMRI session, participants received one session of re-training combining 40 trials of online feedback (that is, 20 trials each for Up and Down) and 20 trials of only post-trial feedback (that is, 10 trials each for Up and Down) in the week before the first fMRI session. This re-training was included to ensure that participants were still able to modulate their own pupil size successfully. Re-training data are not reported here.

**Data acquisition.** All volunteers participated in two fMRI sessions, one using an optimized brainstem imaging sequence, the other acquiring whole-brain data (order of sessions was counterbalanced across volunteers). fMRI data were acquired on a 3 T Philips Ingenia system (software v.5.4.1) using a 32-channel head coil. Anatomical images of the brain used for anatomical co-registration were acquired using a T1-weighted sequence (MPRAGE; 160 sagittal slices, repetition time (TR)/echo time (TE) = 9.3/4.4 ms, voxel size = 0.7 mm$^3$, matrix size = 240 × 240, flip angle = 8°, FOV = 240 mm (AP) × 240 mm (RL) × 160 mm (FH)). A turbo spin echo neuromelanin-sensitive structural scan was acquired for delineation of the LC (not reported here), which was oriented perpendicular to the floor of the fourth ventricle (20 slices, 1.5 mm, no gaps, in-plane resolution: 0.7 × 0.88 mm (reconstructed at 0.35 × 0.35 mm), TR = 500 ms, TE = 10 ms, flip angle = 90°, covering the brainstem only). During the brainstem session, echo-planar imaging (EPI) scans were acquired in 39 sagittal slices (thickness: 1.8 mm, no gaps) aligned parallel to the floor of the fourth ventricle along the brainstem with the following parameters: TR = 2.5 s, TE = 26 ms, flip angle = 82°, SENSE acceleration factor = 2.1, in-plane resolution of 1.8 × 1.8 mm and 240 volumes per run. In addition, a whole-brain single-volume EPI image was acquired to improve co-registration between fMRI data and the anatomical MRI scans (identical sequence parameters to the brainstem fMRI sequence).

During the whole-brain session, EPI images were acquired in 40 slices (thickness of 2.7 mm) with the following parameters: TR = 2.5 s, TE = 30 ms, flip angle = 85°, FOV = 223 mm (AP) × 223 mm (RL) × 116 mm (FH), SENSE factor 2. Images were acquired at an in-plane resolution of 2.7 × 2.7 mm and were reconstructed at a resolution of 1.74 × 1.74 mm. In each session, we acquired 240 volumes per run. For the first seven participants, we used an EPI sequence with slightly different details (that is, especially higher spatial resolution to be able to see brainstem activity in small nuclei; non-isometric: 36 slices (thickness: 3 mm, no gaps, TR = 2.5 s, TE = 30 ms, flip angle = 82°, FOV = 210 mm (AP) × 210 mm (RL) × 108 mm (FH), SENSE factor 2.2, in-plane resolution of 2 × 2 mm reconstructed at a resolution of 1.64 × 1.64 mm)); however, due to the strong distortion observed in one excluded participant, we decided to adjust the sequence.

During both MRI sessions, cardiac cycle and respiratory activity were recorded at a sampling rate of 496 Hz with a peripheral pulse sensor attached to the left index finger and a chest belt, respectively (Philips software v.5.4.1; Invivo; Phillips Medical Systems). These data were used for physiological noise removal as well as for calculating cardiovascular parameters.

Concurrently with fMRI recordings, participants' eye gaze and pupil size of the left eye were tracked at 1,000 Hz using the EyeLink 1000 Plus Long Range Mount (SR Research), including the EyeLink 1000 plus software of SR Research and MATLAB 2019a. The eye tracker was placed at the end of the scanner bore on an aluminum stand and had a first-surface mirror attached to the head coil. At the beginning of each fMRI run, the eye tracker was calibrated using 5-point calibration implemented in the software of the device. Pupil diameter was converted from arbitrary units to mm using the following formula: diameter (mm) = diameter (a.u.)/1,372. The conversion factor was estimated as described in refs. 70,83.

**fMRI paradigm.** The pupil-BF paradigm during fMRI was similar to our training paradigm. Participants were instructed to maintain gaze on the green fixation dot presented in the centre of the screen. Each trial began with a baseline measurement (7.5 s), followed by an Up or Down phase (15 s; without online feedback) during which participants were asked to apply the acquired respective mental strategy. Participants only received post-trial performance feedback (2.5 s). After a short break (randomly jittered between 6–9 s), the next trial started. Online processing and calculation of the feedback displayed to participants was performed as in experiment 1A. The conditions were blocked within each of four fMRI runs, that is, four trials of upregulation followed by four trials of Down (or vice versa). At the beginning of each block, it was indicated on the screen whether this block contained Up or Down trials. After each block, a break of 10 s was added to allow participants to take short breaks in between. Each of the runs comprised two blocks of each condition, leading to 32 trials per condition. The order of conditions was counterbalanced across runs. Further, we counterbalanced across participants what conditions they started the respective fMRI session with.

**Processing and analysis of pupil data.** Pupil data (pre-)processing was conducted as described for experiment 1A, leading to baseline-corrected pupil diameter time series for Up and Down trials for each fMRI session. We statistically compared these time series for each session by subjecting the data to a paired-samples $t$-test (Up vs Down) using the SPM1D toolbox. Similar to experiment 1A, absolute baseline pupil diameter was compared using a repeated-measures ANOVA with the within-subjects factor 'session' (brainstem vs whole-brain) and 'condition' (Up vs Down), and Bayesian repeated-measures ANOVA using default priors.

Finally, preprocessed absolute pupil diameter data were downsampled throughout the whole experiment to match the fMRI volume acquisition by averaging all data within a given TR. Missing pupil diameters were linearly interpolated across adjacent epochs, resulting in a pupil diameter vector for each fMRI run.

**Processing and analysis of cardiac data.** Heart (pulse) rate concurrently measured during fMRI was (pre-)processed as described for experiment 1B, resulting in heart rate and HRV (RMSSD and pNN35) averages for each condition (Up and Down) and session (brainstem and whole-brain). Due to technical issues (whole-brain session of 1 participant), poor data quality (brainstem session of 1 participant), the final $n$ for heart rate and HRV analyses was 24 for each fMRI session. Heart rate data were subjected to a repeated-measures ANOVA with the within-subjects factor 'condition' (Up vs Down) and fMRI 'session' (brainstem vs whole-brain). Since HRV values did significantly deviate from normal distribution (revealed by Shapiro–Wilk tests $P < 0.05$), we compared Up and Down trials for each fMRI session using a Wilcoxon signed-rank test implemented in SPSS. Finally, as for the no-feedback session before pupil-BF training, we calculated Spearman correlation coefficients between the pupil modulation index (Up−Down) and differences in heart rate (Up−Down) or RMSSD and pNN35 (Down−Up), following training during fMRI (averaged across the two sessions). All statistical analyses were computed using SPSS and corrected for multiple comparison using sequential Bonferroni correction.

**MRI analysis.** MRI data were analysed using tools from FSL v.6.0.5.2 (http://fsl.fmrib. ox.ac.uk/fsl), MATLAB R2018a and FreeSurfer v.6.0 (https://surfer.nmr.mgh.harvard.edu/).

**Preprocessing of brainstem data.** Since brainstem fMRI signal is highly susceptible to corruption from physiological sources of noise, we used a specific preprocessing pipeline implementing suggestions from previously published research[6,14,84], including the following steps: brain extraction using FSL's automated brain extraction tool (BET[85]),

motion correction using the linear image registration tool (MCFLIRT[86]), spatial smoothing using a 3 mm full-width-at-half-maximum (FWHM) Gaussian kernel and a 90 s high-pass temporal filter as implemented via FSL's expert analysis tool (FEAT[87]). In addition, each functional run showing excessive motion with an absolute mean displacement greater than 1 mm (that is, ~half a voxel size) was excluded (3 runs for 1 participant, which led to exclusion from further analyses). To further de-noise the data, we performed ICA and FLS's PNM[88], an extended version of RETROICOR[89]. ICA denoising was performed using FSL's multivariate exploratory linear optimized decomposition interface (MELODIC[90]), which allows for decomposition of the fMRI data into spatial and temporal components. The resulting components were visually inspected and classified as 'noise' or 'signal' by two independent researchers using published guidelines[91]. Once labelled, the temporal information of the noise components was extracted and ICA noise regressor lists were generated for later implementation in our GLM. For PNM, cardiac and respiratory phases were assigned to each volume in the concatenated EPI image time series separately for each slice. Our complete physiological noise regression model included 34 regressors: 4th order harmonics to capture the cardiac cycle, 4th order harmonics to capture the respiratory cycle, 2nd order harmonics to capture the interaction between cardiac and respiratory cycles, 2nd order harmonics to capture the interaction between respiratory and cardiac cycles, one regressor to capture heart rate and one regressor to capture respiration volume. Subsequently, we visually inspected the waveforms to ensure that the peak of each respiratory and cardiac cycle was correctly marked and adjusted if necessary. The resulting voxelwise confound lists were later added to our GLM. Note that physiological data acquisition was corrupted for one participant. Therefore, these data were analysed without the PNM regressors. Image co-registration from functional to Montreal Neurological Institute (MNI) standard space was performed by (1) aligning the functional images of each run to each participant's whole-brain EPI image, using the FLIRT employing a mutual information cost function and six degrees of freedom[92]; (2) registering whole-brain EPIs to structural T1-weighted images, using a mutual information cost function and six degrees of freedom and then optimizing using boundary-based registration (BBR[93]); (3) co-registering structural images to the MNI152 (2 mm³) template via the nonlinear registration tool (FNIRT) using 12 degrees of freedom[94] and applying the resulting warp fields to the functional images. Each co-registration step was visually inspected using Freeview (FreeSurfer) to ensure exact alignment, with an emphasis placed on the pons regions surrounding the LC. Unfortunately, 3 out of 25 participants needed to be excluded from brainstem fMRI analyses due to heavy distortions (including brainstem regions; 1 participant), excessive motion (1 participant, see above) and periods of sleep during the measurement (1 participant). As additional control analyses to exclude smearing of noise from the 4th ventricle to the brainstem, we abstained from spatially smoothing the data during preprocessing, while all other (pre)processing steps remained the same (for results, see Supplementary Fig. 6).

**Preprocessing of whole-brain data.** We used a similar preprocessing pipeline as described above but with the following differences: Spatial smoothing was applied using a 5 mm FWHM Gaussian kernel. Functional EPI images were aligned to structural T1-weighted images using FLIRT. Structural images were aligned to the MNI standard space using FNIRT, and the resulting warp fields applied to the functional images. Data from 1 out of 25 participants showed very strong distortions in temporal and frontal regions and was excluded from further analysis. Also, we needed to exclude 1 functional run for 3 participants and 2 functional runs for 1 participant due to excessive motion with an absolute mean displacement greater than half a voxel size (that is, 1.4 mm). We did not apply PNM to the data and, as recommended by FSL (https://fsl.fmrib.ox.ac.uk/fsl/fslwiki/FEAT/UserGuide), ICA was

only performed (*n* = 9) if residual effects of motion were still left in the data even after using MCFLIRT for motion correction.

**fMRI analyses.** Time-series statistical analysis were performed using FEAT with the Oxford Centre for Functional MRI of the Brain's (FMRIB's) improved linear model (FILM). Two equivalent GLMs were estimated for each session (that is, for the whole-brain session and the brainstem session).

First, we directly contrasted brain activity during up- vs downregulation periods during whole-brain scans. Therefore, 'Up' and 'Down', were modelled as main regressors of interest for the complete modulation phase of 15 s. Instruction, baseline and post-trial performance feedback periods were modelled as separate regressors. All regressors were convolved with a double gamma haemodynamic response function (HRF) and its first temporal derivative. White matter (WM) and cerebrospinal fluid (CSF) time series, motion parameters and ICA noise component time series (if applied), respectively, were added as nuisance regressors. Second, we investigated which brain areas would change their activity in close association with pupil diameter. Therefore, pupil diameter was entered as a regressor of interest, together with the same nuisance regressors as described above. To adjust for delays between brain and pupillary responses, we shifted the pupillary signal by 1 s.

For the brainstem session, we defined GLMs with identical regressors that were, however, convolved with the default optimal basis sets (that is, the canonical HRF) and its temporal and dispersion derivatives using FMRIB's linear optimal basis sets toolkit[95]. This convolution was chosen to account for the possibility that brainstem BOLD responses such as in the LC are not well-modelled solely by the canonical HRF[96]. However, to avoid overestimating statistical effects, only the canonical HRF regression parameter estimates were used for subsequent higher-level analyses. In addition, the PNM voxelwise confound lists and the time series of ICA noise components lists (but not motion parameters) were used as nuisance regressors.

All described GLMs were computed separately for each run for each participant and then averaged across runs for each participant in second-level analyses using FMRIB's local analysis of fixed effects[97]. To determine significant effects at the group level, a third-level mixed-effects analyses was performed. Group *z*-statistic images were thresholded at *z* > 3.1 for whole-brain analyses and family-wise-error (FWE)-corrected at the cluster level. For the brainstem analysis, we report results thresholded at *z* > 2.3, FWE-corrected at the cluster level. To visualize brainstem activity, we created a mask covering the brainstem (derived from the Harvard–Oxford subcortical atlas; thresholded at 0.5 and binarized using FSL's image calculator fslmaths) and the NBM (Ch4 cell group derived from the JuBrain/SPM Anatomy Toolbox[98]), showing *z*-statistic images within these predefined regions.

**ROI analysis.** We conducted a series of ROI analyses to test our hypothesis that self-regulation of pupil size is linked to activity changes in the LC and other neuromodulatory, arousal-regulating centres in the brain. Our principal region of interest was (1) the LC; however, given the close interaction with other brainstem areas and neuromodulatory systems involved in pupil control, we defined secondary ROIs, namely (2) the cholinergic NBM, (2) the dopaminergic SN/VTA, (3) the serotonergic DRN and (4) the SC (a specialized motor nucleus in the midbrain implicated in the pupil orientation response) as a control region. Probabilistic anatomical atlases were used to define the location of the brainstem ROIs[43–49] and the NBM (Ch4 cell group[98]; see Fig. 3e) in MNI space. After careful visual inspection, we decided to threshold the SC and DRN mask at 0.1 and the SN/VTA-mask at 0.3. The masks for all ROIs were binarized. In a next step, we extracted the average mean signal intensity (*z*-values) within each ROI from the second-level fixed-effects analysis conducted for each participant for each GLM (that is, for Up vs rest and Down vs rest, and pupil diameter vector

used as regressor). Due to deviation from normal distribution for the NBM (Up > Rest; Shapiro–Wilk test; $P < 0.05$) and SC (pupil regressor; Shapiro–Wilk test; $P < 0.05$), we used Wilcoxon signed-rank tests to test for a significant difference between $z$-values for Up and Down periods within the NBM, as well as for a significant correlation between absolute pupil diameter and BOLD activity within the SC (that is, deviating from 0). For all other ROIs, we used paired-samples $t$-tests and one-sample $t$-tests against 0. All analyses were corrected for multiple comparisons using sequential Bonferroni correction. Even though we conducted preprocessing using PNM and ICA, we performed additional control and specificity analyses. To this end, we used additional control masks covering (1) the whole brainstem including the midbrain, pons and the medulla oblongata[99], and (2) the 4th ventricle (Supplementary Fig. 5a). First, as for our ROIs, we extracted the average mean signal intensity ($z$-values) within each ROI from the second-level fixed-effects analysis conducted for each participant for each GLM (that is, for Up vs rest and Down vs rest) and used two-tailed Wilcoxon signed-rank tests to test for a significant difference between $z$-values for Up and Down periods in these additional control regions. Further, all ROI analyses were repeated for data without spatial smoothing applied during preprocessing.

## Experiment 3. Pupil-BF combined with an oddball task

**Participants.** Experiment 3 involved 22 participants (15 females age $26 \pm 6$ years) of the 25 participants re-recruited for the fMRI experiment. Unfortunately, 3 participants dropped out for personal reasons. Participants received one session of re-training combining 40 trials of online feedback (that is, 20 trials each for Up and Down) and 20 trials of post-trial feedback (that is, 10 trials each for Up and Down) within 7 days before experiment 3, in case the last session of pupil-BF took place more than 10 days before the scheduled pupil-BF oddball session. This re-training was included to ensure that participants were able to modulate their own pupil size successfully. Re-training data are not reported here.

**Paradigm: pupil-BF with an integrated auditory oddball task.** The auditory oddball task during pupil-BF consisted of 186 target (~20%) and 750 standard (~80%) tones (440 and 660 Hz with the pitch for target being counterbalanced across participants) delivered for 100 ms via headphones at ~60 dB (Panasonic RP-HJE125, Panasonic) and embedded in our pupil-BF paradigm during which participants were volitionally up- or downregulating their own pupil size via acquired mental strategies. In addition to these two self-regulation conditions, we added a third control condition in which participants were asked to count back in steps of seven instead of self-regulating pupil size. To ensure that participants stick to this task, they were occasionally prompted to enter the final number (after every 6th counting trial). This condition was added to control for effort effects related to pupil self-regulation and to check whether oddball responses differ in self-regulated as compared with more natural fluctuations in pupil size. At the beginning of the task, participants were reminded to focus on their mental strategies and to pay attention to the tones at the same time.

Each pupil-BF oddball trial began with a short instruction (2 s) indicating the required pupil self-regulation direction (that is, Up or Down) or the counting control condition, followed by a baseline measurement of 4 s in which participants were asked to silently count backwards in steps of seven. Then, participants were asked to apply the acquired mental strategy to up- or downregulate pupil size for 18 s or to continue counting (that is, control). During this self-regulation or counting phase, eight tones (one or two targets and seven or six standards) were played, and participants were asked to respond to targets but not to standards by pressing a key as quickly as possible with the index finger of their preferred hand. The first tone which was always a standard in each self-regulation or counting phase was played after the first jitter phase (1.8–2.2 s after the baseline

phase) to ensure that participants already reached some modulation of pupil size before the first target appeared. The inter-stimulus interval (ISI) between tones was randomly jittered between 1.8–2.2 s. Between two targets, there were at least two standards presented, leading to a minimum ISI of 5.4 s between two targets, allowing the target-evoked pupil dilation response to return to baseline between targets. After each trial, participants received post-trial feedback on average and maximum pupil diameter change for 2 s. After a short break of 2 s, the next trial started. The calculation of feedback was similar to that for experiment 1A, except that only the last 2 s of the baseline phase were considered due to the shortened baseline phase of 4 s. For counting trials without feedback phases, counting was followed by a 4 s break to keep trial timings similar. Instructions and indication of task phases were displayed on the screen as in experiment 1B on day 3. However, to improve visibility further, indication of task phase changes from baseline measurements to self-regulation was displayed in magenta (instead of green) isoluminant to the grey background (Fig. 6a).

The pupil-BF conditions were blocked, each block containing 13 trials of Up, Down and control. The number and order of tones were the same for each pupil-BF condition. After each block, participants could take a self-paced break. In total, the task consisted of 9 blocks leading to 117 pupil-BF trials in total. The order of blocks was pseudorandomized in the sense that each condition needed to be presented once (or twice) before another condition was presented for the 2nd (or 3rd) time.

**Physiological data acquisition.** Eye gaze, pupil diameter, cardiac and respiratory data were recorded throughout the task as described in experiments 1A and B. In addition, we acquired EEG data using 64 Ag/AgCl actiCAP active surface electrodes, an actiCHamp Plus amplifier and the Brain Vision Recorder proprietary software (Brain Products). However, cardiac, respiratory and EEG data are not reported here.

**Data (pre-)processing and statistical analyses.** *Pupil-BF: pupil self-regulation under dual task conditions.* Pupil data (pre)processing was conducted as described for experiment 1A. We had to exclude 2 participants from further analyses, leading to a final $n = 20$ for pupil data analyses: one participant showed a very high blink rate during the oddball, with more than 30% of missing samples during baseline and self-regulation or counting phases for more than 50% of the trials. A second participant showed a similar pattern with more than 30% of missing samples in 35% of the trials, mainly in the control condition (only 20% of the trials were usable after preprocessing). Combined with low behavioural performance (correct responses to targets deviated more than 3 s.d. from the group mean), we decided to exclude this participant from pupil and behavioural data analyses. For each of the remaining participants, we derived 'absolute' and 'baseline-corrected' pupil diameter time series averaged for Up, Down or counting trials throughout the baseline and self-regulation or counting phases of the pupil-BF oddball task. Both absolute and baseline-corrected pupil diameter time series were extracted since we expected to observe target-evoked pupil dilation responses that depend in size on absolute pupil size before tone onset. Baseline correction was applied as described above, choosing a baseline of 1 s before self-regulation and counting.

For statistical analyses of self-regulated pupil size, we averaged preprocessed, baseline-corrected pupil diameter time series for each condition across the entire self-regulation or counting phase to test whether participants were able to successfully self-regulate pupil size even under challenging dual-task conditions. Since these data were deviating from normal distribution (Shapiro–Wilk test; $P < 0.05$), we subjected the data to a robust repeated-measures ANOVA implementing 20% trimmed means with the factor 'condition' (Up vs Down vs control; R package WRS2). In case of a significant main effect, we computed the corresponding post-hoc tests included in this R package

with implemented FWE correction. Next, we averaged absolute pupil diameter samples across the last second of the baseline phase and subjected it to a repeated-measures ANOVA with the within-subjects factor 'condition' (Up vs Down vs control) to compare absolute baseline pupil diameter among the three conditions. Sphericity was assessed using Mauchly's sphericity test and violations were accounted for with the Greenhouse–Geisser correction. In cases of a significant effect, post-hoc tests were conducted using the sequential Bonferroni correction for multiple comparisons. In cases of no significant effect, we ran an analogous Bayesian repeated-measures ANOVA using JASP (with default priors) to evaluate whether there is evidence for the null hypothesis (that is, no baseline differences between conditions). All reported statistical analyses were two-tailed tests.

*Pre-tone pupil size and pupil dilation responses to tones.* We extracted 3.5 s epochs around each standard or target tone from −0.5 to +3 s relative to tone onset. Pre-tone pupil diameter was calculated by averaging 500 ms of pupil diameter data preceding tone presentation. The pupil time series was then (1) normalized to this pre-tone pupil diameter baseline and (2) averaged for each stimulus type (standard vs target) and condition (Up vs Down vs control).

To compare 'target'-evoked pupil dilation responses between conditions, we subjected the baseline-corrected pupil dilation response time series to targets grand-averaged within each condition to a repeated-measures ANOVA with the within-subjects factor 'condition' (Up vs Down vs control) using the SPM1D toolbox. This time series data analysis allowed us to compare both peak dilation towards targets and the shape of the dilation response statistically. The same analysis was repeated for pupil dilation responses to standard tones as a control analysis. In cases of significant main effects, post-hoc tests implemented in the SPM1D software corrected for multiple comparisons using Bonferroni correction were used.

*Behavioural data.* Throughout the experiment, we recorded the correctness of responses and reaction times to tones. Responses to standard tones (that is, false alarms) and responses for which reaction times exceeded 2 s.d. below or above the mean of the respective participant were excluded from further reaction-time analyses. Lastly, we calculated the percentage of hits (that is, responses to targets) and false alarms (that is, responses to standards), and obtained a percentage of correct responses (that is, accuracy) by subtracting these two measures from each other[64].

To investigate whether self-regulation of pupil size modulates behavioural task performance on the oddball task, we subjected individual reaction times to target tones averaged across trials of each condition to a repeated-measures ANOVA with the within-subjects factor 'condition' (Up vs Down vs control). Sphericity was assessed using Mauchly's sphericity test and violations were accounted for with the Greenhouse–Geisser correction. Next, we examined whether not only speed per se but also performance variability indicated by the s.d. of reaction times may be modulated by self-regulated pupil size. We subjected the s.d. of reaction times to a repeated-measures ANOVA with the within-subjects factor 'condition' (Up vs Down vs control). Sphericity was assessed using Mauchly's sphericity test and violations were accounted for with the Greenhouse–Geisser correction. In cases of a significant effect, we conducted post-hoc tests controlled for multiple comparisons using sequential Bonferroni corrections. Reported statistical analyses were two-tailed tests.

*Link between pupil and behavioural data.* In a next step, we computed repeated-measures correlation coefficients between the derived single-trial pre-tone absolute baseline pupil diameter (averaged 500 ms before target onset) and baseline-corrected target-evoked pupil dilation response peaks of all participants using the R package rmcorr (v.0.4.5, 0.6.0)[100]. Since the correlation between a variable $x$ (that is, baseline pupil diameter) and the variable $y−x$ (that is, baseline-corrected pupil dilation response peaks) is susceptible to a regression towards the mean effect, we decided to use relative instead

of subtractive baseline correction for these additional analyses (that is, $\frac{\text{pupil dilation response peak}}{\text{baseline pupil diameter}}$). Even though correlating a relative change from a variable to this variable itself is also not free of bias, it is indeed possible to test for an unbiased relationship by fitting the following model in R for $y$ = Pupil dilation response peak and $x$ = Baseline pupil size, and consider the resulting coefficient

$$\beta_x : lm\left(\log\left(y\right) \sim \log\left(x\right)\right). \qquad (2)$$

This model corresponds to the following equation:

$$\log\left(y\right) = \beta_x \log(x) + \text{intercept} + \epsilon \qquad (3)$$

which is equivalent to:

$$\frac{y}{x} = x^{\beta_x-1} \times e^{\text{intercept}} \times e^{\epsilon}, \qquad (4)$$

Importantly, this is an unbiased estimate of an influence of the baseline pupil size $x$ on the following pupil dilation response peak $y/x$ (that is, randomly generated estimates of $\beta$ by simulations centred around 0). Then, we tested with a one-sample $t$-test the $H_0$: $\beta_x − 1 = 0$, that is, that baseline pupil size has no significant influence on pupil dilation responses.

Finally, to investigate the relationship between pupil size measures and behavioural responses at a single-trial level, we conducted additional repeated-measures correlation between reaction times and baseline pupil size and pupil dilation responses. Since both baseline pupil diameter and reaction times to targets could potentially be influenced by previously reported time-on-task effects[61,101], we ran linear mixed models at single-trial level similar to ref. 102, with the dependent variables reaction times and baseline pupil size, respectively. Time-on-task entered the model as a fixed effect and was operationalized as the number of the respective trial in the experiment in ascending order. A random intercept was modelled for each participant (that is, reaction time ~ time on task + (1|participant) and baseline ~ time on task + (1|participant)). In cases of significant time-on-task effects, we removed a linear trend in the data for each participant using the 'detrend' function in MATLAB (linear method) before subjecting it to repeated-measures correlations in R.

## Reporting summary

Further information on research design is available in the Nature Portfolio Reporting Summary linked to this article.

## Data availability

Processed pupil, fMRI, heart rate, heart rate variability and behavioural data are openly available on the ETH Library Research Collection (https://doi.org/10.3929/ethz-b-000630621). For fMRI regions of interest analyses (ROI), we used probabilistic anatomical/cytoarchitectonic atlases to define the locations of the ROIs (Brainstem Navigator; https://www.nitrc.org/projects/brainstemnav/; https://www.fz-juelich.de/en/inm/inm-7/resources/jubrain-anatomy-toolbox) and other control regions including the complete brainstem (https://www.fil.ion.ucl.ac.uk/spm/toolbox/TPM/).

## Code availability

Unfortunately, scripts used for execution of the pupil-based biofeedback method cannot be made publicly available. The exact code of the pupil-based biofeedback algorithm is proprietary software of ETH Zurich and cannot be shared beyond the description of the algorithm given in the methods section. However, researchers interested in verifying and reproducing our results can do so on location in a secured environment at the Neural Control of Movement Laboratory, ETH Zurich, upon signing a confidentiality agreement.

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

## Acknowledgements

We thank all participants of the study; J. W. de Gee for great help regarding brainstem-specific MRI sequences; V. Zerbi, J. Bohacek, J. M. Shine and O. Harrison for fruitful discussions; L. Graz for his statistical advice; W. Potok and D. Woolley for valuable help and feedback on the manuscript; and R. Lüchinger for the great MRI support. S.N.M. discloses support for the research of this work from the Swiss National Science Foundation (SNSF) Spark grant (CRSK-1_190836) and was supported by the Hochschulmedizin Zürich Flagship project STRESS. J.I. was supported by SNSF grant 32003B_207719. S.K. was supported by the ETH Zurich Postdoctoral Fellowship Program. N.W. was supported by the National Research Foundation, Prime Minister's Office, Singapore, under its Campus for Research Excellence and Technological Enterprise (CREATE) programme and was funded by the SNSF and Innosuisse BRIDGE Discovery grant (40B2-0_203606). The funders had no role in study design, data collection and analysis, decision to publish or preparation of the manuscript.

## Author contributions

S.N.M., M.B. and N.W. were involved in conceptualization and design of the study. S.N.M., J.I. and S.M. acquired the data. S.N.M., S.K. and N.W. planned the analysis. S.N.M., J.I., S.M. and M.C.D. analysed the data. S.N.M. and N.W. interpreted the data. S.N.M. drafted the manuscript and M.B., S.K., J.I., S.M., M.C.D. and N.W. substantively revised it.

## Funding

## Competing interests

S.N.M., M.B. and N.W. are founders and shareholders of an ETH spin-off called 'MindMetrix' that aims to commercialize pupil-based biofeedback, and have a patent application related to the method of pupil-based biofeedback (patent applicant: ETH Zurich; inventors: M.B., S.N.M., N.W., pending patent applications EP21704565.7 and US17/800,455). All other authors declare no competing interests.

## Additional information

**Correspondence and requests for materials** should be addressed to Sarah Nadine Meissner or Nicole Wenderoth.

# Reporting Summary

## Statistics

For all statistical analyses, confirm that the following items are present in the figure legend, table legend, main text, or Methods section.

| n/a | Confirmed | |
|---|---|---|
| ☐ | ☒ | The exact sample size (*n*) for each experimental group/condition, given as a discrete number and unit of measurement |
| ☐ | ☒ | A statement on whether measurements were taken from distinct samples or whether the same sample was measured repeatedly |
| ☐ | ☒ | The statistical test(s) used AND whether they are one- or two-sided *Only common tests should be described solely by name; describe more complex techniques in the Methods section.* |
| ☒ | ☐ | A description of all covariates tested |
| ☐ | ☒ | A description of any assumptions or corrections, such as tests of normality and adjustment for multiple comparisons |
| ☐ | ☒ | A full description of the statistical parameters including central tendency (e.g. means) or other basic estimates (e.g. regression coefficient) AND variation (e.g. standard deviation) or associated estimates of uncertainty (e.g. confidence intervals) |
| ☐ | ☒ | For null hypothesis testing, the test statistic (e.g. *F*, *t*, *r*) with confidence intervals, effect sizes, degrees of freedom and *P* value noted *Give P values as exact values whenever suitable.* |
| ☐ | ☒ | For Bayesian analysis, information on the choice of priors and Markov chain Monte Carlo settings |
| ☒ | ☐ | For hierarchical and complex designs, identification of the appropriate level for tests and full reporting of outcomes |
| ☐ | ☒ | Estimates of effect sizes (e.g. Cohen's *d*, Pearson's *r*), indicating how they were calculated |

*Our web collection on statistics for biologists contains articles on many of the points above.*

## Software and code

Policy information about availability of computer code

| Data collection | For the collection of pupil data during Experiments 1A, 1B and 3, we used MATLAB (version 2013a) and the Tobii_TX300 SDK for MATLAB version 3; for response recording of behavioral data, we used the MATLAB-based presentation software Psychtoolbox 3.0.17. For the collection of (f)MRI, pulse and respiratory data, we used the Philips software version 5.4.1 (Philips 3T Ingenia; Philips; Philips Medical Systems; Invivo); for recording of pupil data during fMRI (Experiment 2), we used Matlab (2019a) and the Eyelink 1000 plus software of SR Research; for the recording of cardiovascular and respiratory data (in Experiment 1B), we used the Biopac MP 160 system with the accompanying AcqKnowledge software version 5.0 |
|---|---|
| Data analysis | For (pre-)processing of pupil data, open-source software from Kret & Sjak-Shie (2018) and Matlab code (2018a) (e.g., for baseline correction) was used. For (pre-)processing of cardiovascular data, we used the Matlab-based toolbox physiozoo (Matlab version 2021a) and Matlab code (Matlab version 2018a). For (f)MRI data analyses, we used FSL version 6.0.5.2 (including the implemented analyses tools BET, MCFLIRT, FEAT, FNIRT, FLIRT, FILM, FLOBS, MELODIC, and PNM), FreeSurfer version 6.0. For all statistical analyses, we used IBM SPSS version 28; R version 4.1.2/4.2.2 including the packages WRS2 (version 1.1-3) and rmcorr (version 0.4.5, 0.6.0), JASP version 0.16.2, and the Matlab-based SPM-1D toolbox (M.0.4.8). |

For manuscripts utilizing custom algorithms or software that are central to the research but not yet described in published literature, software must be made available to editors and reviewers. We strongly encourage code deposition in a community repository (e.g. GitHub). See the Nature Portfolio guidelines for submitting code & software for further information.

## Data

Policy information about availability of data

All manuscripts must include a data availability statement. This statement should provide the following information, where applicable:
- Accession codes, unique identifiers, or web links for publicly available datasets
- A description of any restrictions on data availability
- For clinical datasets or third party data, please ensure that the statement adheres to our policy

Processed data are openly available on the ETH Library Research Collection: https://doi.org/10.3929/ethz-b-000630621
For fMRI regions of interest analyses (ROI), we used probabilistic anatomical/cytoarchitectonic atlases to define the locations of the ROIs (Brainstem Navigator; https://www.nitrc.org/projects/brainstemnavig/; https://www.fz-juelich.de/en/inm/inm-7/resources/jubrain-anatomy-toolbox) as well as the complete brainstem (control analyses; https://www.fil.ion.ucl.ac.uk/spm/toolbox/TPM/).

## Human research participants

Policy information about studies involving human research participants and Sex and Gender in Research.

| | |
|---|---|
| Reporting on sex and gender | The reported results apply to the female and male sex: our sample included male and female participants. This information was determined based on self-reports of the participants. We did not conduct sex-/gender-based analyses in addition to our group level analyses since our study included both sexes and was not powered to conduct separate analyses |
| Population characteristics | See behavioral and social sciences study design information |
| Recruitment | Healthy participants of the present study were recruited via online advertisement on University web pages. We are not aware of any self-selection bias, however, one bias that may impact the results is that the majority of the participants were young healthy university students in Switzerland. All participants gave written informed consent. |
| Ethics oversight | All experimental protocols were approved by the research ethics committee of the canton of Zurich (KEK-ZH 2018-01078) |

Note that full information on the approval of the study protocol must also be provided in the manuscript.

# Field-specific reporting

Please select the one below that is the best fit for your research. If you are not sure, read the appropriate sections before making your selection.

☐ Life sciences     ☒ Behavioural & social sciences     ☐ Ecological, evolutionary & environmental sciences

For a reference copy of the document with all sections, see nature.com/documents/nr-reporting-summary-flat.pdf

# Behavioural & social sciences study design

All studies must disclose on these points even when the disclosure is negative.

| | |
|---|---|
| Study description | The present study is a quantitative experimental study |
| Research sample | Participants were mainly healthy university students (undergraduate, graduate students) from different Universities in Zürich/Switzerland. Since it's mainly including young adultss the sample is not representative considering the general population. In the pupil-NF group of Experiment 1A, participants were 24 +/- 5 years old (mean +/- SD), 16 female and 12 male; in control group I, participants had a mean age (+/- SD) of 24 +/- 5 years, 13 were female and 15 male. In control group II, participants were were 25 +/- 7 years old (mean +/- SD), 13 were female and 3 were male. For Experiment 1B, participants had a mean age (+/- SD) of 26 +/- 7 years, 18 were female and 8 male. Experiments 2 and 3 included a subset of the participants of Experiments 1A and 1B. Since the present study is to our knowledge the first proof-of-concept study of pupil-based biofeedback combined with (brainstem and whole-brain) fMRI consisting of multiple (training) sessions, we decided to recruit an accessible sample of mainly university students. |
| Sampling strategy | Participants were randomly assigned to a pupil-BF and control group I in Experiment 1A. Control group II was recruited after acquiring data of the pupil-BF and control group I. For our pupil-based biofeedback paradigm, we did a power calculation (with a power level of 80%) based on pilot data of a single pupil-based BF session. Our final sample size of Experiment 1A including 28 participants for each group (Biofeedback and Control group I) furthermore exceeds the sample size of many other biofeedback studies. Since we were the first ones to combine pupil-based biofeedback with fMRI and an oddball task, we based our sample size estimations for Experiment 2 and 3 on previous fMRI studies showing a link between pupil size and brain (LC) activity and the oddball task, respectively. |

| | |
|---|---|
| Data collection | We used eye trackers (Tobii TX300 in Experiments 1 and 3; Eyelink 1000 Plus Experiment 2) for the collection of pupil data, an MRI scanner (Philips Ingenia, 3T) for fMRI data collection; ECG, a peripheral pulse sensor and a respiratory belt (Biopac MP 160 system for Experiment 1A and B; Invivo Philips Medical Systems for Experiment 2) to collect heart rate and respiratory data. Behavioral data was acquired and stored offline on a PC. Demographic data and questionnaire data (e.g., for exclusion criteria) were acquired using pen and pencil. During data acquisition, no-one except for researchers and participants were present. Furthermore, during all experiments, participants were either in a Faraday cage or MRI scanner with the experimenter being in the control room, thus participants could focus on themselves without the feeling of someone watching/influencing them during the experiments. Participants were blind regarding the experimental group and the hypotheses of the study; the researchers analyzing the data were not blind to the experimental groups during data collection but blind to the experimental condition/groups during preprocessing of all data. |
| Timing | Data for experiment 1A was collected from October 2019 to February 2020 (pupil-BF and control group I) and from April to June 2023 (control group II); Data for experiment 1B, 2 and 3 was collected between July 2020 and October 2021. This phase was prolonged due to the COVID-19 pandemic and respective regulations. |
| Data exclusions | In experiment 1A, pupil data of one participant had to be excluded due to too many missing data points (more than 30% missing data during baseline/self-regulation of pupil size for more than 50% of trials, pre-set exclusion criterion). Further ECG data (control group II) of 1 participant on day 2 and 1 participant for all days was corrupted after data saving. For 3 participants, there was incomplete trigger information due to technical issues with the trigger box for no feedback trials at day 3. In experiment 1B, pupil data of one participant had to be excluded due to technical issues with the eye tracker and its recording. Further, ECG data of one session of a participant (experiment 1B) needed to be excluded due to poor data quality (non-detectable R-peaks) In experiment 2, pulse rate data of 2 participants needed to be excluded due to technical issues during recording (whole brain session; n=1), and poor and noisy data quality with barely detectable R-peaks (brainstem session; n=1); brainstem fMRI data of 3 participants needed to be excluded due to excessive motion (pre-set criterion, n=1), distortions (n=1), and falling asleep inside the scanner (n=1); whole-brain fMRI data of 1 participant needed to be excluded due to distortions in frontal and temporal regions. Additionally, 1 run for 3 participants and 2 runs for 1 participant needed to be excluded due to excessive motion (pre-set criterion of half a voxel size mean displacement). In experiment 3, pupil data of 2 participants needed to be excluded due to the same reasons as described for Experiment 1A (pre-set criterion). For 1 of these 2 participants, behavioral data was excluded as well (correct responses during the task deviated more than 3SD from the group mean) |
| Non-participation | No participant has declined participation after recruitment. One participant of the control group of Experiment 1A dropped out due to personal reasons after the first day of training; from Experiment 2 to 3, 3 participants dropped out due to personal reasons (e.g., moving, no time to participate) |
| Randomization | Participants were allocated randomly. |

# Reporting for specific materials, systems and methods

We require information from authors about some types of materials, experimental systems and methods used in many studies. Here, indicate whether each material, system or method listed is relevant to your study. If you are not sure if a list item applies to your research, read the appropriate section before selecting a response.

## Materials & experimental systems

| n/a | Involved in the study |
|---|---|
| ☒ ☐ | Antibodies |
| ☒ ☐ | Eukaryotic cell lines |
| ☒ ☐ | Palaeontology and archaeology |
| ☒ ☐ | Animals and other organisms |
| ☒ ☐ | Clinical data |
| ☒ ☐ | Dual use research of concern |

## Methods

| n/a | Involved in the study |
|---|---|
| ☒ ☐ | ChIP-seq |
| ☒ ☐ | Flow cytometry |
| ☐ ☒ | MRI-based neuroimaging |

## Magnetic resonance imaging

### Experimental design

| | |
|---|---|
| Design type | task-based fMRI with a block design |
| Design specifications | per fMRI session: 8 blocks à 4 trials of each condition (up- vs. downregulation). Each block was of ~2.5 minutes length and between each block, there was a 10 s break. Between the trials of a block there was a jittered break of 6-9 s. 1 fMRI run consisted of 2 blocks of each condition (~10 min) after which there was a break for the participants. |
| Behavioral performance measures | We did not record any behavioral data such as button presses but pupil size throughout the experiment. Participants were asked to apply pupil self-regulation while we simultaneously recorded fMRI data. On our control screen, we saw the pupil-based biofeedback as an immediate check of task performance (similar to the participants themselves) and |

additionally investigated the pupil up- and downregulation (mean across all trials of each condition for each participant) and the standard deviation/standard error of the mean across participants in an offline anaylsis

## Acquisition

| | |
|---|---|
| Imaging type(s) | functional, structural |
| Field strength | 3T |
| Sequence & imaging parameters | T1-weighted anatomical sequence: MPRAGE; 160 sagittal slices, TR/TE: 9.3/4.4 ms; voxel size: 0.7 mm^3; matrix size: 240 x 240, flip angle: 8°; field of view 240mm x 240 mm x 160 mm<br>TSE structural scan (not reported in the present analysis): 20 slices, 1.5 mm slice thickness, TR/TE: 500/10 ms; flip angle: 90°, in-plane resolution: 0.7 X 0.8 mm<br>brainstem fMRI: EPI, 39 sagittal slices (thickness 1.8mm); TR: 2.5s, TE: 26ms; flip angle 82°; SENSE acceleration factor: 2.1; in-plane resolution of 1.8 x 1.8 mm (+ 1 whole-brain single volume EPI image with the same parameters to improve co-registration). FOV: 210 mm x 187.6 mm x 70.2 mm<br>whole-brain fMRI: EPI, 40 slices (thickness: 2.7 mm); TR: 2.5s, TE: 30ms; flip angle: 85°, FOV: 223 mm x 223 mm x 116 mm, SENSE factor 2. In-plane resolution of 2.7 x 2.7 mm. (first 7 participants, EPI with slightly different sequence parameters: 36 slices with 3 mm thickness, FOV: 210 mm x 210 mm x 108 mm, SENSE factor 2.2; in plane resolution of 2 x 2 mm.; flip angle 82°, same TR/TE) |
| Area of acquisition | both whole-brain scanning and limited field of views were used (for brainstem imaging). The field of view was determined by our main regions of interested in the brainstem that needed to be covered (mainly Locus Coeruleus but also more anterior nuclei including the Nucleus Basalis of Meynert). The fourth ventricle aided orientation of the scans. |
| Diffusion MRI | ☐ Used ⊠ Not used |

## Preprocessing

| | |
|---|---|
| Preprocessing software | We used FSL version 6.0.5.2 and FreeSurfer version 6.0 for preprocessing of fMRI data. Brain extraction was performed using FSL's automated brain extraction tool (BET; v2.1), motion correction using the Linear Image Registration Tool (MCFLIRT), spatial smoothing using a 3mm full width-at-half-maximum (FWHM) Gaussian kernel for the brainstem data and 5mm FWHM Gaussian kernel for the whole-brain data, and a 90s high-pass temporal filter as implemented via FSL's Expert Analysis Tool (FEAT, v6.0). For noise and artifact removal, see section below. For additional control analyses of the brainstem fMRI data, we re-analyzed the data without applying any spatial smoothing. |
| Normalization | brainstem fMRI data: Image co-registration from functional to MNI standard space was performed by (i) aligning the functional images of each run to each subject's whole-brain EPI image, using FSL's Linear Image Registration Tool (FLIRT) employing a mutual information cost function and six degrees of freedom; (ii) registering whole-brain EPIs to structural T1-weighted images, using a mutual information cost function six and degrees of freedom, and then optimised using boundary-based registration (BBR); (iii) co-registering structural images to the MNI152 template via FSL's nonlinear registration tool (FNIRT) using 12 degrees of freedom, and applying the resulting warp fields to the functional images. Each co-registration step was visually inspected using Freeview (FreeSurfer, version 6.0).<br>Whole-brain data: similar to the description above: Functional EPI-images were aligned to structural T1-weighted images using linear registration (FLIRT). Structural images were aligned to the MNI standard space using nonlinear registration (FNIRT) |
| Normalization template | we used the MNI152 standard-space structural template image provided by FSL |
| Noise and artifact removal | we implemented motion correction (using the Linear Image Registration tool implemented in FSL; MCFLIRT) for both whole-brain and brainstem data, independent component analyses (using MELODIC implemented in FSL) and physiological noise modeling (using FSL's PNM, an extended version of RETROICOR; the latter only for brainstem data). For PNM, we used physiological recordings from the peripheral pulse sensor and respiration data during scanning |
| Volume censoring | we did not apply additional volume censoring in addition to the noise and artifact removal described above. |

## Statistical modeling & inference

| | |
|---|---|
| Model type and settings | To assess fMRI task-related activity, we used univariate models for data analyses. For the brainstem session's first-level analyses, we implemented fixed-effects analyses using FEAT with FMRIB's Improved Linear Model (FILM) integrated for local autocorrelation. To account for potential differences in the brainstem's HRF, we used the FLOBS toolkit implemented in FSL with the default optimal basis function to model the HRF. PNM voxelwise confound lists and ICA noise component time series were added as nuisance regressors. For second level and third level analyses we, however, only passed up the canonical HRF parameter estimates. Third level analyses were performed using mixed effects analyses.<br>For the whole-brain session's data analyses, we took a similar approach, however, we used a double gamma HRF (instead of the optimal basis functions to model the HRF) and its first temporal derivative. White matter, cerebrospinal fluid time series, motion parameters were added as nuisance regressors. |
| Effect(s) tested | GLM analyses of whole-brain and brainstem data: we contrasted brain activity during our two task conditions (up- vs. downregulation of pupil size).<br>As complementary analyses, we performed additional GLMs for both whole-brain and brainstem data using pupil diameter recorded throughout the session as regressor of interest in the models.<br>Our primary ROI analyses (performed on brainstem data) was implemented to compare brain activity in up- vs downregulation phases of pupil self-regulation as well as the correlation with recorded pupil diameter throughout the pupil |

self-regulation task in our a-priori defined ROIs. Furthermore, we compared up- vs. downregulation phases of self-regulation in additional control ROIs.

Specify type of analysis: ☐ Whole brain ☐ ROI-based ☒ Both

Anatomical location(s) | Anatomical locations to test our primary hypothesis were based on a-priori hypothesis and we used probabilistic atlases provided by previous studies

Statistic type for inference (See Eklund et al. 2016) | for the two GLMs (whole-brain data), we used a cluster-wise approach with a threshold of z > 3.1 and a cluster p threshold of < 0.05
for the GLMs of the brainstem data, we implemented a cluster-wise approach with a threshold of z > 2.3 and a cluster p threshold of < 0.05. Additionally, and only reported in the supplementary information, we performed analyses at an uncorrected voxel p threshold of < 0.05.

Correction | Primary ROI analyses contrasting brain activity between up- and downregulation phases as well as the correlation with recorded pupil diameter throughout the task in our a-priori defined ROIs: we used the sequential Bonferroni procedure to correct for multiple comparisons.
GLMs for whole-brain data: FWE-corrected for multiple comparisons.
GLMs for brainstem data: FWE-corrected for multiple comparisons; additional supplementary analyses were performed at an uncorrected p < 0.05.

## Models & analysis

| n/a | Involved in the study |
|---|---|
| ☒ | ☐ Functional and/or effective connectivity |
| ☒ | ☐ Graph analysis |
| ☒ | ☐ Multivariate modeling or predictive analysis |

