## [Peer Review File · Nature Human Behaviour]

Peer Review Information

Journal: Nature Human Behaviour

Manuscript Title: Self-regulating arousal via pupil-based biofeedback

Corresponding author name(s): Sarah Nadine Meissner and Nicole Wenderoth

Reviewer Comments & Decisions:

Decision Letter, initial version:

15th March 2023

Dear Dr Meissner,

Thank you once again for your manuscript, entitled "Through the eye to the brain: Modulating locus coeruleus-mediated arousal via pupil-based neurofeedback", and for your patience during the peer review process.

Your Article has now been evaluated by 3 referees. You will see from their comments copied below that, although they find your work of potential interest, they have raised quite substantial concerns. In light of these comments, we cannot accept the manuscript for publication, but would be interested in considering a revised version if you are willing and able to fully address reviewer and editorial concerns.

In particular, you will see that Reviewer #2 raised concerns over the design of the sham condition, namely over the statement (line 649-650) that "to prevent the development of illusory correlations, participants were informed that this will occur at a fixed rate". We asked Reviewer #1 for their opinion on this issue and they confirmed that if participants were indeed aware that the sham condition was not a true neurofeedback session, more data would need to be collected with a sham condition that participants were not aware of (i.e. blinded).

Therefore, we would only be able to consider a revision of this manuscript if, either:

- a) It is the case that the participants in the sham group did in fact believe that the feedback they were receiving was controlled by their performance in the neurofeedback task, or
- b) You conduct a suitable replication study with a truly blinded sham condition.

We hope you will find the referees' comments useful as you decide how to proceed. If you wish to submit a substantially revised manuscript, please bear in mind that we will be reluctant to approach the referees again in the absence of major revisions including those mentioned. We are committed to providing a fair and constructive peer-review process. Do not hesitate to contact us if there are specific requests from the reviewers that you believe are technically impossible or unlikely to yield a meaningful outcome.

Finally, your revised manuscript must comply fully with our editorial policies and formatting requirements. Failure to do so will result in your manuscript being returned to you, which will delay

its consideration. To assist you in this process, I have attached a checklist that lists all of our requirements. If you have any questions about any of our policies or formatting, please don't hesitate to contact me.

If you wish to submit a suitably revised manuscript, we would hope to receive it within 4 months. I would be grateful if you could contact us as soon as possible if you foresee difficulties with meeting this target resubmission date.

- Include a "Response to the editors and reviewers" document detailing, point-by-point, how you addressed each editor and referee comment. If no action was taken to address a point, you must provide a compelling argument. When formatting this document, please respond to each reviewer comment individually, including the full text of the reviewer comment verbatim followed by your response to the individual point. This response will be used by the editors to evaluate your revision and sent back to the reviewers along with the revised manuscript.
- Highlight all changes made to your manuscript or provide us with a version that tracks changes.

[REDACTED]

Thank you for the opportunity to review your work. Please do not hesitate to contact me if you have any questions or would like to discuss the required revisions further.

Sincerely,
Jamie

Dr Jamie Horder
Senior Editor
Nature Human Behaviour

REVIEWER COMMENTS:

Reviewer #1:

Remarks to the Author:

Meissner and colleagues, in three experiments (plus a direct replication of the primary effect), investigated if healthy humans can be trained to voluntarily control the size of their pupil. They further examined if such modulations of pupil size were accompanied by fMRI activity changes in the brainstem and other parts of the brain, by changes in physiological markers of arousal (heart rate and variability), and by behavioral performance on an oddball task (RT and RT variability).

The manuscript is timely and addresses questions with potentially important implications for clinical applications of the volitional control of arousal, and it is thus of interest to a broad audience. It contains an impressive set of experiments that are generally well designed and include relevant

controls. It is well written and contains a clear exposition of the results, relevant literature, and interpretation. The main strength of the manuscript is that it (in my view) convincingly shows that providing participants with feedback on the size of their pupil improves participants' ability to modulate pupil size, dilating or constricting the pupil on demand.

Nevertheless, the manuscript contains a number of shortcomings that need to be addressed. The most prominent of these are that the manuscript is too specifically framed in terms of the neural mechanisms (locus coeruleus) that are putatively responsible for the observed modulations of pupil size. In addition, the experiment that aims to actually test the neural underpinnings of these effects lacks controls and contains inappropriate statistical procedures (specifics below).

1) The manuscript is written in terms that are quite explicit about the neural underpinnings of changes in pupil size. In my view this is framed in a way that is too specific and unjustified given the data as they are currently presented (see points below) and needlessly so. Voluntary control of pupil dilation / constriction is an interesting finding in its own right, regardless of the exact neurophysiological underpinnings. Moreover, feedback on the pupil itself does not allow for exact control of specific neuromodulatory systems because these systems do not operate in isolation, nor is any such system the sole source of changes in pupil size. The manuscript would thus benefit from a re-write in terms that are more generally about the modulation of arousal rather than specifically targeting the locus coeruleus noradrenaline system. In this case there would be room in the discussion for suggesting the involvement of the locus coeruleus in the observed effects.

2) To establish the specificity of the observed fMRI changes that accompany pupil dilation / constriction, a negative control region is necessary. That is, do any differences between the UP and DOWN conditions in specific ROIs exceed the same contrast in e.g. the 4th ventricle? Moreover, the effects in Figure 3b appear on the weaker side and raise the question if they are specific to the neuromodulatory nuclei (as implied in the text) or if they are generally weakly present across the pons and midbrain. It would thus be useful to compute the same contrast for a mask that includes the whole pons / midbrain / brainstem, and demonstrate if the effects within specific nuclei exceed that within the larger mask. Note also that in a number of instances comparative statements are made (e.g. "activity changes in brain nuclei regulating the brain's arousal levels and in the LC in particular", page 18) that would require an actual comparison of the effect in the LC relative to other regions.

To be clear, if any difference between conditions would be present throughout the brainstem, that would not make the findings less interesting, but it would call for an interpretation of the results that more appropriately captures the general pattern.

3) In Figures 3D and 3E, voxel-level contrasts are presented that are uncorrected for multiple comparisons. As justification for this the authors argue that the brainstem has limited signal-to-noise ratio (page 9, line 205). While it is true that the brainstem has limited SNR, it is a strong argument in favor of applying multiple comparisons correction rather than an argument for omitting it. If the purpose of this figure is only to provide a qualitative picture of the findings, then it would be more suitable to present the contrasts without any threshold instead of an inappropriate statistical threshold.

4) The statistics for a number of important tests seem to be missing.

a) Statistics that indicate if pupil diameter increases with respect to baseline within the NF group and the control groups (Figures 2A-B)

b) Post-hoc tests comparing the NF and control groups within testing days (Figure 2C)

c) Comparison of the UP and DOWN conditions in Figures 3A and 4A

Some of these are very likely to be significant, but some of them may not be.

5) Figure 2B and Figure 2D suggest that there is an effect in the pupil present for participants who received sham feedback, and for those who received no feedback and had not yet received training. This could be interesting in itself because it would suggest that even with minor instructions people are able to exert some control over their pupil. If this is indeed the case then it would be more accurate to state that feedback enhances this ability rather than enables it outright.

6) Some of the results in Figure 6 may have alternative explanations.

a) If I am not mistaken then the correlation between baseline pupil size and pupil dilation amounts to a correlation between a variable X and another variable Y-X. As such it is susceptible to regression towards the mean effects. This can be ruled out by computing the p-value of the correlations using a permutation procedure in which the trial-wise relationship between X and Y is shuffled before subtracting them.

b) Mechanical constraints of the pupil may account for some of the effects (rather than a trade-off between phasic and tonic LC activity). That is, if the pupil is fully dilated, it cannot dilate further upon hearing an oddball sound. Conversely, if the pupil is fully constricted, the possible range for any subsequent dilation is maximal. This should be discussed.

c) The relationship between baseline pupil diameter and RT is possibly confounded by a time-on-task effect. If there are progressive changes in both RT and in pupil diameter over the course of an experimental block or session, then such a linear relationship between RT and pupil will appear. This can be ruled out by linearly regressing out time-on-task from both RT and pupil diameter across trials before computing the correlation between them.

7) In essence the findings in Figure 6 show that various additional tasks (up-regulation of pupil, down-regulation of pupil, and counting backwards) on top of a regular oddball task each modulate oddball task performance in different ways. Counting results in worst performance on the oddball task, and down-regulating the pupil results in the best performance. However, these findings do not necessarily show that down-regulating pupil diameter actually improves performance relative to having no additional task. The latter would be required in order to draw wider conclusions about *improving* task performance or *facilitating behavioral responses* through modulating arousal. Any discussion of this limitation would be welcome.

8) Are participants generally under-aroused (e.g. tired) and can this account for the difference in the ease with which participants improve the ability to up or down-regulate pupil size? That is, if participants generally have relatively constricted pupils due to under-arousal, is this the reason they are able to dilate them seemingly more easily than constrict them?

9) I am unsure if the term 'neurofeedback' (in the title and throughout the manuscript) is appropriate, since the feedback participants receive is not on a neural signal. Perhaps the term biofeedback would be better.

10) The pathway of the LC to the Edinger-Westphal nucleus (discussed on e.g. Page 10): This pathway is known to exist in cats, but to my knowledge has not yet been definitively shown to exist in primates. Any correlation between LC activity and pupil diameter could also come about through shared input to the sympathetic nervous system and LC by the rostral ventrolateral medulla (See discussion in Nieuwenhuis, De Geus & Aston-Jones, 2011, *psychophysiology*; 10.1111/j.1469-8986.2010.01057.x).

11) Methods points / questions:

a) The authors acquired neuromelanin sensitive scans that can be used to delineate participant-specific masks of the locus coeruleus, which would greatly enhance the accuracy of the LC localization. Why did the authors rely on an atlas instead of using these scans?

- b) Were correlations between pupil diameter and fMRI activity adjusted for delays due to the pupillary response? Simply convolving the pupil signal with a canonical HRF will mean that the modeled hemodynamic signal lags the observed hemodynamics because the pupil signal itself lags neural activity. The HRF-convolved signal is typically shifted backwards in time because of this (see e.g. Yellin, Berkovich-Ohana & Malach, 2015, *NeuroImage*; 10.1016/j.neuroimage.2014.11.034).
- c) Page 28: "Missing data was replaced by the group mean so as not to further reduce statistical power". This does not seem appropriate because it artificially inflates the degrees of freedom.
- d) Spatial smoothing was applied in the analysis of brainstem signals, which is suboptimal because it smears activity across adjacent nuclei, as well as across the ROIs and 4th ventricle.

Other points:

Page 17: "Our results reveal that self-regulating pupil diameter as a proxy of LC-mediated arousal facilitates adequate behavioral responses as predicted by current theories of noradrenergic function". I am not sure this is entirely correct. Prominent accounts (e.g. Adaptive Gain theory) predict an inverted-U relationship between arousal and performance rather than a linear one.

Page 18: "healthy adults can learn to self-regulate pupil size during a 3-days training with veridical pupil-NF but not with sham". The results in figure 2 suggest that even with sham feedback there is some effect (see point 5 above).

Page 20 and further: "Our behavioral findings further confirm previous studies reporting enhanced task performance with a relatively low baseline pupil diameter ...". The findings in this literature are rather mixed, with some studies reporting effects in this direction, but many others reporting effects in the opposite direction (examples below, no need to cite them). Perhaps it's best to simply report the findings as they are rather than make them appear in line with the broader literature.

- Kristjansson et al (2009) Detecting phasic lapses in alertness using pupillometric measures. *Appl Ergon* 40: 978-986.
- Smallwood et al. (2011) Pupillometric evidence for the decoupling of attention from perceptual input during offline thought. *PLoS One* 6: e18298.
- Smallwood et al. (2012) Insulation for daydreams: a role for tonic norepinephrine in the facilitation of internally guided thought. *PLoS One* 7: e33706.
- Franklin et al. (2013) Window to the wandering mind: pupillometry of spontaneous thought while reading. *Q J Exp Psychol* 66: 2289-2294.
- Grandchamp et al. (2014) Oculometric variations during mind wandering. *Front Psychol* 5: 31.
- Mittner et al. (2014) When the brain takes a break: a model-based analysis of mind wandering. *J Neurosci* 34: 16286-16295.
- Hopstaken et al. (2015) A multifaceted investigation of the link between mental fatigue and task disengagement. *Psychophysiology* 52: 305-315.
- Hopstaken et al. (2015) The window of my eyes: Task disengagement and mental fatigue covary with pupil dynamics. *Biol Psychol* 110: 100-106.
- Unsworth and Robison (2016) Pupillary correlates of lapses of sustained attention. *Cogn Affect Behav Neurosci* 16: 601-15.
- van den Brink et al. (2016) Pupil diameter tracks lapses of attention. *PLoS One* 11: e0165274
- Konishi et al. (2017) When attention wanders: Pupillometric signatures of fluctuations in external attention. *Cognition* 168: 16-26
- Unsworth and Robison (2018) Pupillary correlates of fluctuations in sustained attention. *J Cog Neurosci* 30: 1241-1253

Reviewer #2:

Remarks to the Author:

Through the eye to the brain: Modulating locus coeruleus-mediated arousal via pupil-based neurofeedback

Summary: the authors describe a series of experiments involving pupil-based neurofeedback in healthy individuals. In the first experiment, they show that pupil-based feedback can be used to regulate pupil size in a group of 27 individuals compared to a sham control group of the same size receiving yoked feedback explicitly described as sham to the participants. Participants trained for 3 sessions. The feedback was used to regulate in both directions. Participants were provided with suggested mental strategies to use for up-regulation as well as down-regulation. No feedback trials performed pre- and post-training showed a significant improvement in the feedback group compared to sham. In a second experiment, the authors took participants from the feedback group and performed fMRI of the whole brain and brainstem independently during self-regulation to observe which parts of the brain were correlated with such control. Using uncorrected statistics but with independent component analysis and preprocessing techniques intended to remove physiological artifact, the authors found several brainstem areas that correlated with up-regulation vs. down-regulation hypothesized earlier to be physiologically relevant to the task. The researchers found that pupil size self-regulation was correlated with a change in heart rate. In a third experiment, the authors used an auditory oddball task on those who performed pupil size feedback training to investigate whether such stimuli affect pupil dilation response to different sounds. They found that indeed, the ability to self-regulate pupil size was affected by the oddball task, an indication that locus coeruleus was involved in self-regulation. The authors conclude that pupil size can be self-regulated, that a network of salient brain areas are involved in this process, particularly the locus coeruleus, and that such self-regulation is correlated with physiologically relevant behaviors such as heart rate. Overall, I believe this is very solid and innovative work. The experiments were well-planned out and the results are mostly convincing. There are some questions, however:

Major issues:

- The bidirectional control of pupil size is very convincing. My biggest concern here is the sham condition. If I understand correctly, the authors inform the participants that the sham is indeed sham feedback, correct? This is not typical in experiments using such sham that I am familiar with and could potentially result in reduced attention of the sham group and thus worse performance. It would appear that the authors did not want to induce "illusory correlations" (line 650) between the sham feedback and the pupil size, but I believe this is exactly the purpose of sham feedback. Could the authors clarify their intent here and defend the use of this control?
- My experience with uncorrected statistics in fMRI data has been that there is no true rule of thumb, which is very frustrating because then a paper comes along that tries to set new standards and some fall below those standards. In lines 474-482, the authors explain their use of physiological noise correction and it's clear they did the best they could, but I still feel uncomfortable with uncorrected statistics. I would like to see the authors provide some objective defense of this, i.e. not that someone else did it.
- Not really a major issue, but something that needs to be corrected - "The effectiveness of self-regulation critically depends on whether participants can acquire a suitable mental strategy". I would disagree with this statement as a number of neurofeedback studies including those from the group of Mitsuo Kawato were able to show the ability to self-regulate a prescribed set of voxels without any relevant mental strategy.
- I do not recall seeing any stats table on Down > Up contrasts. Is there anything to be learned there?
- A better lead in to the purpose behind Experiment 3 would be appreciated, i.e., explaining the significance of the modulation of LC of auditory oddball task. Reading through the rest of the manuscript it becomes clear, but not until then.

- I question the use of p-values for correlations in Fig 6F as they mean less when number of data points increase. Given the amount of data, the regression value is more informative and 0.13 seems very weak. I'm not sure I'm on board with the conclusion of pupil dilation being related to baseline without some extra bootstrapping here.
- In line 752, the authors use mean data to replace missing data. My experience is that this is not an acceptable practice, but I would like to hear from the authors on why this was necessary.
- I'm a little skeptical of the conclusion between pupil NF and heart rate. Fig 5E shows a good correlation between pupil modulation index and change in heart rate only after training, however, it may not be the feedback training, but the repeated exposure to the explicit mental strategies. I would think having a differential correlation with controls here would be more convincing.
- I'm a bit concerned about the change in sequence mentioned in Line 815. How do we know the change in sequence didn't affect results? Was there any subsequent analysis?

Minor issues:

- Line 290, should it be "trended towards significance"? "tended" does not seem right here.
- Line 333, "suggests"
- Line 392, comma instead of decimal in "94.3"
- Line 460, no comma needed after "diameter"
- Line 756, define "RMSSD"

Reviewer #3:

Remarks to the Author:

The authors present evidence in support of the hypothesis that humans can use pupil-based closed-loop feedback to volitionally up and down regulate pupil size and activity in arousal-related brain structures including the Locus Coeruleus.

Evidence is provided in three parts: First, through two experiments set up as between- (Exp. 1-A) and within-subject (Exp. 1-B) design respectively, for which the authors found significant differences in pupil-diameter and heart rate (but not heart rate variability) as a function of closed-loop pupil-feedback condition. Second, through a study that uses functional magnetic resonance imaging (Exp. 2), for which the authors present significant pupil-feedback-related effects in the brainstem as well as the whole brain. Third, through behavioral effects that the authors observed while subjects self-regulated their pupil-size while performing an odd-ball task (Exp. 3).

In my assessment, the evidence in support of the authors' hypothesis that humans can volitionally regulate pupil size and activity in the brainstem is convincing and represents a significant scientific advancement. I believe that this manuscript will be of great interest to readers interested in closed-loop feedback and other disciplines.

I could not find any major flaws around data, methods or technical details. I provide some more detailed questions and comments below, including about how sham feedback was generated. While I could not find information related to pre-registration for the studies that were performed, I also could not find any indication for any major concerns about how the studies were conducted.

In my assessment, all statistical methods used, have been applied correctly and all analyses were appropriately powered.

The authors correctly interpret individual test outcomes and integrate the evidence from across experiments and approaches appropriately. What adds to the confidence in the robustness of these findings is that the authors present evidence that includes between- as well as within-subject experiments, replication in independent samples of novice participants, multiple physiological signals

including pupil size, electrocardiography and fMRI as well as behavioral results. Ultimately, it remains difficult to answer this research question with absolute certainty, which is a topic the authors address appropriately as part of the limitations of this study in the discussion. The manuscript is very well written and from what I can tell cites all necessary literature appropriately.

More minor comments and questions:

Could the authors please clarify what “quasi real-time” means concretely?

L104: Could you clarify concretely what is meant when you write that the control group “did not receive veridical feedback”? How exactly was the sham feedback generated? If it helps, please provide equations, to make this important topic unambiguously clear for the reader.

L130: I have found it a bit difficult to understand how the pupil-modulation index is defined. Could you maybe include the formula?

L274: Why was ECG recorded for only 15 subjects? If subjects were excluded, e.g. due to insufficient data quality, please explain the details (unless they are already included in the report to the publisher).

L286: Where you explain that RMSSD is or may be linked to changes in activity of the parasympathetic nervous system, could you add the missing reference? Please consider trying a different short-term HRV metric, called pNN-X.

Here, a paper that uses pNN-X:

Mietus, J. E., Peng, C. K., Henry, I., Goldsmith, R. L., & Goldberger, A. L. (2002). The pNNx files: re-examining a widely used heart rate variability measure. *Heart*, 88(4), 378-380.

Here, additional literature on interpreting short-term HRV:

Berntson, G. G. & Cacioppo, J. T., (2004) Heart rate variability: Stress and psychiatric conditions. In Malik, M. & Camm, A. J. (eds.), *Dynamic electrocardiography*, 57–64 (John Wiley & Sons).

Malik, M., & Camm, A. J., (1993). Components of Heart Rate Variability – What they really mean and what we really measure. *Am J Cardiol*, 72, 821-822.

Here, another paper where the metric was used for a similar purpose:

Faller, J., Cummings, J., Saproo, S., & Sajda, P. (2019). Regulation of arousal via online neurofeedback improves human performance in a demanding sensory-motor task. *Proceedings of the National Academy of Sciences*, 116(13), 6482-6490.

L343: Where you write “714-3000ms”; if you intend to say “714 to 3000 ms”, then please consider using the latter way to write this, as a best practice and to avoid confusion with a minus sign.

L483: Faller and colleagues provided feedback based on EEG and observed changes in pupil size that were interpreted as linked to LC activity – could that study (with some caveats) be seen as supporting the idea you formulate at the end of this paragraph?

L516, paragraph: A similar effect was observed by Faller and colleagues for pupil size in the sham condition by Faller and colleagues (see Figure F, Panel B).

L568: The sentence starting with "Developing an eye blink-related strategy..." might be correct, but seems unnecessarily hard to parse – please consider to rephrase it.

L573: What was the luminance setting in the room? Were these studies conducted in an office environment or inside a specialized noise-shielded room or box?

L611: Could you comment on the degree of fidelity you expect from how you calculate luminance from RGB color values via the formula you mention. My understanding is, that using a formula such as this is seen as a crude approximation of luminance. If appropriate, please consider discussing this topic as a limitation.

A photo or better even video of the paradigm – maybe as part of the supplementary material - would be very helpful.

Generally, I recommend to use box- or violin rather than bar plots.

Could you clarify whether these studies were conducted in a double-blind manner where appropriate? If appropriate please consider discussing this as a limitation.

Author Rebuttal to Initial comments

Reviewer #1:

Remarks to the Author:

Meissner and colleagues, in three experiments (plus a direct replication of the primary effect), investigated if healthy humans can be trained to voluntarily control the size of their pupil. They further examined if such modulations of pupil size were accompanied by fMRI activity changes in the brainstem and other parts of the brain, by changes in physiological markers of arousal (heart rate and variability), and by behavioral performance on an oddball task (RT and RT variability).

The manuscript is timely and addresses questions with potentially important implications for clinical applications of the volitional control of arousal, and it is thus of interest to a broad audience. It contains an impressive set of experiments that are generally well designed and include relevant controls. It is well written and contains a clear exposition of the results, relevant literature, and interpretation. The main strength of the manuscript is that it (in my view) convincingly shows that providing participants with feedback on the size of their pupil improves participants' ability to modulate pupil size, dilating or constricting the pupil on demand.

Nevertheless, the manuscript contains a number of shortcomings that need to be addressed. The most prominent of these are that the manuscript is too specifically framed in terms of the neural mechanisms (locus coeruleus) that are putatively responsible for the observed modulations of pupil size. In addition, the experiment that aims to actually test the neural underpinnings of these effects lacks controls and contains inappropriate statistical procedures (specifics below).

1) The manuscript is written in terms that are quite explicit about the neural underpinnings of changes

in pupil size. In my view this is framed in a way that is too specific and unjustified given the data as they are currently presented (see points below) and needlessly so. Voluntary control of pupil dilation / constriction is an interesting finding in its own right, regardless of the exact neurophysiological underpinnings. Moreover, feedback on the pupil itself does not allow for exact control of specific neuromodulatory systems because these systems do not operate in isolation, nor is any such system the sole source of changes in pupil size. The manuscript would thus benefit from a re-write in terms that are more generally about the modulation of arousal rather than specifically targeting the locus coeruleus noradrenaline system. In this case there would be room in the discussion for suggesting the involvement of the locus coeruleus in the observed effects.

Authors' response:

We appreciate the reviewer's input and have revised all sections of our manuscript including restructuring the introduction section and the conceptual Figure 1. Our aim was to emphasize that several, interconnected neuromodulatory systems control arousal even though LC is considered to play a key role. Even though we now put the main focus more on general arousal, we decided to maintain the special role of the LC-NA system in some parts of the introduction, as the link between the noradrenergic system and non-luminance-related pupil size changes is very well supported by literature, especially in attention tasks such as the oddball paradigm used in Experiment 3. We acknowledge that this system does not work in isolation, and we have made efforts to clarify this, especially in the discussion section. We hope that these revisions meet the reviewer's requirements. If not, we would welcome further feedback on this issue.

Please find the changes in the abstract (line 22-24), introduction (restructuring of sections page 3 and 4, specific changes page 3, line 59-60, 70, 81-82), results (page 6, line 139-140, 149, page 7, line 195-198) and discussion (page 11, 320-321, 324-325, page 14, 410-411) conclusion: page 15, line 462-474).

- 2) To establish the specificity of the observed fMRI changes that accompany pupil dilation / constriction, a negative control region is necessary. That is, do any differences between the UP and DOWN conditions in specific ROIs exceed the same contrast in e.g. the 4th ventricle? Moreover, the effects in Figure 3b appear on the weaker side and raise the question if they are specific to the neuromodulatory nuclei (as implied in the text) or if they are generally weakly present across the pons and midbrain. It would thus be useful to compute the same contrast for a mask that includes the whole pons / midbrain / brainstem, and demonstrate if the effects within specific nuclei exceed that within the larger mask. Note also that in a number of instances comparative statements are made (e.g. "activity changes in brain nuclei regulating the brain's arousal levels and in the LC in particular", page 18) that would require an actual comparison of the effect in the LC relative to other regions.

To be clear, if any difference between conditions would be present throughout the brainstem, that would not make the findings less interesting, but it would call for an interpretation of the results that more appropriately captures the general pattern.

Authors' response:

We thank the reviewer for this valuable input. Following the suggestions, we repeated the same analyses of up- versus downregulation for a mask covering the complete brainstem including midbrain,

pons, and the medulla oblongata, and for a mask covering the 4th ventricle (methods: page 30, line 971-979, results: page 7, line 172-176). We found no statistical evidence that the BOLD signal extracted from the 4th ventricle would differ significantly between UP versus DOWN ($t(21) = 0.09$; $p_{corrected} = 1$). In fact, a Bayesian post-hoc analysis revealed moderate evidence in favour of H_0 : $BF_{10} = 0.224$, error % = 0.02 (i.e. no difference between UP versus DOWN; Supplementary Figure 5B).

For the brainstem mask including the midbrain, we found a trend level effect but only at uncorrected levels (complete brainstem: $t(21) = 2.00$; $p = 0.12$, $p_{uncorrected} = 0.06$; Supplementary Figure 5B). Accordingly, the Bayesian post-hoc analysis revealed only anecdotal evidence for H_1 for the brainstem mask ($BF_{10} = 1.177$, error % = 0.02). Next, we compared whether differences between up- and downregulation in each of our pre-defined ROIs showing significant differences between up- and downregulation (i.e., LC, SN/VTA, and NBM which showed trend-level effects in our main analyses) were significantly larger than up- versus downregulation differences extracted from the whole brainstem. This comparison reached significance for the LC ($z = 2.45$; $p = 0.04$) but not for the SN/VTA and NBM after correcting for multiple comparison (SN/VTA: $z = 1.93$; $p = 0.10$; NBM: $z = 1.87$; $p = 0.06$; Supplementary Figure 5A-C).

Together these additional analyses indicate that up- versus down-regulation does not seem to modulate brainstem activity as a whole but that there is anatomical specificity with the strongest statistical effect being found in the LC ROI. However, even though we find strongest effect sizes in the LC when comparing phases of up- with phases of downregulation (similar for no smoothing data, see comment below), we would like to point out that the noradrenergic system does not work in isolation and that pupil up- versus downregulation did affect several brainstem areas that are involved in arousal and autonomic function regulation as listed in Supplementary Table 2A and 2B.

We thank the reviewer for suggesting these additional analyses that are now reported in the Supplementary Figure 5A-C.

- 3) In Figures 3D and 3E, voxel-level contrasts are presented that are uncorrected for multiple comparisons. As justification for this the authors argue that the brainstem has limited signal-to-noise ratio (page 9, line 205). While it is true that the brainstem has limited SNR, it is a strong argument in favor of applying multiple comparisons correction rather than an argument for omitting it. If the purpose of this figure is only to provide a qualitative picture of the findings, then it would be more suitable to present the contrasts without any threshold instead of an inappropriate statistical threshold.

Authors' response:

We agree with the reviewer and thank them for raising this important point. We have now removed all uncorrected, voxel-level fMRI results from the main manuscript and only report them in the supplements for a qualitative comparison. All results reported in the main manuscript are either cluster-corrected (methods, page 30, line 948/949) or were obtained for pre-defined anatomical ROIs and corrected for multiple comparison using sequential Bonferroni correction. Accordingly, changes were made in the result section (page 7, line 184-188), Figure 3, and Supplementary Figure 5.

- 4) The statistics for a number of important tests seem to be missing.
- Statistics that indicate if pupil diameter increases with respect to baseline within the NF group and the control groups (Figures 2A-B)
 - Post-hoc tests comparing the NF and control groups within testing days (Figure 2C)
 - Comparison of the UP and DOWN conditions in Figures 3A and 4A
- Some of these are very likely to be significant, but some of them may not be.

Authors' response:

We thank the reviewer for this valuable input and calculated additional statistics that we added to the main manuscript or supplementary information:

- To compare up- and downregulation to baseline values, we took the mean across the up- and downregulation time series for the modulation and baseline phase of each training session and group and conducted Wilcoxon paired t-tests (results: page 5, line 106-107; since most of these values were not normally distributed; all values are sequential Bonferroni-Holm-corrected for multiple comparisons):*

In the biofeedback group, all upregulation phases were significantly different from the baseline condition, except for the no feedback condition that only trended towards significance after correction for multiple comparisons (up day 1: $z = 4.37$; $p = 0.014$; up day 2: $z = 4.42$; $p = 0.015$; up day 3: $z = 4.23$; $p = 0.013$; up nofb: $z = 2.64$; $p = 0.08$). During downregulation, only downregulation phases on day 3 and during the no feedback session after training were significantly different from baseline after correcting for multiple comparisons (down day 1: $z = 1.73$; $p = 0.08$ (uncorr); down day 2: $z = 2.40$; $p = 0.13$; down day 3: $z = 3.39$; $p = 0.011$; down nofb: $z = 4.54$; $p = 0.016$).

In the control group, however, no upregulation phases and downregulation phases during training were significantly different from the baseline (up day 1: $z = 0.65$; $p = 0.52$ (uncorr); up day 2: $z = 0$; $p = 1$ (uncorr); up day 3: $z = 1.25$; $p = 0.21$ (uncorr); up nofb: $z = 0.24$; $p = 0.81$ (uncorr); down day 1: $z = 2.52$; $p = 0.11$; down day 2: $z = 1.61$; $p = 0.11$ (uncorr); down day 3: $z = 2.40$; $p = 0.13$). The only phase reaching significance was during downregulation after training ($z = 3.40$; $p = 0.012$). Here, even though lower in magnitude, control participant reached some downregulation that was significantly different from baseline pupil size.

Since we focus on the statistics of pupil modulation indices in the main manuscript, we decided to add this information to the supplements (Supplementary Figure 1).

- We agree with the reviewer, that it would be interesting to see differences between groups on different sessions separately. However, since the statistical model only revealed a significant main effect of group but no significant interaction effect between session and group, we are hesitant to add this information to the manuscript.*

For the reviewer's information only, we calculated these post-hoc tests in R which revealed significant differences between groups for each testing session:

Post-hoc tests in R: (formula to test for only the between-subjects main effect with bootstrapping:

*sppba(PupilModulation ~ Group * Session, ID, PupilModulation_Control_Biofeedback, avg = false).
Respective R Output:*

Test statistics:

Estimate

D1 Control vs. BF -0.1391 D2

Control vs BF -0.2063 D3 Control

vs. BF -0.1952 Nofb Control vs.

BF -0.2076

Test whether the corresponding population parameters are the same:

p-value: $p < 0.001$.

Non-parametric Mann Whitney U tests comparing modulation indices of each training session between groups revealed similar results:

D1 BF vs. Control: $U = 587$; $p < .001$ D2 BF

vs Control: $U = 608$; $p < .001$ D3 BF vs

Control: $U = 578$; $p < .001$ Nofb BF vs

Control: $U = 591$; $p < .001$

c) As suggested, we compared up- and downregulation time series statistically using the SPM1D toolbox and found significant differences between these two conditions for both whole-brain and brainstem sessions for most of the modulation phase:

Whole-brain: 1 significant cluster between 965.31ms to 15000ms; $p < 0.001$; $z^ = 3.38$; Brainstem: 1 significant cluster between 983.72ms to 15000ms; $p < 0.001$; $z^* = 3.45$*

We added this additional calculation to the manuscript (methods, page 26, line 840-842; results: page 6, line 147-148) and adjusted Figure 3A and 4A accordingly.

- 5) Figure 2B and Figure 2D suggest that there is an effect in the pupil present for participants who received sham feedback, and for those who received no feedback and had not yet received training. This could be interesting in itself because it would suggest that even with minor instructions people are able to exert some control over their pupil. If this is indeed the case then it would be more accurate to state that feedback enhances this ability rather than enables it outright.

Authors' response:

We thank the reviewer for raising this important point. In addition to testing up- and downregulation against baseline pupil size as reported above in Experiment 1B, we additionally compared the pupil modulation index (time series) against 0 using the SPM1D toolbox prior to training (methods, page 24, line 744-747). This analysis revealed that the pupil modulation index was indeed already significantly different from 0 even before training (Supplementary Figure 4D, solid black line indicates time windows with significant clusters, i.e. significant differences as compared to 0; $p < 0.05$):

Thus, we agree that some ability to control pupil size may be already set free via pure application of mental strategies alone. This ability seems to be significantly enhanced by means of biofeedback. We changed the respective sections in the manuscript accordingly (results: page 5, line 128-131; page 6, line 135-138; discussion: page 11, line 322-323; 332) and added a figure indicating the time windows of significant pupil size regulation prior to pupil-BF training to the supplementary information (Supplementary Figure 4D).

6) Some of the results in Figure 6 may have alternative explanations.

- a) If I am not mistaken then the correlation between baseline pupil size and pupil dilation amounts to a correlation between a variable X and another variable Y-X. As such it is susceptible to regression towards the mean effects. This can be ruled out by computing the p-value of the correlations using a permutation procedure in which the trial-wise relationship between X and Y is shuffled before subtracting them.

Authors' response:

We thank the reviewer for catching this fallacy. We worked with our statistician to address this concern and use now a relative baseline correction (i.e., pupil dilation response/baseline pupil size averaged across 500 ms prior to stimulus onset) as well as a log transform to be able to give an unbiased estimate of the relationship. Even though a relative baseline is less problematic, it's still not free from biases and potentially spurious correlations: the correlation coefficients of the $y/x \sim x$ of randomly generated data centers around -0.7, thus giving a biased estimate of the relationship as shown in the following histogram.

Therefore, instead of computing a repeated measures correlation, we calculated the beta estimates for each participant using the following linear model with an implemented log transformation

y = pupil dilation response peak,
 x = baseline pupil size, and $\log(y)$
 $\sim \log(x)$,

that corresponds to the following equation: $\log(y) = \beta \log(x) + \text{intercept} + \epsilon$. Resolving this formula leads to:

$y/x = x^{\beta - 1} e^{\text{intercept}} e^{\epsilon}$ which is an unbiased estimate of the influence of the baseline pupil size x on the subsequent baseline-corrected pupil dilation response peak y/x .

Here, β estimates of randomly generated data center around 0 as shown in the following histogram:

Thus, testing the resulting coefficient β against 1 would reveal whether there is a true relationship between our pupil baseline value and relative pupil size changes in our data (note that the null hypothesis is $\beta - 1 = 0$; i.e., the baseline has no influence on the relative change of the pupil dilation response in our model). The resulting betas were estimated for each participant separately (mean = 0.80; SD = 0.09) and a one-sample t-test revealed indeed a significant difference from 1 ($t(19) = -10.42$, $p < .001$) indicating a significant and unbiased relationship between baseline pupil size and baseline-corrected pupil dilation responses. We repeated the same analysis for each condition separately. Again, the beta estimates significantly differed from 1 for all conditions ($p < .001$). We added the unbiased analysis approach and its respective results to the methods (page 34-35, line 1106-1121) and results (page 10, line 283-287; Figure 6F, left panel). Within-condition calculations were added to Supplementary Figure 9A.

- b) Mechanical constraints of the pupil may account for some of the effects (rather than a trade-off between phasic and tonic LC activity). That is, if the pupil is fully dilated, it cannot dilate further upon hearing an oddball sound. Conversely, if the pupil is fully constricted, the possible range for any subsequent dilation is maximal. This should be discussed.

Authors' response:

We agree and thank the reviewer for addressing this important point. It is true that mechanical constraints could potentially influence pupil responses during the oddball task. We did not run additional control experiments ourselves, but two previous studies modulated baseline pupil size through different luminance conditions and could show that this did not systematically affect task-evoked pupil dilation responses (Gilzenrat et al., 2010; Reilly et al., 2019). Importantly, luminance-modulated baseline pupil size ranged on average between 2.8 and 4mm (Reilly et al., 2019) and 3.3 and 4.6mm (Gilzenrat et al., 2010), meaning that our reported pupil sizes fall into this modulated range (compare Supplementary Figure 9A). Based on these previous studies, it seems to be more parsimonious to argue that the pupil dilation responses to the oddball stimulus depend on the brain's arousal state reflected by baseline pupil size. We nevertheless added a discussion of this potential limitation in the discussion section on page 14, line 424-428.

- c) The relationship between baseline pupil diameter and RT is possibly confounded by a time-on-task

effect. If there are progressive changes in both RT and in pupil diameter over the course of an experimental block or session, then such a linear relationship between RT and pupil will appear. This can be ruled out by linearly regressing out time-on-task from both RT and pupil diameter across trials before computing the correlation between them.

Authors' response:

This is another very interesting point, the reviewer raises. We are aware that in literature, time-on-task effects are reported but that they, especially regarding pupil diameter changes, lack consistency. Even though Murphy et al., (2011) found a progressive increase over time in baseline diameter during the oddball task, other studies either find no significant change in baseline pupil diameter (Beatty et al., 1982) or rather a decrease over time during cognitive tasks (Hopstaken et al., 2015; van den Brink 2016). For task performance, we'd rather expect a worsening in performance as reported before (Murphy et al., 2011; Beatty et al., 1982) and thus an increase in reaction times across the task. To approach this issue of inconsistent results, we investigated potential time-on-task-effects in our own data. We took a similar approach as published in Arnau et al., 2021 and ran separate linear mixed models on single-trial level with the dependent variable reaction times and pupil baseline, respectively. Time-on-task was entered into the model as a fixed effect and was operationalized as the number of the respective trial in the experiment in ascending order. A random intercept was modelled for each participant (reaction time \sim time on task + (1|participant) and baseline \sim time on task + (1|participant)). We indeed found time-on-task effects as reaction times to target tones increased, while baseline pupil size decreased with the time spent on the task (reaction times: estimate: 0.00039; $t = 7.97$; $p < .001$; baseline pupil: estimate: -0.00075; $t = -6.95$; $p < .001$). We removed the linear trend across trials for both baseline pupil size and reaction times for each participant using the detrend function in MATLAB. Then, we again subjected reaction times and baseline pupil size to a repeated measures correlation in R which revealed a slightly increased correlation coefficient as compared to the original analysis: $r_{rm} = 0.16$; $p < .001$; 95%-CI $_{rm}$ [0.13; 0.20]. The new analyses are now reported in the methods (page 35, line 1122-1132) and results sections (page 11, line 306-311, Figure 6F right panel, Supplementary Figure 9BC).

- 7) In essence the findings in Figure 6 show that various additional tasks (up-regulation of pupil, down-regulation of pupil, and counting backwards) on top of a regular oddball task each modulate oddball task performance in different ways. Counting results in worst performance on the oddball task, and down-regulating the pupil results in the best performance. However, these findings do not necessarily show that down-regulating pupil diameter actually improves performance relative to having no additional task. The latter would be required in order to draw wider conclusions about *improving* task performance or *facilitating behavioral responses* through modulating arousal. Any discussion of this limitation would be welcome.

We agree with the reviewer that we cannot draw wider conclusions about generally improving task performance by pupil downregulation. This question would be interesting to address in future studies and we added this limitation to the discussion of the manuscript on page 14, line 431-433.

- 8) Are participants generally under-aroused (e.g. tired) and can this account for the difference in the ease with which participants improve the ability to up or down-regulate pupil size? That is, if

participants generally have relatively constricted pupils due to under-arousal, is this the reason they are able to dilate them seemingly more easily than constrict them?

Authors' response:

The reviewer addresses an interesting point. Especially during the MRI in a lying position, it is possible that upregulation is easier than downregulation due to a generally under-aroused state. During the oddball task, participants might have had a generally heightened arousal from the additional task, making it easier for the participants to downregulate. A true test for this hypothesis would be either to sleep deprive or stress participants to bring them to a more extreme arousal state.

9) I am unsure if the term 'neurofeedback' (in the title and throughout the manuscript) is appropriate, since the feedback participants receive is not on a neural signal. Perhaps the term biofeedback would be better.

Authors' response:

We agree that our measurement is based on a peripheral physiological signal. We therefore changed the term to "biofeedback (BF)" or "pupil-BF" which is now used throughout the manuscript.

10) The pathway of the LC to the Edinger-Westphal nucleus (discussed on e.g. Page 10): This pathway is known to exist in cats, but to my knowledge has not yet been definitively shown to exist in primates. Any correlation between LC activity and pupil diameter could also come about through shared input to the sympathetic nervous system and LC by the rostral ventrolateral medulla (See discussion in Nieuwenhuis, De Geus & Aston-Jones, 2011, psychophysiology; 10.1111/j.1469-8986.2010.01057.x).

Authors' response:

We thank the reviewer for raising this interesting explanation. We implemented this explanation for a correlation between LC activity and pupil diameter via the rostral ventrolateral medulla to the manuscript results section (page 6, line 159-161) where we introduce the potential link between LC activity and pupil size changes.

11) Methods points / questions:

a) The authors acquired neuromelanin sensitive scans that can be used to delineate participant-specific masks of the locus coeruleus, which would greatly enhance the accuracy of the LC localization. Why did the authors rely on an atlas instead of using these scans?

Authors' response:

A recent study by Mäki-Marttunen et al. (2021, NeuroImage) showed that there is a very high correlation between BOLD signals (during resting state scans) extracted from (i) the LC Atlas-defined mask, (ii) a group-average of LC Neuromelanin-Hand drawn masks, and (iii) from individual LC Neuromelanin-Hand drawn masks. Based on these findings, we decided to consistently use Atlas-defined masks for all brainstem ROIs. Further, we additionally inspected co-registration results

between anatomical and functional scans very carefully in FreeSurfer to assure valid results using Atlas-defined masks. Overall, we are confident that our results reflect activity in our pre-defined ROIs including the LC, even though we did not quantify that in further detail for the latter.

- b) Were correlations between pupil diameter and fMRI activity adjusted for delays due to the pupillary response? Simply convolving the pupil signal with a canonical HRF will mean that the modeled hemodynamic signal lags the observed hemodynamics because the pupil signal itself lags neural activity. The HRF-convolved signal is typically shifted backwards in time because of this (see e.g. Yellin, Berkovich-Ohana & Malach, 2015, NeuroImage; 10.1016/j.neuroimage.2014.11.034).

Authors' response:

We thank the reviewer for raising this issue. Following the suggestions, we changed our analyses accordingly and shifted pupil diameter time series to account for delays of the pupillary response. Based on animal recordings stimulating LC and measuring pupil dilation responses (Reimer et al., 2016) and the suggested human neuroimaging studies measuring simultaneous neuronal or hemodynamic responses and pupil size (Yellin et al., 2015 and Pfeffer et al., 2022), we shifted pupil time series by 1s to account for known delays. The new results are now reported in the manuscript and, overall, the correlation appears to be a bit stronger than in our previous analysis. We adjusted the methods (page 29, line 934/935) and results section accordingly (including Figure 3 and 4; page 7, line 177-182; line 186-188; page 8, line 212).

- c) Page 28: "Missing data was replaced by the group mean so as not to further reduce statistical power". This does not seem appropriate because it artificially inflates the degrees of freedom.

Authors' response:

*We agree with the reviewer that it's not ideal to replace the missing data of one session for one participant with the mean. We repeated all data analyses without replacing data leading to similar results as before (significant main effect of condition, significant condition * session interaction):*

*ANOVA on heart rate data during training: condition*session interaction; $F(2.336,30.367) = 3.367$, $p = 0.04$; main effect condition: $F(1,13) = 21.352$, $p < .001$.*

ANOVA on differences in RMSSD during training: $F(2.54,20.29) = 0.3419$, $p = 0.76$

Correlations between pupil data and cardiovascular data: with heart rate during no feedback before pupil-BF training: $\rho = 0.051$; $p = 0.86$; with RMSSD during no feedback before pupil-BF training: $\rho = -0.32$, $p = 0.26$

We changed all reported results in the manuscript respectively (including Figure 5, page 8/9, line 225-238).

- d) Spatial smoothing was applied in the analysis of brainstem signals, which is suboptimal because it smears activity across adjacent nuclei, as well as across the ROIs and 4th ventricle.

Authors' response:

We indeed applied smoothing (with a reduction of the default extent to 3 mm). We chose this strategy to find a good balance between increasing the signal-to-noise ratio by smoothing, while keeping anatomical specificity sufficiently high. However, we share the concern of the reviewer, particularly for the LC which is in very close proximity to the 4th ventricle. Therefore, we conducted additional brainstem analyses without applying any spatial smoothing (methods: page 28, line 906-909; results: page 7, line 175/176, line 182). We repeated the key ROI analyses of up- versus downregulation for our pre-defined brainstem nuclei and also for our new brainstem control analyses (i.e., complete brainstem mask and 4th ventricle). We further repeated our correlation analyses using pupil diameter as a regressor with the introduced 1s-shift (see above). These analyses using unsmoothed data confirmed our main findings reported in the manuscript with similar patterns of results and are now summarized in Supplementary Figure 6.

Other points:

Page 17: "Our results reveal that self-regulating pupil diameter as a proxy of LC-mediated arousal facilitates adequate behavioral responses as predicted by current theories of noradrenergic function". I am not sure this is entirely correct. Prominent accounts (e.g. Adaptive Gain theory) predict an inverted-U relationship between arousal and performance rather than a linear one.

Authors' response:

We thank the reviewer for raising this point. We added further discussion about the Adaptive Gain theory and its inverted-U-shape predictions at a later stage in the discussion section. Here, we simply rephrased the sentence on page 11, line 316/317 to "self-regulating pupil diameter as a proxy of (LC-mediated) arousal influences behavioral responses (...)" to reduce the emphasis of facilitating influences and to keep it more general.

Page 18: "healthy adults can learn to self-regulate pupil size during a 3-days training with veridical pupil-NF but not with sham". The results in figure 2 suggest that even with sham feedback there is some effect (see point 5 above).

Authors' response:

We agree with the reviewer and changed our interpretation accordingly on page 11, line 321-323.

This reads now:

"We found that healthy adults can learn to self-regulate pupil size during a 3-days training. This ability was significantly reduced when receiving no veridical feedback (Fig. 2, Supplementary Figures 3 and 4)."

Page 20 and further: "Our behavioral findings further confirm previous studies reporting enhanced task performance with a relatively low baseline pupil diameter ...". The findings in this literature are rather mixed, with some studies reporting effects in this direction, but many others reporting effects in the opposite direction (examples below, no need to cite them). Perhaps it's best to simply report the findings as they are rather than make them appear in line with the broader literature.

- Kristjansson et al (2009) Detecting phasic lapses in alertness using pupillometric measures. Appl

- Ergon 40: 978-986.
- Smallwood et al. (2011) Pupillometric evidence for the decoupling of attention from perceptual input during offline thought. PLoS One 6: e18298.
 - Smallwood et al. (2012) Insulation for daydreams: a role for tonic norepinephrine in the facilitation of internally guided thought. PLoS One 7: e33706.
 - Franklin et al. (2013) Window to the wandering mind: pupillometry of spontaneous thought while reading. Q J Exp Psychol 66: 2289-2294.
 - Grandchamp et al. (2014) Oculometric variations during mind wandering. Front Psychol 5: 31.
 - Mittner et al. (2014) When the brain takes a break: a model-based analysis of mind wandering. J Neurosci 34: 16286-16295.
 - Hopstaken et al. (2015) A multifaceted investigation of the link between mental fatigue and task disengagement. Psychophysiology 52: 305-315.
 - Hopstaken et al. (2015) The window of my eyes: Task disengagement and mental fatigue covary with pupil dynamics. Biol Psychol 110: 100-106.
 - Unsworth and Robison (2016) Pupillary correlates of lapses of sustained attention. Cogn Affect Behav Neurosci 16: 601-15.
 - van den Brink et al. (2016) Pupil diameter tracks lapses of attention. PLoS One 11: e0165274
 - Konishi et al. (2017) When attention wanders: Pupillometric signatures of fluctuations in external attention. Cognition 168: 16-26
 - Unsworth and Robison (2018) Pupillary correlates of fluctuations in sustained attention. J Cog Neurosci 30: 1241-1253

Authors' response:

We thank the reviewer for this valuable input. Our results are in line with some of these previous studies (e.g., studies that showed – like us – slower responses and more task-unrelated thoughts when baseline pupil size was larger), but not necessarily with all of them. Please note that some of the divergent results were obtained for sustained attention and vigilance tasks that are different from the oddball task (e.g., van den Brink et al., 2016; Unsworth and Robison; Kristjanson et al., 2009). Nevertheless, we rephrased the respective paragraph in the discussion section on page 14, line 433-443 to be more precise and take a possible inverted-U-shape relationship between arousal levels indicated by baseline pupil size and task performance better into account.

Reviewer #2:

Remarks to the Author:

Through the eye to the brain: Modulating locus coeruleus-mediated arousal via pupil-based neurofeedback

Summary: the authors describe a series of experiments involving pupil-based neurofeedback in healthy individuals. In the first experiment, they show that pupil-based feedback can be used to regulate pupil size in a group of 27 individuals compared to a sham control group of the same size receiving yoked feedback explicitly described as sham to the participants. Participants trained for 3 sessions. The feedback was used to regulate in both directions. Participants were provided with suggested mental strategies to use for up-regulation as well as down-regulation. No feedback trials performed pre- and post-training showed a significant improvement in the feedback group compared

to sham. In a second experiment, the authors took participants from the feedback group and performed fMRI of the whole brain and brainstem independently during self-regulation to observe which parts of the brain were correlated with such control. Using uncorrected statistics but with independent component analysis and preprocessing techniques intended to remove physiological artifact, the authors found several brainstem areas that correlated with up-regulation vs. down-regulation hypothesized earlier to be physiologically relevant to the task. The researchers found that pupil size self-regulation was correlated with a change in heart rate. In a third experiment, the authors used an auditory oddball task on those who performed pupil size feedback training to investigate whether such stimuli affect pupil dilation response to different sounds. They found that indeed, the ability to self-regulate pupil size was affected by the oddball task, an indication that locus coeruleus was involved in self-regulation. The authors conclude that pupil size can be self-regulated, that a network of salient brain areas are involved in this process, particularly the locus coeruleus, and that such self-regulation is correlated with physiologically relevant behaviors such as heart rate. Overall, I believe this is very solid and innovative work. The experiments were well-planned out and the results are mostly convincing. There are some questions, however:

Major issues:

- The bidirectional control of pupil size is very convincing. My biggest concern here is the sham condition. If I understand correctly, the authors inform the participants that the sham is indeed sham feedback, correct? This is not typical in experiments using such sham that I am familiar with and could potentially result in reduced attention of the sham group and thus worse performance. It would appear that the authors did not want to induce “illusory correlations” (line 650) between the sham feedback and the pupil size, but I believe this is exactly the purpose of sham feedback. Could the authors clarify their intent here and defend the use of this control?

Authors' response:

We thank the reviewer for raising this point. Our participants were not informed that they were in a control group or that they received sham feedback, they were simply not aware of a “true feedback group” (as recommended by Sorger et al., 2019, Neuroimage). We chose this control group to test whether biofeedback would improve performance relative to applying the same mental strategies without any feedback while matching the visual input.

Indeed, this strategy was used to avoid the induction of illusions (i.e. participants think the signal is related to them). Additionally, we aimed to control for other aspects in accordance with good-practice guidelines as published by Sorger et al., 2019; NeuroImage (Table 1; please see colored rectangles added around the respective control implementation in the Table 1: bi-directional pupil-BF group (in purple); control group I (in pink)). However, we realize that we did not explain the setup of our control group in an intuitive way. Therefore, we rephrased the methods (page 19, line 583-595) and results (page 4, line 96-100) section including Figure 1 to avoid possible confusion for the readers.

Table 1. Explicit Control Conditions Commonly Employed in Neurofeedback Studies and the Confounding Factors they Address.

Factors to be controlled for to establish causality	No control	No-training Control		Individual-Regulation Control	Placebo Control				Mental Rehearsal Control	
		Treatment as usual*	List of matched participant Walter †		Alternative feedback		Sham feedback		Inside the fMRI scanner	Outside the fMRI scanner
					Feedback from an Alternative Brain Signal	Feedback based on task-brain signals	Yoked feedback	Artificially generated feedback		
Equal Motivation/ Disruption of Success	✗	✗	✗	✓	✓ ⁺	✓ ⁺	✓ ⁺	✓ ⁺	✗	✗
Demonstrate neurophysiological specificity	✗	✗	✗	✓	✓	✗	✗	✗	✗	✗
Exclude placebo effects	✗	✗	✗	✓	✓ ⁺	✓ ⁺	✓ ⁺	✓ ⁺	✗	✗
Exclude global (possibly non-specific) effects	✗	✗	✗	✓	✓	✓	✓	✓	✓	✗
Exclude Behavioral effects	✗	✗	✗	✗	✗	✗	✗	✗	✓	✓
Remarks	Extremely economical	Economical Crucial (and fastest) control group in clinical context	Very economical	Up- and downregulation of the same region might not always be possible Not always ethically justifiable	Region signal properties should be matched across groups Might be too conservative: cortical region might be trained via functional connectivity Risk of unbinding participants	Risk of unbinding participants	Risk of unbinding participants	Should be generated considering properties of the hemodynamic response Risk of unbinding participants	Not always possible (implicit neurofeedback) Control participants should not be aware of the existence of the neurofeedback group	Not always possible (explicit neurofeedback) Can considerably minimize scanning costs Control participants should not be aware of the existence of the neurofeedback group

* if ability to control is balanced across groups
 † if subjects are not blinded by signal not matching mental changes
 ‡ if control signal does not have an antagonistic relationship with the behavioral/clinical variable of interest.
 + signifies a situation in a clinical context.

Further, we understand the concern of the reviewer and decided to conduct an additional control experiment in which participants received sham feedback (i.e., yoked feedback) but underwent the same amount of training, got the same instructions on mental strategies and on biofeedback as the veridical biofeedback group (indicated by the orange circle in Table 1; methods: page 19, line 593-608, page 20, line 618-619, page 21, line 660-663, 670-673). Thus, in contrast to our previous control group, this additional control group believed that they receive true feedback on their own performance. We recruited 16 participants (age range 20-44; 13 females) for our second control group, a sample size which allows us to discover an effect size of ~0.5 with a power of 95% (note that this is half the size of the first control experiment). Additionally, we measured ECG on all three days of training to be able to analyze heart rate and heart rate variability in a control group.

Comparing pupil modulation indices (differences in baseline-corrected pupil size changes between up- and downregulation) between this second control group and our previous biofeedback group revealed a significant main effect of group ($F(1,20.26) = 19.80, p < .001$; Supplementary Figure 3C) with larger pupil modulation in the pupil-BF than in the second control group. Thus, taken together, acquiring data from a second control group with the aim to keep motivational and perceived success factors similar as in our pupil-BF group in general confirmed our previous results of control group I reported in the main manuscript.

Due to word restrictions, we included the second control group mainly as supplementary information (results: page 5, line 120-122, Supplementary Figure 3A-C). Pupil time series data for up- and downregulation during the three days of training as well as the pupil modulation index time series (up- down) at the beginning (day 1) and after training (no feedback) is depicted as well in Supplementary Figure 3A and B.

- My experience with uncorrected statistics in fMRI data has been that there is no true rule of thumb, which is very frustrating because then a paper comes along that tries to set new standards and some fall below those standards. In lines 474-482, the authors explain their use of physiological noise correction and it's clear they did the best they could, but I still feel uncomfortable with uncorrected statistics. I would like to see the authors provide some objective defense of this, i.e. not that someone else did it.

Authors' response:

We follow the argument of the reviewer and decided to only report low-threshold, cluster-corrected results ($z > 2.3$; $p < .05$) in the main manuscript (methods, page 30, line 948/949). Since there were only significant clusters for the pupil covariation analysis (shifted by 1s; results page 7, line 186-188) but not for the up>down contrast in the brainstem, we decided to report the previous uncorrected results of the up>down contrast in a rather qualitative way in the supplementary information (Supplementary Figure 5D, Supplementary Table 2A) but not in the main manuscript anymore (results page 7, line 184-186, change in Figure 3).

- Not really a major issue, but something that needs to be corrected – “The effectiveness of self-regulation critically depends on whether participants can acquire a suitable mental strategy”. I would disagree with this statement as a number of neurofeedback studies including those from the group of Mitsuo Kawato were able to show the ability to self-regulate a prescribed set of voxels without any relevant mental strategy.

Authors' response:

We thank the reviewer for this input. We agree that some self-regulation studies don't rely on explicit, verbalizable mental strategies but that rather implicit learning is taking place. Therefore, we changed the respective sentence in the introduction and emphasized that these processes can also take place implicitly (page 4, line 77-78).

- I do not recall seeing any stats table on Down > Up contrasts. Is there anything to be learned there?

Authors' response:

We thank the reviewer for raising this point. The Down > Up contrasts are only presented as Supplementary Figure (5E-G) as well as in the Supplementary Table 3B with statistical information. In our opinion, it does not necessarily add more information to the main story in the manuscript. If wished, we can move this information to the main manuscript though.

- A better lead in to the purpose behind Experiment 3 would be appreciated, i.e., explaining the significance of the modulation of LC of auditory oddball task. Reading through the rest of the manuscript it becomes clear, but not until then.

Authors' response:

We thank the reviewer for pointing that out. We tried to explain the significance of pupil- and LC modulation for the auditory oddball task better and moved some parts of the results section of Experiment 3 to the introduction (e.g., page 3, line 54-57; page 4, line 71-75). We hope that our

modifications help the reader to better understand the reasoning behind Experiment 3.

- I question the use of p-values for correlations in Fig 6F as they mean less when number of data points increase. Given the amount of data, the regression value is more informative and 0.13 seems very weak. I'm not sure I'm on board with the conclusion of pupil dilation being related to baseline without some extra bootstrapping here.

Authors' response:

We agree that with the large amount of data points, a p-value is not very informative anymore. For the relationship between pupil baseline size and pupil dilation response, we report an r of -0.3 in the original manuscript which is a medium effect size. However, please see the changes implemented following the suggestion of reviewer 1 to exclude the possibility of a potential bias between the relationship of a variable and an increase change from this variable itself (i.e. relating baseline pupil size to a baseline-corrected pupil dilation response).

For the relation between baseline pupil size and reaction times, however, we indeed only reach an r of 0.13 (without considering potential time-on-task effects mentioned by reviewer 1 that we now take into account in additional analyses. This raises the r to 0.16) which is indicating only a weak relationship. Even when we apply bootstrapping with the repeated measures correlation, the results remain the same since our data is approximately normally distributed. However, we tried to make it clearer in the results section on page 11, line 304-305 that even though we find a significant relationship between baseline pupil size and reaction time, this relationship is only weak on a single-trial level.

- In line 752, the authors use mean data to replace missing data. My experience is that this is not an acceptable practice, but I would like to hear from the authors on why this was necessary.

Authors' response:

*Reviewer 1 raised the same point, and we thank the reviewers for this input. We agree that replacing missing data by mean values artificially decreases the standard deviation and increases the degrees of freedom. We repeated all data analyses without replacing data leading to similar results as reported before (significant main effect of condition, significant condition * session interaction):*

*ANOVA on heart rate data during training: condition*session interaction; $F(2.336,30.367) = 3.367$, $p = 0.04$; main effect condition: $F(1,13) = 21.352$, $p < .001$*

*ANOVA on differences in RMSSD during training:
 $F(2.54,20.29) = 0.3419$, $p = 0.76$*

Correlations between pupil data and cardiovascular data: with heart rate during no feedback before pupil-BF training: $\rho = 0.051$; $p = 0.86$; with RMSSD during no feedback before pupil-BF training: $\rho = -0.32$, $p = 0.26$

We changed all reported results in the manuscript respectively (page 8/9, line 226-238, see also

Figure 5).

- I'm a little skeptical of the conclusion between pupil NF and heart rate. Fig 5E shows a good correlation between pupil modulation index and change in heart rate only after training, however, it may not be the feedback training, but the repeated exposure to the explicit mental strategies. I would think having a differential correlation with controls here would be more convincing.

Authors' response:

That is a very interesting point, and we haven't really considered this opportunity until now. To be able to address this point, we measured ECG in our additionally acquired second control group (since we did not have heart rate data recorded in control group 1) and correlated differences between up- and downregulation trials in heart rate with the pupil modulation index (up-down) at the beginning (i.e., on day 1) and at the end of training (i.e., on day 3). Interestingly, similarly to our pupil-BF group, and as already assumed by the reviewer, we found a significant correlation between pupil and heart rate data on day 3 (Spearman's $\rho = 0.65$; $p = 0.016$) which was similar in effect size as in the pupil-BF group ($\rho = 0.63$). On day 1 prior to training, the relationship between pupil and heart rate was negative (i.e., bigger difference in pupil \rightarrow smaller difference in heart rate) and did not reach significance (only trend-level: $\rho = -0.46$; $p = 0.08$). These findings may indeed indicate that this relation is rather due to repeated exposure to the application of mental strategies (results: page 9, line 243-246; Supplementary Figure 3F). We rephrased our discussion section accordingly to take these findings into account (page 15, line 458-462).

For additional analyses on heart rate and heart rate variability data throughout the training of control group 2, please see Supplementary Figure 3D-F.

- I'm a bit concerned about the change in sequence mentioned in Line 815. How do we know the change in sequence didn't affect results? Was there any subsequent analysis?

Authors' response:

We understand the concern of the reviewer. Generally, the two EPI sequences were still largely overlapping regarding the main parameters. Further, there is a paper by the groups of Felix Blankenburg and Nikolaus Weiskopf that explicitly tested the influence of different EPI sequences on the results of random effects fMRI analyses and found that inter-subject variability had the largest impact on these results while differences in the EPI sequences had a minor impact on the results at a group level (Kirilina et al., 2016; NeuroImage).

Regarding our own data, additional control analyses indicate that the choice of EPI sequence had no systematic effect on the results:

First, we show in the following figure the z-values in up- vs downregulation (top) and down- vs upregulation (bottom) extracted from peak voxel activity for the significant clusters for the old vs. the new sequence separately. Overall, we observed a highly similar pattern. Further, statistical analyses (repeated measures ANOVAs) on these values comparing up- and downregulation for participants with old and new sequences revealed only significant main effects of self-regulation (i.e., up vs. down; all F

> 18.27, all $p < .001$) but no significant main effects of the sequence and/or an interaction effect between the type of sequence and modulation condition for each of the eight clusters (seven clusters up vs down; one cluster down vs. up; all $p > .11$). We hope that these additional analyses can mitigate the concerns of the reviewer.

Up vs. Down Clusters:

Down vs. Up Cluster:

Minor issues:

- Line 290, should it be “trended towards significance”? “tended” does not seem right here.
- Line 333, “suggests”
- Line 392, comma instead of decimal in “94.3”
- Line 460, no comma needed after “diameter”
- Line 756, define “RMSSD”

Authors’ response:

We thank the reviewer for this valuable input and changed all mentioned issues in the manuscript accordingly.

Reviewer #3:

Remarks to the Author:

The authors present evidence in support of the hypothesis that humans can use pupil-based closed-loop feedback to volitionally up and down regulate pupil size and activity in arousal-related brain structures including the Locus Coeruleus.

Evidence is provided in three parts: First, through two experiments set up as between- (Exp. 1-A) and within-subject (Exp. 1-B) design respectively, for which the authors found significant differences in pupil-diameter and heart rate (but not heart rate variability) as a function of closed-loop pupil-feedback condition. Second, through a study that uses functional magnetic resonance imaging (Exp. 2), for which the authors present significant pupil-feedback-related effects in the brainstem as well as the whole brain. Third, through behavioral effects that the authors observed while subjects self-regulated their pupil-size while performing an odd-ball task (Exp. 3).

In my assessment, the evidence in support of the authors' hypothesis that humans can volitionally regulate pupil size and activity in the brainstem is convincing and represents a significant scientific advancement. I believe that this manuscript will be of great interest to readers interested in closed-loop feedback and other disciplines.

I could not find any major flaws around data, methods or technical details. I provide some more detailed questions and comments below, including about how sham feedback was generated. While I could not find information related to pre-registration for the studies that were performed, I also could not find any indication for any major concerns about how the studies were conducted. In my assessment, all statistical methods used, have been applied correctly and all analyses were appropriately powered.

The authors correctly interpret individual test outcomes and integrate the evidence from across experiments and approaches appropriately. What adds to the confidence in the robustness of these findings is that the authors present evidence that includes between- as well as within-subject experiments, replication in independent samples of novice participants, multiple physiological signals including pupil size, electrocardiography and fMRI as well as behavioral results. Ultimately, it remains difficult to answer this research question with absolute certainty, which is a topic the authors address appropriately as part of the limitations of this study in the discussion. The manuscript is very well written and from what I can tell cites all necessary literature appropriately.

More minor comments and questions:

Could the authors please clarify what "quasi real-time" means concretely?

Authors' response:

With "quasi real-time" we refer to the delay of displaying the feedback to participants about their own pupil size due to filtering/online preprocessing and averaging of consecutive samples of pupil size (with a sampling rate of 60 Hz, averaging alone amounts to a delay of at least 16-32 ms). That's why we prefer to use the term "quasi real-time" instead of real-time. We addressed this now more clearly in the manuscript on page 17, line 519/520.

L104: Could you clarify concretely what is meant when you write that the control group “did not receive veridical feedback”? How exactly was the sham feedback generated? If it helps, please provide equations, to make this important topic unambiguously clear for the reader.

Authors' response:

We thank the reviewer for this comment since it's indeed important to make it absolutely clear to the reader how the feedback for the control group was generated. In principle, we replayed the feedback of the participants of the veridical biofeedback group – thus yoked feedback (Sorger et al., 2019; Neuroimage). Every control participant was randomly matched to a participant of the biofeedback group and saw the exact same feedback as the biofeedback participant. We chose this approach to make sure that our visual input (even though the choice of color was isoluminant to the gray background of the screen) is not artificially driving pupil size.

We added a clarification to the manuscript (methods: page 19, line 583-595; results: page 4, line 96-100) and hope that the way we generated the sham feedback got clearer now.

L130: I have found it a bit difficult to understand how the pupil-modulation index is defined. Could you maybe include the formula?

Authors' response:

We added more detailed information together with a formula about the pupil modulation index in the methods section of Experiment 1A on page 20, line 643-651. We hope this clarifies how we calculated pupil modulation indices.

L274: Why was ECG recorded for only 15 subjects? If subjects were excluded, e.g. due to insufficient data quality, please explain the details (unless they are already included in the report to the publisher).

We did not exclude ECG data of subjects not reported in the manuscript - only bad data of 1 subject prior to training and during session 1 as reported in the methods section. Unfortunately, we indeed recorded ECG data in only 15 participants of our second pupil-BF cohort due to technical issues with our old ECG system within the first session and subsequent acquisition of a new system. Since our main goal was to replicate pupil results of Experiment 1A, we decided to continue data collection for the sample and added ECG measurements only at a later stage.

L286: Where you explain that RMSSD is or may be linked to changes in activity of the parasympathetic nervous system, could you add the missing reference? Please consider trying a different short-term HRV metric, called pNN-X.

Here, a paper that uses pNN-X:

Mietus, J. E., Peng, C. K., Henry, I., Goldsmith, R. L., & Goldberger, A. L. (2002). The pNNx files: re-examining a widely used heart rate variability measure. *Heart*, 88(4), 378-380.

Here, additional literature on interpreting short-term HRV:

Berntson, G. G. & Cacioppo, J. T., (2004) Heart rate variability: Stress and psychiatric conditions. In Malik, M. & Camm, A. J. (eds.), Dynamic electrocardiography, 57–64 (John Wiley & Sons).

Malik, M., & Camm, A. J., (1993). Components of Heart Rate Variability – What they really mean and what we really measure. *Am J Cardiol*, 72, 821-822.

Here, another paper where the metric was used for a similar purpose:

Faller, J., Cummings, J., Saproo, S., & Sajda, P. (2019). Regulation of arousal via online neurofeedback improves human performance in a demanding sensory-motor task. *Proceedings of the National Academy of Sciences*, 116(13), 6482-6490.

Authors' response:

We thank the reviewer for pointing out the missing reference. We added it accordingly on page 8, line 234 and page 22, line 693.

As suggested, we calculated the pNN35 (see Faller et al. 2019) for the pupil-BF training and fMRI data, respectively.

Using this metric, we observed significant differences between up- and downregulation phases during training calculating non-parametric Wilcoxon signed rank tests (training data difference between up and down averaged across all training sessions: $z = 2.84$; $p = 0.005$) and after training during both fMRI sessions (whole-brain Wilcoxon signed rank test: $z = 3.057$; $p = 0.002$; brainstem Wilcoxon signed rank test: $z = 3.371$; $p = 0.002$; sequential Bonferroni corrected) with larger HRV during down- than upregulation. Similar to reported RMSSD during training in our main manuscript, we did not find any significant effects in a robust repeated measures ANOVA on the difference in pNN35 between up- and downregulation throughout training ($p = 0.33$; a significant result would indicate a learning effect throughout training). We further calculated whether the pNN35 was related to the pupil modulation index. However, similar to RMSSD values, we did not find a significant correlation between this HRV measure and the pupil modulation index neither pre ($\rho = 0.10$; $p = 0.73$) nor post training during fMRI ($\rho = 0.27$; $p = 0.19$).

Using the pNN35 instead of the RMSSD revealed an equivalent result pattern but the statistical analysis revealed more robust results which reached significance. We now report the respective analyses in the manuscript on page 24, line 752-756, and page 8/9, line 233-241 and in Supplementary Figure 7. However, we are still cautious to over-interpret this data calculated on time windows as short as 15s, since RMSSD is still preferred over the pNN-xx by most researchers. This might be potentially due to the good validation of RMSSD for ultra-short-term measurement periods and the rather arbitrary thresholding of the pNN-xx (Shaffer & Ginsberg 2017).

Even though we believe that RMSSD may be the most adequate measure to estimate HRV in our

study, a general prolongation of self-regulation may lead to more robust and reliable results. In another experiment carried out in our lab where we introduce 30s of pupil self-regulation, we indeed see a clear effect on HRV estimated via RMSSD (unpublished data). Here, pupil downregulation was associated with higher RMSSD values at the end of training than upregulation (Wilcoxon Signed Rank Tests: $z = 2.971$; $p = 0.003$).

L343: Where you write “714-3000ms”; if you intend to say “714 to 3000 ms”, then please consider using the latter way to write this, as a best practice and to avoid confusion with a minus sign.

Authors’ response:

We thank the reviewer for raising this concern, we changed it in all appearances to avoid confusion.

L483: Faller and colleagues provided feedback based on EEG and observed changes in pupil size that were interpreted as linked to LC activity – could that study (with some caveats) be seen as supporting the idea you formulate at the end of this paragraph?

Authors’ response:

We thank the reviewer for this input and the proposed link to the study of Faller et al. 2019. Indeed, the results of the study could be in line with this idea. We hope that future studies can reveal whether the nature of observed cortical activity is causal or consequential to arousal modulation. We now added the citation of Faller and colleagues to the discussion in a slightly different context (page 15, line 452-453).

L516, paragraph: A similar effect was observed by Faller and colleagues for pupil size in the sham condition by Faller and colleagues (see Figure F, Panel B).

Authors’ response:

We thank the reviewer for making the link between the two results. We added it to the paragraph in the discussion on page 15, line 452-453.

L568: The sentence starting with “Developing an eye blink-related strategy...” might be correct, but seems unnecessarily hard to parse – please consider to rephrase it.

Authors’ response:

We thank the reviewer for catching this sentence that is hard to parse. We rephrased it to make it easier to understand.

L573: What was the luminance setting in the room? Were these studies conducted in an office environment or inside a specialized noise-shielded room or box?

Authors’ response:

We conducted all experiments (except for the fMRI experiments) in a faraday cage with dim light conditions that was placed within lab space with no windows. We went for this setting to have a shielded room with constant lighting conditions and no noise to allow participant to focus on the experiment. We added this information to the general methods section of the manuscript (page 16,

line 483/484).

L611: Could you comment on the degree of fidelity you expect from how you calculate luminance from RGB color values via the formula you mention. My understanding is, that using a formula such as this is seen as a crude approximation of luminance. If appropriate, please consider discussing this topic as a limitation.

Authors' response:

We thank the reviewer for raising this important point. Indeed, this formula takes into account that green light contributes the most while blue light contributes the least to perceived intensity – but this can still slightly differ from individual to individual. Another factor we kept constant was the number of pixels shown on the screen through our online feedback. Furthermore, we introduced post-trial feedback only (without constant feedback during pupil self-regulation) in our replication study on day 3 as well as during oddball and fMRI recordings, so that we can assume that the observed effects are not driven by changes in luminance introduced by the used colors.

We added this information and concern in more detail to discussion sections on page 12, line 341-347.

A photo or better even video of the paradigm – maybe as part of the supplementary material - would be very helpful.

We uploaded a video of two exemplary trials of the online feedback paradigm as part of the supplementary information. We hope that this helps to better understand the trial routine of the paradigm.

Generally, I recommend to use box- or violin rather than bar plots.

Authors' response:

We implemented the changes as suggested (Figure 2C, Figure 3C, Figure 4D, Figure 5AB, and Figure 6CE) except for Figure 3B and 5CD since we wanted to put emphasis on individual changes from one condition to the other here.

Could you clarify whether these studies were conducted in a double-blind manner where appropriate? If appropriate please consider discussing this as a limitation.

Authors' response:

The participants of Experiment 1A were blinded but we did not blind the experimenter. However, the experimenters took special care to provide identical instructions to the participants and all measurements were run automatically, with the participant sitting alone in a shielded room without any interference from the experimenter. Furthermore, data preprocessing was done in an automatized way for all conditions and participants together, i.e. not considering whether a dataset belongs to the control and pupil-BF group. Overall, we believe that it is unlikely that there was an observer bias.

We added this information to the manuscript on page 12, line 347-351.

Decision Letter, first revision:

18th August 2023

Dear Dr. Meissner,

Thank you for your patience as we've prepared the guidelines for final submission of your Nature Human Behaviour manuscript, "Self-regulating arousal via pupil-based biofeedback" (NATHUMBEHAV-23010116A). Please carefully follow the step-by-step instructions provided in the attached file, and add a response in each row of the table to indicate the changes that you have made. Please also address the additional marked-up edits we have proposed within the reporting summary. Ensuring that each point is addressed will help to ensure that your revised manuscript can be swiftly handed over to our production team.

We would hope to receive your revised paper, with all of the requested files and forms within two-three weeks. Please get in contact with us if you anticipate delays.

Nature Human Behaviour offers a Transparent Peer Review option for new original research manuscripts submitted after December 1st, 2019. As part of this initiative, we encourage our authors to support increased transparency into the peer review process by agreeing to have the reviewer comments, author rebuttal letters, and editorial decision letters published as a Supplementary item. When you submit your final files please clearly state in your cover letter whether or not you would like to participate in this initiative. Please note that failure to state your preference will result in delays in accepting your manuscript for publication.

In recognition of the time and expertise our reviewers provide to Nature Human Behaviour's editorial process, we would like to formally acknowledge their contribution to the external peer review of your manuscript entitled "Self-regulating arousal via pupil-based biofeedback". For those reviewers who give their assent, we will be publishing their names alongside the published article.

Cover suggestions

As you prepare your final files we encourage you to consider whether you have any images or illustrations that may be appropriate for use on the cover of Nature Human Behaviour.

ORCID

Non-corresponding authors do not have to link their ORCIDs but are encouraged to do so. Please note that it will not be possible to add/modify ORCIDs at proof. Thus, please let your co-authors know that if they wish to have their ORCID added to the paper they must follow the procedure described in the following link prior to acceptance:

Nature Human Behaviour has now transitioned to a unified Rights Collection system which will allow our Author Services team to quickly and easily collect the rights and permissions required to publish your work. Approximately 10 days after your paper is formally accepted, you will receive an email in providing you with a link to complete the grant of rights. If your paper is eligible for Open Access, our Author Services team will also be in touch regarding any additional information that may be required to arrange payment for your article.

Please note that *Nature Human Behaviour* is a Transformative Journal (TJ). Authors may publish their research with us through the traditional subscription access route or make their paper immediately open access through payment of an article-processing charge (APC). Authors will not be required to make a final decision about access to their article until it has been accepted. Find out more about Transformative Journals

Authors may need to take specific actions to achieve compliance with funder and institutional open access mandates. If your research is supported by a funder that requires immediate open access (e.g. according to Plan S principles) then you should select the gold OA route, and we will

direct you to the compliant route where possible. For authors selecting the subscription publication route, the journal's standard licensing terms will need to be accepted, including self-archiving policies. Those licensing terms will supersede any other terms that the author or any third party may assert apply to any version of the manuscript.

[REDACTED]

Best regards,
Alex McKay
Editorial Assistant
Nature Human Behaviour

On behalf of

Giacomo Ariani
Editor
Nature Human Behaviour

Reviewer #1:

Remarks to the Author:

The authors did an excellent job revising the manuscript and have addressed all of my comments. I now support publication of the manuscript in its current form.

Reviewer #2:

Remarks to the Author:

Thanks to the authors for addressing all of my comments. I'm satisfied with these changes.

Reviewer #3:

Remarks to the Author:

The authors were able to address my questions/comments.

One remaining minor point: I believe there may be a typo in a newly added equation as described below.

L646: Variable n , is defined as the 15000 data points, therefore n seems appropriate for the denominator. For the numerator, from what I understand, you mean to refer to “pupilsizedup” and “pupilsizedown” at particular locations inside the timeseries. I believe notation where you indicate that you sum over $t=1$ to $t=15000$ for $(\text{pupilsizedup}(t) - \text{pupilsizedown}(t))$ and divide the result by $n=15000$ would best describe what was done.

Author Rebuttal, first revision:

Response to the comment of Reviewer #3:

The authors were able to address all my questions/comments. I believe there may be a typo in a newly added equation as described below.

Minor points:

L646: Thank you for adding the equation for pupil modulation index. I believe there may be a typo in the formula. Variable n , is defined as the 15000 data points, therefore n seems appropriate for the denominator. For the numerator, from what I understand, you mean to refer to “pupilsizedup” and “pupilsizedown” at particular locations inside the timeseries. I believe the following notation would more correctly describe what was done:

$$\text{Pupilmodulationindex} = 15000 \sum_{t=1} (\text{pupilsizedup}(t) - \text{pupilsizedown}(t)) / n$$

Authors' response:

We thank the reviewer for catching this typo. Yes indeed, the numerator must refer to a particular location inside the 15s time series. We corrected it accordingly.

Final Decision Letter:

Dear Dr Meissner,

We are pleased to inform you that your Article "Self-regulating arousal via pupil-based biofeedback", has now been accepted for publication in Nature Human Behaviour.

Please note that *Nature Human Behaviour* is a Transformative Journal (TJ). Authors may publish their research with us through the traditional subscription access route or make their paper immediately open access through payment of an article-processing charge (APC). Authors will not be required to make a final decision about access to their article until it has been accepted. Find out more about Transformative Journals

We welcome the submission of potential cover material (including a short caption of around 40 words) related to your manuscript; suggestions should be sent to Nature Human Behaviour as electronic files (the image should be 300 dpi at 210 x 297 mm in either TIFF or JPEG format). Please note that such pictures should be selected more for their aesthetic appeal than for their scientific content, and that colour images work better than black and white or grayscale images. Please do not try to design a cover with the Nature Human Behaviour logo etc., and please do not submit

composites of images related to your work. I am sure you will understand that we cannot make any promise as to whether any of your suggestions might be selected for the cover of the journal.

With best regards,

Giacomo Ariani
Editor
Nature Human Behaviour